# Flow: Per-instance Personalized Federated Learning

**Kunjal Panchal**
College of Information and Computer Sciences
University of Massachusetts, Amherst
Amherst, MA 01003
kpanchal@umass.edu

**Sunav Choudhary**
Adobe Research
Bangalore, India
schoudha@adobe.com

**Nisarg Parikh**
Khoury College of Computer Sciences
Northeastern University
Boston, MA 02115
parikh.nis@northeastern.edu

**Lijun Zhang, Hui Guan**
College of Information and Computer Sciences
University of Massachusetts, Amherst
Amherst, MA 01003
{lijunzhang, huiguan}@umass.edu

## Abstract

Federated learning (FL) suffers from data heterogeneity, where the diverse data distributions across clients make it challenging to train a single global model effectively. Existing personalization approaches aim to address the data heterogeneity issue by creating a personalized model for each client from the global model that fits their local data distribution. However, these personalized models may achieve lower accuracy than the global model in some clients, resulting in limited performance improvement compared to that without personalization. To overcome this limitation, we propose a per-instance personalization FL algorithm *Flow*. *Flow* creates dynamic personalized models that are adaptive not only to each client's data distributions but also to each client's data instances. The personalized model allows each instance to dynamically determine whether it prefers the local parameters or its global counterpart to make correct predictions, thereby improving clients' accuracy. We provide theoretical analysis on the convergence of *Flow* and empirically demonstrate the superiority of *Flow* in improving clients' accuracy compared to state-of-the-art personalization approaches on both vision and language-based tasks. The source code is available on GitHub [1].

## 1 Introduction

Federated Learning (FL) is a distributed machine learning paradigm that enables edge devices, known as "clients", which collaboratively train a machine learning model called a "global model" [1]. However, because the server, which is the FL training orchestrator, does not have access to or knowledge of client data distributions, it poses a challenge of statistical heterogeneity [2, 3]. This heterogeneity hinders the server's ability to train an ML model on a large quantity and variety of data and also impacts the client's ability to benefit from a generalizable model without sharing any information about its data. To address this challenge, personalization has been studied [4, 5] to improve prediction performance. Recent literature consistently demonstrates that personalized models achieve higher prediction performance than the global model aggregated over clients [5–10]. These approaches typically create a personalized model specific to each client's data distribution, and thus we refer to them as "per-client personalization".

---

[1]https://github.com/Astuary/Flow

37th Conference on Neural Information Processing Systems (NeurIPS 2023).

However, we have identified two factors that limit the performance improvement of existing per-client personalization approaches in terms of clients' accuracy. First, as reported in the evaluation in Sec. 5, we found that personalized models can achieve lower accuracy than the global model on up to 31% clients, causing limited improvement in accuracy averaged across all clients. Second, even if personalized models achieve higher accuracy on a client compared to the global model, they can still produce incorrect predictions on up to 11% of data instances on clients that could be correctly handled by the global model, causing limited accuracy improvement from personalization on that client. These observations reveal a significant drawback of existing per-client personalization approaches: *each client's personalized model is a static network that cannot accommodate the heterogeneity of the local client's data instances.* As a result, every data instance on a client is constrained to use its personalized model for prediction, even though some instances could benefit from the better generalizability of the global model.

To overcome the above limitation, this paper proposes a per-instance and per-client personalized FL algorithm *Flow* via dynamic routing to improve clients' accuracy. *Flow* creates *dynamic* personalized models that are adaptive to each client's individual data instances (per-instance) as well as their data distribution (per-client). In a FL round of *Flow*, each client has both the global model parameters that are shared across clients and the local model parameters that are adapted to the local data distribution of the client by fine-tuning the global model parameters. *Flow* creates a dynamic personalized model per client that consists of local and global model components and a dynamic routing module. The dynamic routing module allows each data instance on a client to determine whether it prefers the local model parameters or their global model counterparts to make correct predictions, thereby improving the personalized model's accuracy on the client compared to that of a local model or a global model. At the same time, through dynamic routing, *Flow* could identify instances in each client that agree with the global data distribution to further improve the performance of the global model, offering a good *starting point* for any new client to personalize from. Since *Flow* is a client-side personalization approach, it can work with server-side optimization methods like FEDYOGI [11].

We theoretically analyze how dynamic routing affects the convergence of the global model and personalized model in *Flow* and also empirically demonstrate the effectiveness of *Flow* in improving clients' accuracy compared to state-of-the-art personalization approaches on cross-device language and vision tasks. For the newly joined clients, the global model from *Flow* achieves 2.81% (Stackoverflow), 3.46% (Shakespeare), and 0.95%-1.41% (CIFAR10) better accuracy on the global model against its best performing baselines respectively. After personalization, the dynamic personalized model from *Flow* sees improvements of 3.79% (Stackoverflow), 2.25% (Shakespeare), and 3.28%-4.58% (CIFAR10) against the best performing baselines. Our in-depth analysis shows that *Flow* achieves the highest percentage of clients who benefit from personalization compared to all baselines and reduces the number of instances that are misclassified by the personalized model but correctly classified by the global model, contributing to the better personalized accuracy from *Flow*.

We summarize the contributions as follows:

- We propose a per-instance and per-client personalization approach *Flow* that creates personalized models via dynamic routing, which improves both the performance of the personalized model and the generalizability of the global model.
- We derive convergence analysis for both global and personalized models, showing how the routing policy influences convergence rates based on the across- and within- client heterogeneity.
- We empirically evaluate the superiority of *Flow* in both generalization and personalized accuracy on various vision and language tasks in cross-device FL settings.

## 2 Related Work

**Personalized Federated Learning.** Personalization in FL has been explored primarily on client-level granularity. APFL [6] interpolates client-specific local model weights with the global model weights which are sent by the server. Meanwhile, *Flow* is an intermediate-output interpolation method and also includes a dynamic policy to interpolate at instance-level granularity. We note that DAPPER [12] interpolates the client dataset with a global dataset, which is impractical for the cross-device use cases FL focused on in this paper. Regularization is another popular way of creating a personalized model. It encourages a personalized model of each client to be close to the global model as explored

in DITTO [13] and PFEDME [14]. Finetuning partial or entire global model to get the personalized model has been studied in LGFEDAVG [7], which finetunes and personalizes the feature extractor and globally shares the classifier head, FEDBABU [15] which freezes the classifier head and only updates and aggregates the feature extractor, and FEDREP [8] which finetunes and personalizes the classifier head and globally shares the feature extractor. Learning personalized representations of the global model has been explored in FEDHN [16] where a central hypernetwork model is trained to generate a personalized model for each client. PARTIALFED [17] makes a layer-wise choice between global and local models to create a personalized model, but it sends the personalized models back to the server in lieu of a separate global model. This method has limitations in terms of training the base global model (on which the personalized model would be based) with FEDAVG, which has been observed to be insufficient against non-iid data [18]. Besides, the strategy does not create a dynamic personalized model during inference. None of the above work has explored instance-level personalization.

Personalized models from previous work KNNPER [19] exhibits per-instance personalization behavior. The personalized model of a client makes a prediction based on features extracted from the global model from an instance as well as features from the instance's nearest neighbors. However, KNNPER trains the global model parameters in the same way as FEDAVG, which has been shown to perform poorly in heterogeneous data settings [18]. In contrast, *Flow*'s dynamic routing mechanism results in a global model with better-generalized performance amidst heterogeneous instances, which ultimately leads to a higher boost in the performance of personalized models as well.

**Dynamic Routing.** Motivated by saving compute effort for "easy to predict" instances, instance-level dynamic routing has been a matter of discussion in works related to early exiting [20–22], layer skipping [23], and multi-branching [24, 25]. SKIMRNN [26] explored temporal dynamic routing where depending on the input instance, a trained policy can determine whether to skim through the instance with a smaller RNN layer (which just updates the hidden states partially), or read the instance with a standard (bigger) RNN layer (updating the entire hidden state). Our work is motivated by the question of *Depending on the models' utility and instances' heterogeneity, which route to pick?*. This can be achieved by using a routing policy to dynamically pick from two versions of a model, which are equivalent in terms of computational cost but different in terms of the data they are trained on.

## 3 Our Approach

We introduce *Flow*, a per-instance and per-client personalization method that dynamically determines when to use a client's local parameters and when to use the global parameters based on the input instance. Table 1 summarizes the notation used in this paper.

Algorithm 1 describes the workflow of *Flow*. During each FL round, the server samples $M$ participating clients with a sampling rate of $p$. Upon receiving the global model $w_g$ and policy parameters $\psi_g$ (Line 3), each participating client personalizes and trains $w_g$ in five major stages: (1) Split the training dataset into two halves: $\zeta_{m,\ell}$ and $\zeta_{m,g}$ (Line 6). $\zeta_{m,\ell}$ will be used to update the parameters in the local model parameters while $\zeta_{m,g}$ will be used to train the parameters in the global model and a routing module. (2) Derive the local parameters $w_\ell$ by finetuning the global parameters $w_g$ for $K_1$ epochs with $\zeta_{m,\ell}$ (Line 7). (3) Construct a *dynamic personalized model* $w_{p,m}$ by integrating the local versions of the global parameters $w_{g,m}$ and policy parameters $\psi_{g,m}$, and the local parameters $w_\ell$. Here, the routing policy $\psi_g$ determines whether the execution

Table 1: Notations

| | |
|---|---|
| $K$ | #Epochs for training |
| $M$ | Total number of sampled clients |
| $m$ | Client index $\in [M]$ |
| $\mathcal{M}$ | Set of available clients |
| $p$ | Client sampling rate |
| $\eta_\ell$ | Local learning rate |
| $w_g$ | Global model |
| $\psi_g$ | Policy module |
| $w_{g,m}$ | Local version of the global model |
| $w_{\ell,m}$ | Local model of $m^{th}$ client |
| $w_{p,m}$ | Personalized model of $m^{th}$ client |
| $\mathcal{D}_m$ | Data distribution of $m^{th}$ client |

path of an instance should use $w_\ell$ or $w_g$. (4) Train the routing policy $\psi_{g,m}$ and the local version of the global parameters $w_{g,m}$ alternatively for $K_2$ epochs (Lines 9-10) with $\zeta_{m,g}$. Although $K_1$ and $K_2$ can be different, we later use $K$ to denote both $K_1$ and $K_2$ for ease of theoretical analysis. (5) Send the local version of the global model parameters $w_{g,m}$ and the policy parameters $\psi_{g,m}$ back to the server for aggregation (Line 14). Aggregation strategies are orthogonal to this work and we adopt FEDAVG [1].

Next, we discuss the design of the *dynamic personalized model* $w_p$ in detail. Figure 1 illustrates the design of $w_p$. In *Flow*, $w_p$ is made of three components, the local model $w_\ell$ and global model $w_g$ and the routing module $\psi_g$ that selects the execution of local or global model layers for each data instance.

**Local Parameters.** We use *finetuning* to get the local parameters $w_{\ell,m}^{(r)}$ in the $r$-th round since finetuning has proven to be a less complex yet very effective method of personalization [27, 28]. Given a global model of the previous round, $w_{g,m}^{(r)}$, we finetune it for $K$ epochs to get $w_{\ell,m}^{(r)}$. This local model would be reflective of the client's data distribution $\mathcal{D}_m$. Note that we use one half of the dataset to get $w_{\ell,m}^{(r)}$ and reserve the other half of the training dataset for updating the global model and the policy (i.e., the routing module) parameters. This is to make sure that the policy parameters, which would decide between local and global layers based on each input instance, are not overfitted in favor of the local parameters. In Figure 1, the local parameters are shaded green, denoted by $w_\ell$. The update rule for $w_{\ell,m}^{(r)}$ is:

$$w_{\ell,m}^{(r,K)} \leftarrow w_{\ell,m}^{(r,0)} - \eta_\ell \sum_{k=1}^{K} \nabla f_m(w_{\ell,m}^{(r,0)}; \zeta_{m,\ell}), \text{ where } w_{\ell,m}^{(r,0)} := w_g^{(r)}. \tag{1}$$

---

**Algorithm 1:** Pseudocode of *Flow*

1   Server randomly initializes the global model $w_g$ and the policy module $\psi_g$
2   **for** *each round* **do**
3      Send $w_g, \psi_g$ to sampled $M$ clients
4      **for** $m \in [M]$ *in parallel* **do**
5          $w_{g,m} \leftarrow w_g; \psi_{g,m} \leftarrow \psi_g; w_{\ell,m} \leftarrow w_{g,m}$
6          Creating two mutually exclusive datasets $\zeta_{m,\ell}, \zeta_{m,g} \leftarrow \mathcal{D}_m$
7          Train $w_{\ell,m}$ for $K_1$ epochs
8          **for** $k \in [K_2]$ *epochs* **do**
9             Update $\psi_{g,m}$ according to Equation 4
10            Update $w_{g,m}$ according to Equation 5
11          **end**
12          Send back $w_{g,m}, \psi_{g,m}, n_m := |\zeta_{m,g}|$
13      **end**
14      Update $w_g$ and $\psi_g$ with weighted average of each client's $w_{g,m}$ and $\psi_{g,m}$
15 **end**

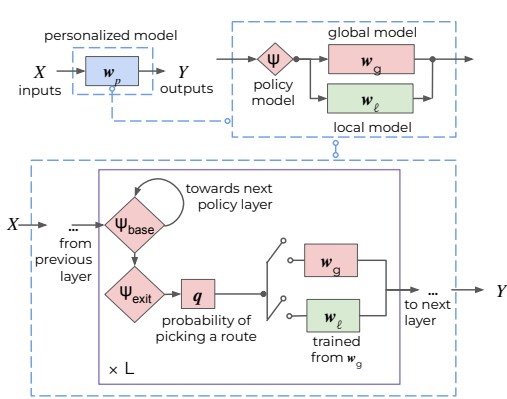

Figure 1: Illustration of the dynamic personalized model design proposed by *Flow*.

**Routing Module.** The routing module is a model with fully connected layers, shown as $\psi_{\text{base}}$ in Figure 1. After each layer $\psi_{\text{base}}$, the model has early exits denoted as $\psi_{\text{exit}}$ which outputs a probability of choosing the layer in the global model $w_g$ or the local model $w_\ell$. This probability at layer $j \in [L]$ is computed as,

$$\mathbf{q}^{(j)} = [\mathbf{q}_0^{(j)}, \mathbf{q}_1^{(j)}] = \text{softmax}(\psi_{\text{exit}}^{(j)}(\psi_{\text{base}}^{(j)}(\hat{x}))) \in [0,1]^2 \text{ where } \begin{cases} \hat{x} \leftarrow x_m & \text{if } j = 0; \\ \hat{x} \leftarrow \psi_{\text{base}}^{(j-1)}(\cdot) & \text{otherwise,} \end{cases} \tag{2}$$

where $\mathbf{q}_0^{(j)}$ and $\mathbf{q}_1^{(j)}$ are the probability of picking the global parameter $w_{g,m}^{(j)}$ and the local parameter $w_{\ell,m}^{(j)}$ respectively.

In order to train the routing parameters, we compute the training loss based on the output of the personalized model $w_{p,m}^{(r)}$, which averages the global model's output and local model's output weighted by $\mathbf{q}^{(j)}$, for each layer $j \in [L]$:

$$w_{p,m}^{(j)}(\hat{x}) \leftarrow \mathbf{q}_0^{(j)} \cdot w_{g,m}^{(j)}(\hat{x}) + \mathbf{q}_1^{(j)} \cdot w_{\ell,m}^{(j)}(\hat{x}) \text{ where } \begin{cases} \hat{x} \leftarrow x_m & \text{if } j = 0; \\ \hat{x} \leftarrow w_{p,m}^{(j-1)}(\cdot) & \text{otherwise.} \end{cases} \tag{3}$$

Let $f_m(\cdot)$ denote the loss function on client $m$. Using the above personalized model $w_{p,m}^{(r)}$, *Flow* updates the policy parameters as follows,

$$\psi_{g,m}^{(r+1)} \leftarrow \psi_{g,m}^{(r)} - \eta_\ell \nabla_{\psi_{g,m}^{(r)}} \left[ f_m(w_{p,m}^{(r)}; w_g^{(r)}, \zeta_{m,g}) - \frac{\gamma}{L} \sum_{j \in [L]} \log(\mathbf{q}_0^{(j)}) \right], \qquad (4)$$

where $-\frac{\gamma}{L} \sum_{j \in [L]} \log(\mathbf{q}_0^{(j)})$ is a regularization term that encourages the global model to learn on heterogeneous instances to improve the global model's generalizability.

**Global Parameters.**   Global parameters $w_g$ are trained alternatively with the routing parameters $\psi_g$. *Flow* updates the global parameters as follows,

$$w_{g,m}^{(r+1)} \leftarrow w_{g,m}^{(r)} - \eta_\ell \nabla_{w_{g,m}^{(r)}} f_m(w_{p,m}^{(r)}; \psi_{g,m}^{(r+1)}, \zeta_{m,g}) \qquad (5)$$

The goal of the alternative training is for the instances that are characterized better by the global data distribution to get diverted to the global model and the rest of the instances that are captured better by the local data distribution routed to the local model. This improves the stability of the global model compared to approaches like APFL or KNNPER which still use the global model trained on all the instances, regardless of the level of heterogeneity. The effectiveness of the alternative training is validated empirically and discussed in Appendix D.2.

**Soft versus Hard Policy.**   During training, we use soft policy where the probability in Equation 2 ranges over $[0, 1]$ to update the parameters in the global model and the routing module. But during inference, we use a discrete version of the policy round$(\mathbf{q}^{(j)}) \in \{0, 1\}^2$ for $\mathbf{q}^{(j)}$ in Eq. 3. The rationale behind this is twofold: (a) Using hard policy saves compute resources during inference by executing either the global or the local layer for an instance instead of both. (b) Our empirical evaluation shows negligible difference in performance of personalized models with soft and hard policies during the inference.

The dynamic nature of the personalized model in *Flow* introduces additional storage and computational overhead compared to the canonical method FEDAVG with Fine Tuning (called FEDAVGFT). However, compared to other state-of-the-art personalization methods such as DITTO and APFL, *Flow* requires similar or even less storage and computational overhead. Detailed analysis and comparison with baselines is in Appendix C.

## 4   Theoretical Analysis

In this section, we give convergence bounds for the global model $w_g$ and the personalized model $w_p$ of an arbitrary client $m$ in smooth non-convex cases. The bounds for strong and general convex cases are available in Appendix E, Sections E.3 and E.5. To derive the bounds, we adopt two commonly used assumptions in FL convergence analysis from AFO (Assumption 2, [11]) and SCAFFOLD (Assumption 1, [18]): (1) We assume local variance between a client's expected and true gradient is $\sigma_\ell$-bounded. (2) We also assume that the dissimilarity between aggregated gradients of local expected risk and the true global gradient is $(G, B)$-bounded, where both $G$ and $B$ are constants capturing the extent of gradient dissimilarities. A detailed description is in Appendix Section E.2.

Now we present the bounds on the norm of expected global (and local) risks on global (and personalized) models respectively, at $R$-th (last) round. The proofs are in Appendix Sections E.3 and E.5.

**Theorem 4.1** (Convergence of the Global Model). *If each client's objective function $f_m$ (and hence the global objective function $F$) satisfies $\beta$-smoothness, $\sigma_\ell$-bounded local gradient variance, $(G, B)$-dissimilarity assumptions, using the learning rate $\frac{1}{2\beta} \leq \eta_\ell \leq \frac{1}{2\sqrt{5}\beta BK^2\sqrt{R}}$ [for non-convex case] in Flow, then the following convergence holds:*

$$\frac{1}{R} \sum_{r=1}^{R} \mathbb{E} \left\| \nabla F(w_g^{(r)}) \right\|^2 \leq \frac{2}{\eta_\ell \mathbf{q}_0^2 R} \left[ \mathbb{E} \left[ F(w_g^{(1)}) \right] - \mathbb{E} \left[ F(w_g^{(R+1)}) \right] \right]$$

$$+ \frac{\sigma_\ell^2}{2\sqrt{5}BM\mathbf{q}_0^2 K\sqrt{R}} + \frac{2\mathbf{q}_1^2 G^2}{\mathbf{q}_0^2 B^2 R} + \frac{\sigma_\ell^2}{B^2 KR}.$$

**Discussion.** Here, $\mathbf{q}_0^2$ (and $\mathbf{q}_1^2$) are the probability of picking the global (and local) weights averaged over all instances sampled from the global data distribution. Given a fixed $\mathbf{q}_0^2$ and $\mathbf{q}_1^2$, a larger $G$ increases the bound, indicating slower convergence of the global model due to higher heterogeneity. We made two additional observations on the convergence of *Flow* depending on $\mathbf{q}_0^2$ in the bound.

One the one hand, when $\mathbf{q}_0^2 \to 1$, *Flow* matches the linear (strong convex) and sub-linear (general convex and non-convex) convergence rates of FEDAVG [18]. Specifically, $\mathbf{q}_0^2 = 1$ results in convergence rate of $\mathcal{O}(1/R^2)$ (strong convex), $\mathcal{O}(1/R^{2/3})$ (general convex), and $\mathcal{O}(1/R)$ (non-convex). When $\mathbf{q}_0^2 = 1$, which also indicates $\mathbf{q}_1^2 = 0$, the local parameters do not influence global model updates since all instances on a client will be routed to the global model and their gradients solely depend on the global model parameters. *Flow* degrades to FEDAVG in this case in terms of the convergence of the global model.

On the other hand, when $\mathbf{q}_0^2 \to 0$ indicating all instances are likely to be routed to the local model, the bound goes to infinity. It is because the global model parameters won't get updated and thus won't be able to converge. This observation validates the necessity of the regularization term in Eq. 4 to encourage instances to pick global model parameters.

Now we present the convergence bounds of the personalized model for the $m$-th client. The bound uses a definition of gradient diversity (noted as $\delta_m$) to quantify the diversity of a client's gradient with respect to the global aggregated gradient, following the prior work [29]. Higher diversity implies higher heterogeneity of the client.

**Theorem 4.2** (Convergence of the Personalized Model)**.** *If each client's objective function $f_m$ satisfies $\beta$-smoothness, $\sigma_\ell$-bounded local gradient variance, $(G, B)$-dissimilarity assumptions, and using the learning rate $\eta_\ell \leq \frac{1}{K\sqrt{12\beta R}}$ [for non-convex case] in Flow, then the following convergence holds:*

$$\frac{1}{R}\sum_{r=1}^{R}\mathbb{E}||\nabla f_m(w_{p,m}^{(r,K)})||^2 \leq \frac{2}{R}\left(\mathbb{E}\left[f_m(w_{p,m}^{(1,K)})\right] - \mathbb{E}\left[f_m(w_{p,m}^{(R,K)})\right]\right)$$

$$+ \mathcal{O}\left(\frac{\beta^2}{R^2K^2}\left(\sigma_\ell^2 + \left(\delta_m^{\psi_g} + \frac{\delta^{\psi_g}}{M}\right)K\right)\left(G^2 + \frac{\mathbf{q}_{1,m}^2 G^2}{\mathbf{q}_{0,m}^2 R}\right)\right)$$

**Discussion.** The theorem implies two main properties of the personalized models in *Flow*. First, for all convex and non-convex cases, the convergence rate of the personalized model in *Flow* is affected by the routing policy through the ratio $\mathbf{q}_{1,m}^2 G^2/\mathbf{q}_{0,m}^2$. We know that a higher value of the gradient dissimilarity constant $G$, indicates higher heterogeneity between the aggregated and expected global model. The ratio of $\mathbf{q}_{1,m}^2/\mathbf{q}_{0,m}^2$ would be higher for a heterogeneous client, since the client would get a higher probability of picking the local route ($\mathbf{q}_{1,m}^2 \to 1$). The higher $G$ and $\mathbf{q}_{1,m}^2$ results in slower convergence. On the contrary, a homogeneous client would benefit from a low value of $\mathbf{q}_{1,m}^2/\mathbf{q}_{0,m}^2$, which would offset the high value of $G$. Hence a homogeneous client's personalized model would converge faster than the one of a heterogeneous client. Second, we observe that gradient diversity of the policy model, $\delta_m^{\psi_g}$, linearly affects the personalized model's convergence. Since the policy model is also globally aggregated, a heterogeneous client would have a high $\delta_m^{\psi_g}$ and need more epochs per round to converge.

## 5 Experiments and Results

We empirically evaluate the performance of *Flow* against various personalization approaches for five non-iid vision and language tasks in terms of clients' accuracy.

**Datasets, Tasks, and Models.** We have experiments on two language and three vision tasks. The first three datasets which are described below represent real-world heterogeneous data where each author or user is one client. (a) **Stackoverflow** [30] dataset is used for the next word prediction task, using a model with 1 LSTM and 2 subsequent fully connected layers. (b) **Shakespeare** [31] dataset is for the next character prediction task, using a 2 layer LSTM + 1 layer fully connected model. (c) **EMNIST** [32] dataset is for 62-class image classification, which uses 2 CNN layers followed by 2 fully-connected layers. The models for the above three datasets have been described in [11]. The next two datasets are federated and artificially heterogeneous versions of CIFAR10/100

datasets. (d-e) **CIFAR10/100** [33] datasets are for 10- and 100-class image classification tasks with ResNet18 [34] model. Both CIFAR10/100 have two heterogeneous versions each: 0.1-Dirichlet is *more* heterogeneous, and 0.6-Dirichlet is *less* heterogeneous. Details about the datasets and the hyperparameters are in Appendix B.

**Baselines and Metrics.** We compare *Flow* with the following baselines: the classic FL algorithms FEDAVG [35], state-of-the-art personalized FL approaches including FEDAVGFT (FedAvg + Finetuning) [36], PARTIALFED [17], APFL [6], FEDREP [8], LGFEDAVG [7], DITTO [13], HYPCLUSTER [12], and KNNPER [19], and the LOCAL baseline which trains a local model on each client's dataset without any collaboration. We use the adaptive version of PARTIALFED as it shows better performance compared to the alternatives in [17]. We evaluate *Flow* and these baselines in terms of **generalized accuracy** and **personalized accuracy**, which correspond to the accuracy of the global model and the personalized model on clients' test data split.

We use Flower [37] library to implement *Flow* and all its baselines. We use an NVidia 2080ti GPU to run all the experiments with 3 runs for each. The random seeds used are 0, 44, and 56. We do not observe significant difference in results using other random seeds (see results in Appendix D.4).

## 5.1 Performance Comparison

**Generalized and Personalized Accuracy.** The performance of *Flow* and its baselines are reported in Table 2 for four datasets. Note that the LOCAL baseline has only personalized accuracy as it doesn't create a global model collaboratively. The FEDAVG has only generalized accuracy as it doesn't personalize the global model for each client. Since PARTIALFED is a stateful approach, we are unable to run it on the cross-device datasets (Stackoverflow, Shakespeare, EMNIST). Results for the rest of the datasets and their variance across 3 different runs are reported in Appendix D.

Table 2: Generalized ($Acc_g$) and Personalized ($Acc_p$) accuracy (the higher, the better) for *Flow* and baselines.

| Datasets | Stackoverflow | | Shakespeare | | CIFAR10 (0.1) | | CIFAR10 (0.6) | |
|---|---|---|---|---|---|---|---|---|
| Baselines | $Acc_g$ | $Acc_p$ | $Acc_g$ | $Acc_p$ | $Acc_g$ | $Acc_p$ | $Acc_g$ | $Acc_p$ |
| LOCAL | - | 15.93% | - | 18.70% | - | 49.78% | - | 62.74% |
| FEDAVG | 23.15% | - | 52.00% | - | 60.98% | - | 67.50% | - |
| FEDAVGFT | 23.83% | 24.41% | 52.12% | 53.68% | 61.23% | 73.03% | 68.19% | 72.21% |
| KNNPER | 23.16% | 24.49% | 51.87% | 53.10% | 59.62% | 75.14% | 69.22% | 70.14% |
| PARTIALFED | - | - | - | - | 62.57% | 73.20% | 66.93% | 70.38% |
| APFL | 22.96% | 25.70% | 52.38% | 53.64% | 62.87% | 72.86% | 69.53% | 72.53% |
| DITTO | 22.59% | 24.36% | 52.44% | 53.95% | 62.06% | 72.06% | 68.12% | 70.31% |
| FEDREP | 18.92% | 21.04% | 46.71% | 50.09% | 64.85% | 68.62% | 69.77% | 63.61% |
| LGFEDAVG | 22.61% | 24.03% | 51.08% | 51.43% | 56.63% | 73.19% | 67.48% | 68.94% |
| HYPCLUSTER | 23.75% | 22.43% | 51.92% | 52.74% | 63.64% | 71.55% | 65.44% | 72.40% |
| *Flow* (Ours) | **26.64%** | **29.49%** | **55.90%** | **56.20%** | **66.26%** | **76.47%** | **70.88%** | **77.11%** |

Overall, *Flow* achieves 1.11-3.46% higher generalized accuracy and 1.33-4.58% higher personalized accuracy over the best performing baseline. In particular, *Flow* outperforms KN-NPER, another per-instance per-client personalization approach, by 1.66-6.64% and 1.33-6.97% in generalized and personalized accuracy metrics respectively. KN-NPER allows each instance to personalize the prediction of the global model based on its k-nearest neighbors at inference time. However, the global model is trained through the classic FL method FEDAVG, which is agnostic to the heterogeneity of data instances. In contrast, *Flow* trains the global model differently via a parameterized dynamic routing module, which learns to put emphasis on data instances that are more aligned with the global data distribution to improve the performance of the global model.

*Flow* also outperforms the per-client personalization approaches including APFL, DITTO, HYPCLUSTER, and FEDAVGFT by 1.35-3.46 % (generalized accuracy) and 2.25-4.58% (personalized accuracy).

Table 3: % of clients for which $Acc_p > Acc_g$ (the higher, the better).

| | Stackoverflow | Shakespeare | CIFAR10 (0.1) | CIFAR10 (0.6) |
|---|---|---|---|---|
| FEDAVGFT | 79.26% | 79.00% | 97.18% | 99.33% |
| KNNPER | 82.73% | 68.87% | 90.00% | 90.00% |
| PARTIALFED | - | - | 88.30% | 84.80% |
| APFL | 69.66% | 79.22% | 87.48% | 90.63% |
| DITTO | 74.59% | 73.74% | 90.52% | 89.61% |
| FEDREP | 91.53% | 79.78% | 92.30% | 84.64% |
| LGFEDAVG | 83.47% | 88.43% | 88.41% | 89.59% |
| HYPCLUSTER | 80.46% | 74.84% | 95.11% | 98.18% |
| *Flow* (Ours) | **92.74%** | **89.77%** | **98.33%** | **99.62%** |

We see improvements in generalized accuracy of *Flow* because of the fact that the global model in *Flow* is trained based on the instances which align more with the global distribution. We see improvements in personalized accuracy due to the limitation of the per-client approaches where some instances being correctly classified by the global model are incorrect on the personalized model. We next give insights on why *Flow* achieves better performance in personalized and generalized accuracy compared to these personalized baselines.

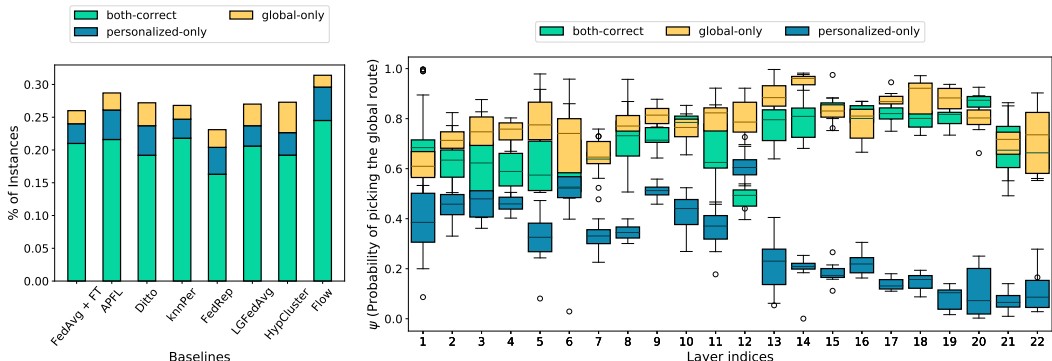

Figure 2: $w_g$ and $w_p$ accuracies for Stackoverflow.

Figure 3: Behavior of the routing policy from $\psi_g$ for all instances at each layer for Stackoverflow.

**Percentage of Clients Benefiting from Personalization.** The goal of personalization is to achieve higher prediction accuracy in each client by creating a per-client personalized model from the global model. We can thus measure the effectiveness of personalization by computing the percentage of clients for which the personalized model achieves higher task accuracy than the global model. The higher the percentage is, the better (or more effective) the personalization approach is.

Table 3 reports the results. We observed that *Flow* achieves the highest percentage of clients who benefit from personalization compared to all personalization baselines, echoing the better personalized accuracy from *Flow*. The percentage of clients who prefer personalized models can be as low as 68.87% (KNNPER on Shakespeare), which means personalization hurts up to 31% of clients' accuracy, as mentioned in the introduction. As a contrast, *Flow* improves the percentage of clients benefiting from personalization to 89.77%-99.62% because each instance, in a client, has a choice between the global model parameters and the local model parameters and can choose the one that better fits it. Note that the comparison is in favor of the baselines since *Flow* also achieves better generalized accuracy, which makes it even harder for personalized models to further improve prediction accuracy.

**Breakdown of Correctly Classified Instances.** Figure 2 further shows the breakdown of the percentage of instances that are (a) correctly classified by the global model but not the personalized model (noted as **global-only**, colored in yellow 🟡), (b) correctly classified by the personalized model but not the local model (noted as **personalized-only**, colored in blue 🔵), and (c) correctly classified by both models (noted as **both-correct**, colored in green 🟢 in *Flow* and baselines on Stackoverflow. The percentage of instances in y-axis is averaged over the test splits of all clients.

Overall, *Flow* increases the **both-correct** bars compared to all the baselines, which are the instances that contribute to the generalized performance of the global model. This explains the better generalized accuracy of *Flow*. *Flow* also increases the **personalized-only** bars and decreases the **global-only** bars, which correspond to the heterogeneous instances that prefer personalized models instead of the global model. This further explains the better personalized accuracy of *Flow*. Notably, for Stackoverflow dataset, existing personalization approaches still result in up to 4.74-7.93% of instances incorrectly classified by the personalized model but correctly by the global model. *Flow* reduces it to 1.12-2.42%. Similarly for the CIFAR10 (0.6) dataset, as mentioned in the introduction, we notice up to 11.4% of instances falling under the global-only category, which *Flow* reduces to 2.55%. It echoes the aforementioned effectiveness of personalization in *Flow*. The instance-wise accuracy breakdown for the rest of the datasets is detailed in Appendix D, Figure 10.

**Analysis of Routing Decisions.** We further analyze the behavior of routing decisions for instances of a client that fall in the above three cases, **global-only**, **personalized-only**, and **both-correct**. Figure 3 shows the per-layer routing policies of the dynamic personalized model from *Flow* on Stackoverflow. For instances that fall into each category, we average the policy value from Eq. 2 and report the statistics on the probability of picking the global parameters for each layer. The statistics for the rest of the datasets are detailed in Appendix D.

For instances that are correctly classified by $w_g$ but not by $w_p$ (**global-only**), we see a clear trend of the routing parameters getting more confident about picking the global parameters. As a contrast,

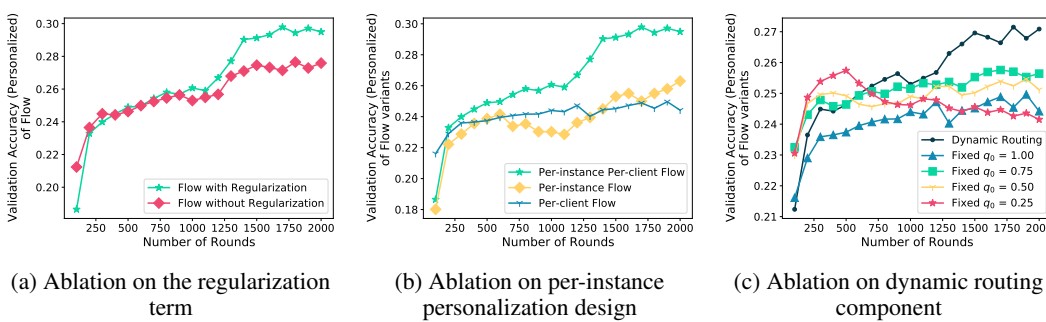

Figure 4: Ablation studies on Stackoverflow dataset.

for instances that are correctly classified by $w_p$ but not by $w_g$ (**personalized-only**), we see the trend of routing policy being more confident in picking the local parameters. For instances that can be correctly classified by both models (**both-correct**), the routing policy still prefers the global parameters over local parameters. This is due to the regularization term in Eq. 4, which encourages instances to pick the global model over the local model in order to improve the generalizability of the global model. Our ablation study in the next section demonstrates the importance of regularization.

## 5.2 Ablation Studies

Here we highlight some results of three ablation studies on regularization, per-instance personalization, dynamic routing, and hard policies during inference. More results are in Appendix D.

**Regularization.** The regularization term in Eq. 4 promotes the global model layers whenever possible. It helps boost the generalized performance of the global model, which in turn also produces better personalized models. We use the results on the Stackoverflow dataset in Figure 4a to illustrate the importance of regularization. At the end of the training, we get $26.64\% \pm 0.23\%$ generalized accuracy with regularization, compared to $24.16\% \pm 0.34\%$ without regularization, and $29.49\% \pm 0.28\%$ personalized accuracy with regularization, compared to $27.59\% \pm 0.36\%$ without regularization. The importance of regularization in the policy update rule is also highlighted in Theorem 4.1, which states that only picking local route does not lead to global model convergence; the regularization term can encourage the global model to converge faster.

**Per-instance Personalization.** The dynamic personalized model in *Flow* allows each instance to choose between the local model and global model layers. To verify this per-instance personalization design, we create two *variants* of *Flow*, named "Per-Instance" *Flow* (PI-FLOW) where both paths of a dynamic model are global models, and "Per-Client" *Flow* (PC-FLOW) which is simply FEDAVGFT. Compared to the personalized accuracy of per instance and per client *Flow* ($29.49\% \pm 0.28\%$), PI-FLOW and PC-FLOW achieve $26.31\% \pm 0.19\%$ and $24.41\% \pm 0.26\%$ respectively on Stackoverflow dataset, as shown in Figure 4b. The results demonstrate the effectiveness of the per-instance personalization design in *Flow*.

**Dynamic Routing.** This ablation study aims to learn whether dynamically interpolating global and local routes has any advantages over fixing the routing policy throughout the training. In Figure 4c, we compare the validation accuracy curves of dynamic routing in *Flow* and dynamic routing with instance-agnostic static routing during the training phase. We observe that (a) For the case of fixed policy of $\mathbf{q}_0 = 0.25$, the validation accuracy has bad performance due to the fixed policy only choosing the local route. This is due to using hard policy during inference, and (b) The cases of fixed policy $\mathbf{q}_0 \in [0.50, 0.75, 1.00]$ will only pick the global route during inference, which are also outperformed by the dynamic routing variant. With dynamic routing, the choice between local and global parameters depends on each instance during inference.

**Soft versus Hard Decisions during Inference.** We did not use soft decisions during inference since it only negligibly improves the accuracy of *Flow*. The test accuracies after personalization for *Flow* on Stackoverflow with hard decisions are $29.49\% \pm 0.28\%$, while with soft decisions, we observed $29.57\% \pm 0.22\%$. The rest of the datasets show a similar trend (see Appendix D, Table 8).

## 6 Conclusion

This paper proposed *Flow*, a per-instance and per-client personalization method to address the statistical heterogeneity issue in Federated Learning. *Flow* is motivated by the observation that the personalized models from existing personalized approaches achieve lower accuracy in a significant portion of clients compared to the global model. To overcome this limitation, *Flow* creates dynamic personalized models with a routing policy that allow instances on each client to choose between global and local parameters to improve clients' accuracy. We derived error bounds for global and personalized models of *Flow*, showing how the routing policy affects the rate of convergence. The theoretical analysis validates our empirical observations related to clients preferring either a global or a local route based on the heterogeneity of individual instances. Extensive evaluation on both vision and language-based prediction tasks demonstrates the effectiveness of *Flow* in improving both the generalized and personalized accuracy.

## Acknowledgments and Disclosure of Funding

This material is based upon work supported by the National Science Foundation under Grant No. **2312396**, **2220211**, and **2224054**. Any opinions, findings, and conclusions or recommendations expressed in this material are those of the author(s) and do not necessarily reflect the views of the National Science Foundation. The work is also generously supported by Adobe Gift Fund.

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

## A    Limitations, Future Work, and Broader Impact

Learning on naturally heterogeneous datasets can be challenging, as the true data distributions of individual clients are unknown, making it difficult to correlate the divergence between client data distribution and the global data distribution with routing policy decisions. In our approach, we estimate the distribution divergence by measuring the difference between inference losses on global and local models, which helps us reason about routing probabilities for global and local routes. To further improve our understanding of the model performance, we plan to propose a metric that quantifies the difference in performance when a particular dataset is included versus excluded.

*Flow* has shown the promise of per-instance personalization in improving clients' accuracy. This approach also holds the potential of preserving privacy by protecting against gradient leakage [38–40] and membership inference [41, 42] attacks that are easier to carry out in vanilla FL. Studying the relationship between personalization and privacy, and comparing our approach to traditional methods like Differential Privacy (DP) [43, 44] can reveal properties of personalization that go beyond improved accuracy.

## B    Datasets and Hyperparameters

**Stackoverflow**    The Stackoverflow dataset [30] is comprised of separate clients designated for training, validation, and testing. The dataset contains a total of 342,477 train clients, whose combined sample count equals 135,818,730. Similarly, the dataset contains 38,758 validation and 204,088 test clients, whose combined sample counts equal 16,491,230 and 16,586,035, respectively. This dataset is naturally heterogeneous [45] since each user of Stackoverflow represents a client, with their posts forming the dataset for that client. The heterogeneity of the dataset arises from the fact that users have different writing styles, meaning the clients' datasets are not i.i.d., and the total number of posts from each user varies, leading to different dataset sizes per client.

We have trained *Flow* and its baselines on the Stackoverflow dataset for 2000 rounds. The one layer LSTM we have used has 96 as embedding dimension, 670 as hidden state size, and 20 as the maximum sequence length [11]. The batch size used for each client on each baseline is 16. The vocabulary of this language task is limited to 10k most frequent tokens. The default learning rate used is 0.1. The number of clients per round is set to 10, as is the common practice in [13, 46, 12, 10, 47]. For client-side training, the default epoch count is 3 for all the algorithms.

For KNNPER, we used 5 nearby neighbors, and the mixture parameter is $\lambda = 0.5$. For APFL, mixture hyperparameter $\alpha$ is set to 0.25. DITTO has regularization hyperparameter $\lambda = 0.1$. There are 2 clusters by default for HYPCLUSTER. *Flow* and its variants were tested on the following choices of regularizing hyperparameters $\gamma \in \{$1e-1, 1e-2, 1e-3, 1e-4$\}$, where 1e-3 gave the best personalized accuracy.

**Shakespeare**    The Shakespeare dataset [31] consists of 715 distinct clients, each of which has its own training, validation, and test datasets. The combined training datasets of all clients contain a total of 12,854 instances, while the combined validation and test datasets contain 3,214 and 2,356 instances, respectively. The Shakespeare dataset is considered heterogeneous due to the fact that each client is a play written by William Shakespeare, and these plays have varying settings and characters.

All the baselines and *Flow* variants have been run for 1500 rounds, with 10 clients per round. The 2 layer LSTM used [11] has 8 as embedding size, vocabulary size of 90 most frequently used characters, and 256 as hidden size. The default epoch count is 5 for each client, for each algorithm. The batch size is 4 since bigger batch sizes resulted in the divergence of the global model across all the different runs. The default learning rate is 0.1.

Since each client has a sample count under 20, we have used 3 as the nearest neighbor sample count for KNNPER. $\lambda$ and $\alpha$, the mixture parameters, for KNNPER and APFL respectively, are set to 0.45 and 0.3. The regularization parameter $\lambda$ for DITTO is set to 0.1. For *Flow*, the learning rate is set to 0.11 and the regularization parameter is picked from $\gamma \in \{$1e-1, 1e-2, **1e-3**, 1e-4$\}$ similar to Stackoverflow.

**EMNIST**    The EMNIST dataset [32] comprises 3400 distinct clients, each of which has its own training, validation, and test datasets. The combined total number of instances in the train datasets of all clients is 671,585, whereas the validation and test datasets of all clients combined contain 77,483 instances each. The heterogeneity of EMNIST clients is due to the individual writing styles of each client, with each client representing a single person. This is discussed in Appendix C.2 of [11].

The default round count for all the baselines and *Flow* variants is 1500, with 10 clients participating per round. Similar to AFO [11], we have used a shallow convolution neural network with 2 convolution layers. Each client uses 3 local epochs for on-device training. The default batch size is 20, and the default learning rate is 0.01.

For LOCAL only training, we have used 10 epochs per client with a learning rate of 0.05. The nearest sample count for KNNPER is 10 and the mixture parameter is $\lambda = 0.4$. For APFL, we have the default mixture parameter as $\alpha = 0.25$. DITTO has regularization hyperparameter as $\lambda = 0.1$. There are 2 clusters for the clustering

algorithm HYPCLUSTER. And for *Flow*, along with its variants, we have picked $\gamma \in \{$1e-1, **1e-2**, 1e-3, 1e-4$\}$ as the regularizing hyperparameter.

**CIFAR10**    The CIFAR10 dataset is derived from the centralized version of the CIFAR10 dataset [33], which comprises 50,000 images. The federated CIFAR10 dataset consists of 500 unique clients, each of which has 100 training samples and 20 testing samples. The training and testing samples for each client are determined according to the Dirichlet distribution [11]. The heterogeneity of a client is determined by the Dirichlet distribution parameter $\alpha \in [0, 1]$, where a client is more heterogeneous than $\alpha \to 0$. In this context, heterogeneity refers to the dissimilarity of the dataset instances sampled from a distribution. We conducted experiments on clients with $\alpha$ values of 0.1 and 0.6.

We ran all the experiments for 4000 rounds for the CIFAR10 dataset. ResNet18 [34] is used for all the algorithms. The default batch size is 20 and the default learning rate is 0.05. Each client individually trains their local versions of the global model for 3 epochs.

For LOCAL only training, 20 epochs per client were used. The learning rate was 0.1 for the same. The nearest sample count and the mixture hyperparameter for KNNPER are set to 5 and 0.5. PARTIALFED learning rate is set to 0.11, with the local epoch count is 5. APFL has mixture hyperparameter set as $\alpha = 0.2$. And DITTO has a regularization hyperparameter set as $\lambda = 0.01$. *Flow* and its variants have their regularization hyperparameter as $\gamma \in \{$1e-1, 1e-2, **1e-3**, 1e-4$\}$.

**CIFAR100**    Like CIFAR10, the CIFAR100 dataset [48] is derived from the CIFAR100 dataset [33] consisting of 50,000 images. The number of clients and the count of training and testing images are identical to those of CIFAR10. Similarly, we also conducted experiments with the Dirichlet parameter set to $\alpha = 0.1$ and $\alpha = 0.6$.

Similar to CIFAR10, we have a 4000 round count for all the algorithms ran on the CIFAR100 dataset. We have again used ResNet18 [34]. The default local epoch count is 3, and the default learning rate is 0.05. We have used 20 batch size for all the algorithms. For each round, 10 clients participate as is the norm stated in the Stackoverflow dataset description.

LOCAL only training has 20 epochs per client, and 0.1 learning rate. 5 nearest samples are used for KNNPER, while the mixture parameter $\lambda$ is set to 0.4. PARTIALFED, just like in CIFAR10, has 0.11 learning rate and 5 local epochs per client. APFL has 0.25 as mixture parameter $\alpha$. DITTO has 1e-2 as regularization parameter $\lambda$. For both CIFAR10 and CIFAR100, we have 2 as the default cluster count for HYPCLUSTER. *Flow* and its variants get $\{$1e-1, 1e-2, **1e-3**, 1e-4$\}$ as the regularization hyperparameter $\gamma$.

# C    Computation, Communication, and Storage

While *Flow* introduces additional storage and computational overhead compared to the canonical method FEDAVG with Fine Tuning (called FEDAVGFT), compared to other state-of-the-art personalization methods such as DITTO and APFL, *Flow* requires similar or even less storage and computational overhead.

To illustrate, Table 4 compares the storage, computational overhead, and communication cost for personalized models from *Flow* and baselines using the CNN for EMNIST, and RNN for Stackoverflow.

Referring to Table 4, for *Flow*, $\psi_g$ is the policy module. For our experiments on *Flow*, the policy module has 1.23% for Stackoverflow RNN, 8.39% for Shakespeare RNN, 27.86% for EMNIST CNN, 35.51% for CIFAR10/100 parameters of $w_g$. For EMNIST CNN, for one epoch, the dynamic routing takes $0.3M$ FLOPs, while the rest of the model computations take $2.9M$ FLOPs. Hence the overhead is 10.34%. For Stackoverflow RNN one epoch, the dynamic routing takes $0.89M$ FLOPs, while the rest of the model computations take $12.34M$ FLOPs, making the overhead 7.21%.

Below, we highlight the computational and storage analysis on Stackoverflow dataset for the best performing methods, APFL, and HYPCLUSTER. The computational overhead is calculated for one epoch of training, which is then multiplied with however many epochs the baseline needs for convergence. APFL needs to train two separate local models for more numbers of epochs than *Flow*, hence the higher computational overhead. In comparison with HYPCLUSTER, *Flow* introduces 7.23% overhead. With respect to both APFL and HYPCLUSTER, *Flow* introduces only 1.23% storage overhead.

Moreover, from Table 4, we can see that compared to some other state-of-the-art personalization techniques (e.g., FEDREP, APFL), *Flow* requires similar or even less computations, even with the dynamic routing operations.

Table 4: Computational and Communication costs, and Storage of *Flow* and its baselines.
$A$ = Local Storage of Personalized Model (unit: parameter count) for general case,
$B$ = Local Storage of Personalized Model (unit: parameter count) for Stackoverflow RNN case,
$C$ = Computational Overhead of Personalized Model of the RNN used for Stackoverflow (unit: FLOPs for training),
$D$ = Communication Cost (unit: parameter count) for general case, and
$E$ = Communication Cost (unit: parameter count) for Stackoverflow RNN case.

| Baselines | $A$ | $B$ | $C$ | $D$ | $E$ |
|---|---|---|---|---|---|
| FEDAVGFT | $|w_g|$ | $72.38M$ | Not Applicable | $|w_g|$ | $72.38M$ |
| KNNPER | $|w_g|$ + (#instances * |intermediate representation| ) | $72.42M$ | $12.46M$ | $|w_g|$ | $72.38M$ |
| PARTIALFED | $|w_g|$ + 2∗# layers of $w_g$ | $72.38M$ | $36.9M$ | $|w_g|$ | $72.38M$ |
| APFL | $3|w_g|$ | $217.14M$ | $73.8M$ | $|w_g|$ | $72.38M$ |
| DITTO | $2|w_g|$ | $144.76M$ | $36.9M$ | $|w_g|$ | $72.38M$ |
| FEDREP | $|w_g|$ | $72.38M$ | $51M$ | $|w_g|$ (base)| | $70.98M$ |
| LGFEDAVG | $|w_g|$ | $72.38M$ | $10.5M$ | $|w_g|$ (head)| | $1.39M$ |
| HYPCLUSTER | $2|w_g|$ | $144.76M$ | $36.9M$ | $|w_g|$ | $72.38M$ |
| FLOW (ours) | $2|w_g| + |\psi_g|$ | $145.651M$ | $39.57M$ | $|w_g| + |\psi_g|$ | $73.271M$ |

# D  Additional Results

## D.1  Generalized and Personalized Accuracy

Generalized (Personalized) accuracy is calculated based on the global (personalized) model, where each participating client's test dataset is used to compute accuracy of the global (personalized) model.

Generalized accuracy is formulated as

$$Acc_g = \frac{1}{M} \sum_{m \in [M]} \frac{\sum_{(x,y) \in \mathcal{S}_m^{\text{test}}} \mathbb{1}\{y = w_g(x)\}}{\mathcal{S}_m^{\text{test}}}. \tag{6}$$

Personalized accuracy is formulated as

$$Acc_p = \frac{1}{M} \sum_{m \in [M]} \frac{\sum_{(x,y) \in \mathcal{S}_m^{\text{test}}} \mathbb{1}\{y = w_{p,m}(x)\}}{\mathcal{S}_m^{\text{test}}}. \tag{7}$$

We have reported Generalized (Personalized) Accuracy $Acc_g$ ($Acc_p$) of *Flow*, averaged across 1000 clients in Table 5, for all the datasets. Similarly, variance of accuracies across 3 different runs (based on seeds 0, 44, 56) is reported in Table 6. The learning curves for both Generalized and Personalized accuracies for all datasets for *Flow* and its baselines are in Figures 5 and 6.

Table 5: Generalized ($Acc_g$) and Personalized ($Acc_p$) accuracy (the higher, the better) for *Flow* and baselines. Variance across different runs is reported in Appendix D, Table 6.

| Datasets | Stackoverflow | | Shakespeare | | EMNIST | | CIFAR10 (0.1) | | CIFAR100 (0.1) | | CIFAR10 (0.6) | | CIFAR100 (0.6) | |
|---|---|---|---|---|---|---|---|---|---|---|---|---|---|---|
| Baselines | $Acc_g$ | $Acc_p$ | $Acc_g$ | $Acc_p$ | $Acc_g$ | $Acc_p$ | $Acc_g$ | $Acc_p$ | $Acc_g$ | $Acc_p$ | $Acc_g$ | $Acc_p$ | $Acc_g$ | $Acc_p$ |
| LOCAL | - | 15.93% | - | 18.70% | - | 28.18% | - | 49.78% | - | 36.19% | - | 62.74% | - | 21.31% |
| FEDAVG | 23.15% | - | 52.00% | - | 85.10% | - | 60.98% | - | 28.11% | - | 67.50% | - | 30.33% | - |
| FEDAVGFT | 23.83% | 24.41% | 52.12% | 53.68% | 89.57% | 90.14% | 61.23% | 73.03% | 29.60% | 31.02% | 68.19% | 72.21% | 31.15% | 37.24% |
| KNNPER | 23.16% | 24.49% | 51.87% | 53.10% | 85.20% | 88.28% | 59.62% | 75.14% | 28.08% | 33.62% | 69.22% | 70.14% | 30.66% | 34.39% |
| PARTIALFED | - | - | - | - | - | - | 62.57% | 73.20% | **34.79%** | 40.64% | 66.93% | 70.38% | 37.72% | **40.18%** |
| APFL | 22.96% | 25.70% | 52.38% | 53.64% | 88.40% | 89.44% | 62.87% | 72.86% | 31.05% | 32.56% | 69.53% | 72.53% | 36.37% | 36.74% |
| DITTO | 22.59% | 24.36% | 52.44% | 53.95% | 89.08% | 91.30% | 62.06% | 72.06% | 28.14% | 35.45% | 68.12% | 70.31% | 35.11% | 36.07% |
| FEDREP | 18.92% | 21.04% | 46.71% | 50.09% | 89.95% | 89.77% | 64.85% | 68.62% | 26.10% | 33.72% | 69.77% | 63.61% | 28.42% | 31.02% |
| LGFEDAVG | 22.61% | 24.03% | 51.08% | 51.43% | 87.43% | 91.70% | 56.63% | 73.19% | 31.65% | 39.63% | 67.48% | 68.94% | 35.01% | 33.90% |
| HYPCLUSTER | 23.75% | 22.43% | 51.92% | 52.74% | 89.47% | 90.49% | 63.64% | 71.55% | 31.57% | 33.04% | 65.44% | 72.40% | 34.76% | 36.22% |
| *Flow* (Ours) | **26.64%** | **29.49%** | **55.90%** | **56.20%** | **90.88%** | **94.18%** | **66.26%** | **76.47%** | 34.00% | **42.42%** | **70.88%** | **77.11%** | **39.70%** | 40.08% |

*Flow* sees an improvement of 1.11-3.46% in $Acc_g$ and 1.33-4.58% in $Acc_p$ over the best performing baseline. Besides the main observations listed in Section 5, we discuss results on the CIFAR100 dataset here. For CIFAR100 (0.6), *Flow* (40.08% ± 0.27%) matches the personalized accuracy of the highest performing baseline, PARTIALFED (40.18% ± 0.19%), while achieving 1.98% point increase in generalized accuracy. And for CIFAR100 (0.1), *Flow* improves personalized accuracy by 1.78% points. For generalized accuracy, *Flow* (34.00% ± 0.32%) reaches close to the best performing baseline, PARTIALFED (34.79% ± 0.29%). The reason behind the on-par performance of *Flow* with PARTIALFED can be attributed to the statefulness of PARTIALFED.

Table 6: Variance of generalized and personalized accuracies across 3 different runs (seeds = 0, 44, 56) for *Flow* and its baselines.

| Datasets | SO NWP | | Shakespeare | | EMNIST | | CIFAR10 (0.1) | | CIFAR100 (0.1) | | CIFAR10 (0.6) | | CIFAR100 (0.6) | |
|---|---|---|---|---|---|---|---|---|---|---|---|---|---|---|
| Baselines | $Acc_g$ | $Acc_p$ | $Acc_g$ | $Acc_p$ | $Acc_g$ | $Acc_p$ | $Acc_g$ | $Acc_p$ | $Acc_g$ | $Acc_p$ | $Acc_g$ | $Acc_p$ | $Acc_g$ | $Acc_p$ |
| LOCAL | - | 0.25% | - | 0.46% | - | 1.14% | - | 1.56% | - | 0.43% | - | 0.89% | - | 0.25% |
| FEDAVG | 0.07% | - | 0.39% | - | 1,32% | - | 1.12% | - | 0.31% | - | 0.82% | - | 0.15% | - |
| FEDAVGFT | 0.09% | 0.26% | 0.51% | 0.59% | 1.16% | 1.21% | 0.99% | 0.89% | 0.46% | 0.62% | 1.10% | 1.26% | 0.33% | 0.42% |
| KNNPER | 0.16% | 0.24% | 0.36% | 0.41% | 0.95% | 1.02% | 1.41% | 1,57% | 0.34% | 0.57% | 0.91% | 1.06% | 0.24% | 0.29% |
| PARTIALFED | - | - | - | - | - | - | 1.36% | 1.39% | 0.29% | 0.46% | 0.96% | 1.97% | 0.09% | 0.19% |
| APFL | 0.19% | 0.20% | 0.41% | 0.53% | 1.41% | 1.50% | 1.24% | 1.31% | 0.36% | 0.72% | 0.70% | 0.97% | 0.42% | 0.59% |
| DITTO | 0.12% | 0.15% | 0.49% | 0.56% | 1.12% | 1.22% | 1.35% | 1.41% | 0.43% | 0.69% | 0.84% | 0.87% | 0.28% | 0.34% |
| FEDREP | 0.15% | 0.29% | 0.50% | 0.65% | 0.89% | 0.94% | 0.95% | 1.02% | 0.59% | 0.79% | 0.96% | 1.14% | 0.14% | 0.10% |
| LGFEDAVG | 0.08% | 0.16% | 0.32% | 0.56% | 1.10% | 1.17% | 1.21% | 1.24% | 0.47% | 0.51% | 0.82% | 0.96% | 0.23% | 0.21% |
| HYPCLUSTER | 0.20% | 0.19% | 0.56% | 0.73% | 0.90% | 1.13% | 1.43% | 1.49% | 0.39% | 0.47% | 0.98% | 0.76% | 0.35% | 0.46% |
| *Flow* | 0.23% | 0.28% | 0.40% | 0.49% | 1.16% | 1.21% | 1.23% | 1.25% | 0.32% | 0.36% | 0.78% | 0.86% | 0.21% | 0.27% |

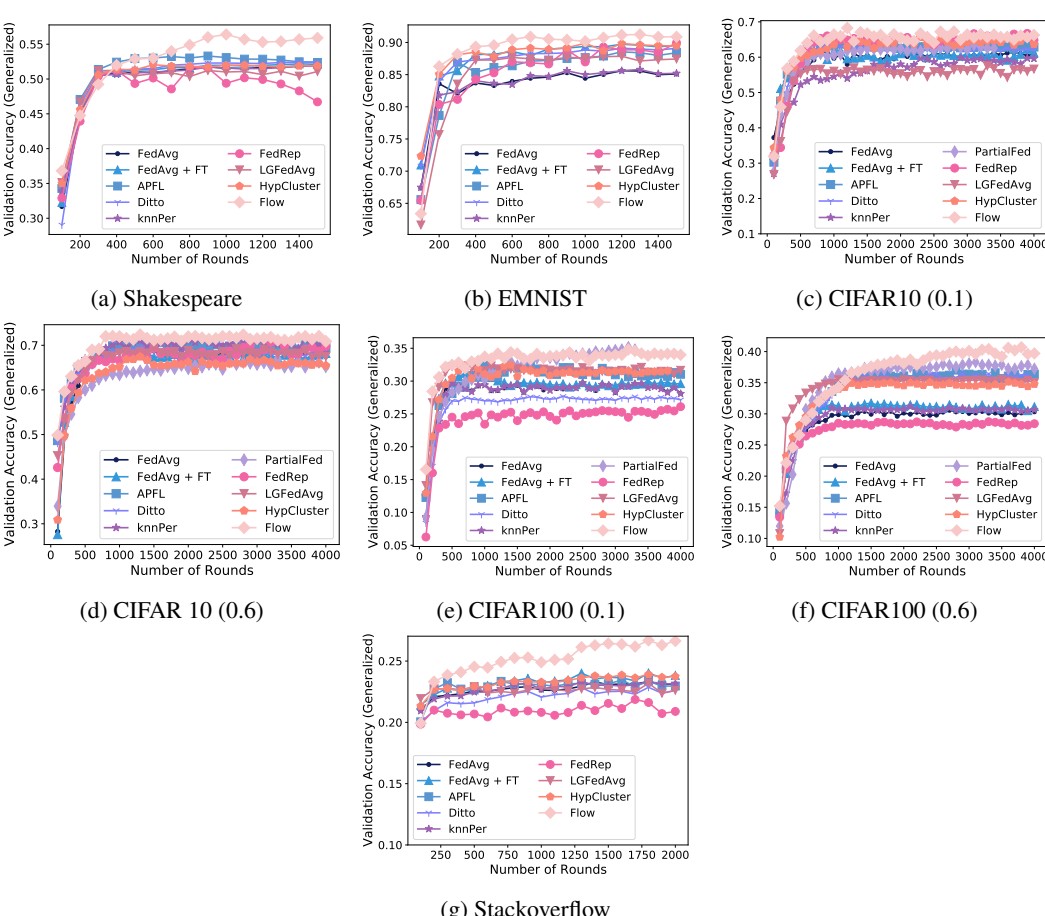

(a) Shakespeare     (b) EMNIST     (c) CIFAR10 (0.1)

(d) CIFAR 10 (0.6)     (e) CIFAR100 (0.1)     (f) CIFAR100 (0.6)

(g) Stackoverflow

Figure 5: Learning curves on Generalized Accuracy Metric of *Flow* and its baselines.

With the assumption of full device participation, PARTIALFED makes use of each client's previous state of the personalized model to further train its layer-wise model building policy. With *Flow*, both the assumptions of full device participation and statefulness of the personalized model are not necessary. Since the clients do not necessarily have to carry their personalized model states to the upcoming rounds, the personalized model recreated by *Flow* might be unable to compete against stateful approaches like PARTIALFED. Although because of the per-instance routing, *Flow* still manages to outperform PARTIALFED for the CIFAR10 (0.1/0.6) datasets, and gives comparable performance for the CIFAR100 (0.1/0.6) datasets.

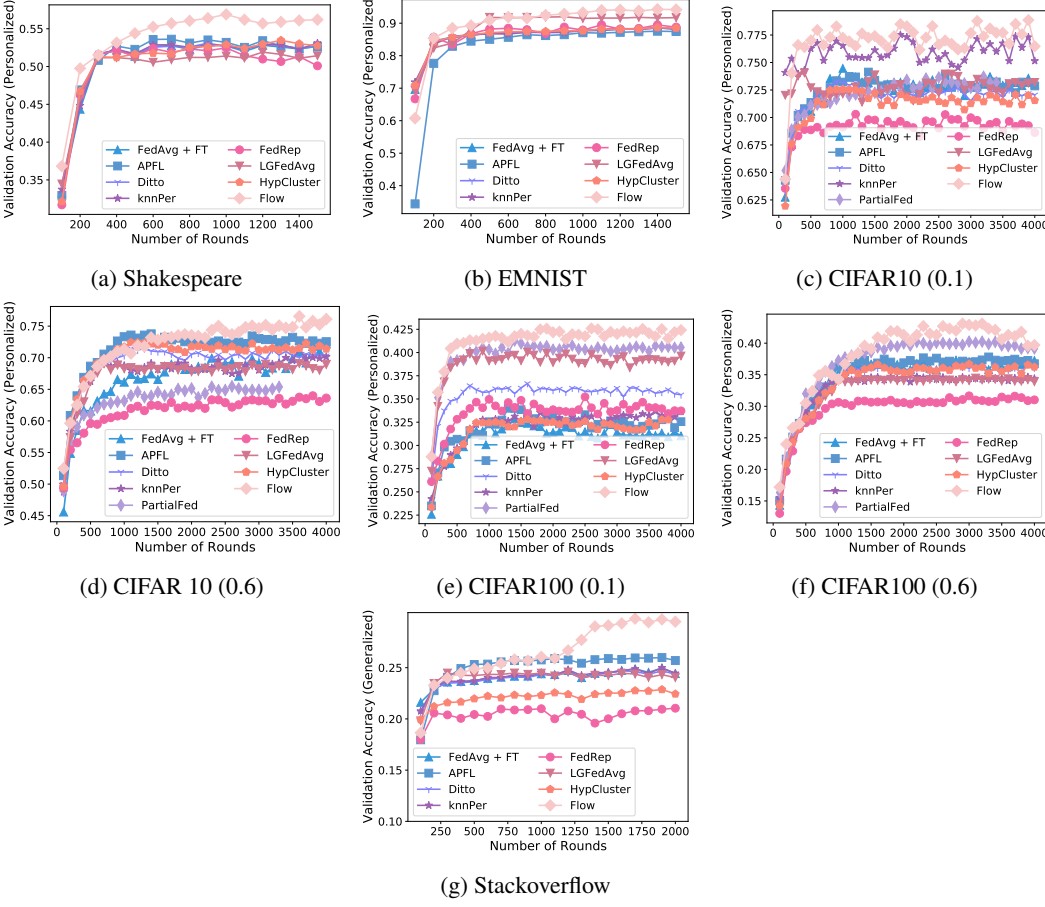

Figure 6: Learning curves on Personalized Accuracy Metric of *Flow* and its baselines.

## D.2 Effectiveness of the Alternative Training of the Global Model

We experimented with three modes of training for the global model: (a) Global model trained first, then the policy module, (b) Policy module trained first, then the global model, (c) Global model and Policy module trained alternatively. The results are shown in Figure 7. We see that the alternate training results in a more stable training compared to the other two modes of training. These results are in conformance with other works [49, 50] which have also used alternative training for policy and model weights training.

## D.3 Dataset Split

We experimented with the Stackoverflow Next Word Prediction task on initial rounds for the local and global training dataset splits. The plot is shown in Figure 8. We observe that for the dataset split size of 0.75:0.25 for local and global datasets respectively, the global model does not get sufficient samples to converge, resulting in worse personalized model performance since the personalized model is based on the global model. While a split of 0.25:0.75 for local and global datasets has closer performance to that of a 0.50:0.50 split, lesser data (and hence fewer iterations) to the local model leads to local model weights being similar to that of global model, diminishing the impact of personalization.

## D.4 More Seeds

We have re-run Stackoverflow experiment with seeds in {1, 2, 3, 4, 5 , 6, 7, 8, 9}, the results show no material difference. See Figure 9.

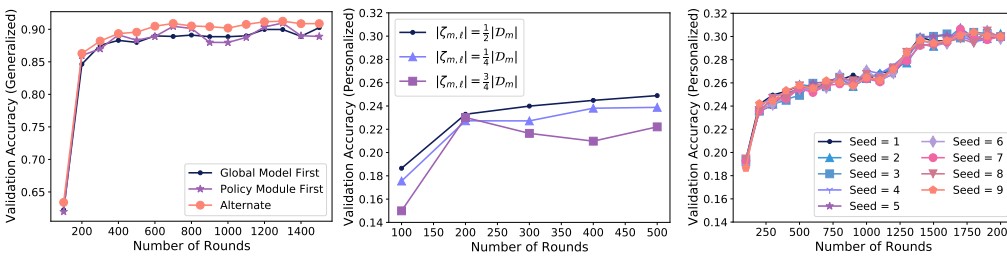

Figure 7: Results for different modes of training for EMNIST

Figure 8: Earlier stages of Stackoverflow NWP training with different local:global dataset splits

Figure 9: Stackoverflow NWP training with seed choices of [1,9]

## D.5 Percentage of Clients Benefiting from Personalization

In this section we discuss the effect of personalization, by comparing each client's performance on their individual personalized models with their performance on the global model. The evaluation, just as in section D.1, is done on the test datasets of all the clients. The goal with any personalization method is to make each client's personalized model more beneficial (for us, in terms of accuracy) compared to the global model. Hence we want $Acc_p > Acc_g$, to incentivize personalization for each client. As shown in Table 7, compared to the best performing baseline, *Flow* improves the utility of personalization by up to 3.31% points.

Table 7: % of clients for which $Acc_p > Acc_g$ (the higher, the better).

|  | Stackoverflow | EMNIST | Shakespeare | CIFAR10 (0.1) | CIFAR100 (0.1) | CIFAR10 (0.6) | CIFAR100 (0.6) |
|---|---|---|---|---|---|---|---|
| FEDAVGFT | 79.26% | 81.48% | 79.00% | 97.18% | 91.74% | 99.33% | 88.54% |
| KNNPER | 82.73% | 89.97% | 68.87% | 90.00% | 94.71% | 90.00% | 96.37% |
| PARTIALFED | - | - | - | 88.30% | 90.32% | 84.80% | 98.64% |
| APFL | 69.66% | 93.39% | 79.22% | 87.48% | 86.18% | 90.63% | 92.03% |
| DITTO | 74.59% | 79.26% | 73.74% | 90.52% | 91.45% | 89.61% | 97.45% |
| FEDREP | 91.53% | 82.20% | 79.78% | 92.30% | 78.81% | 84.64% | 99.54% |
| LGFEDAVG | 83.47% | 66.16% | 88.43% | 88.41% | 86.39% | 89.59% | 91.73% |
| HYPCLUSTER | 80.46% | 80.70% | 74.84% | 95.11% | 93.70% | 98.18% | 99.73% |
| *Flow* (Ours) | **92.74%** | **96.70%** | **89.77%** | **98.33%** | **97.29%** | **99.62%** | **99.75%** |

## D.6 Breakdown of Correctly Classified Instances

Here we show a detailed view of how instances (across all the clients) get classified correctly between global and personalized models for each of the baselines. For the plots in Figures 10, $y$-axis represent % of instances correctly classified by (a) Both the global and the personalized models (**both-correct**), (b) Only the global model (**global-only**), and (c) Only the personalized model (**personalized-only**). This % of instances metric is averaged across all clients, and is based on their test datasets. The goal here is to increase the % of instances for **both-correct** and **personalized-only**, and reduce the % of instances for **global-only**. We make the following observations for each of the datasets: Since *Flow* improves both the generalized and personalized accuracies, we see higher **both-correct** for Stackoverflow (by 2.75% points), Shakespeare (by 4.34% points), EMNIST (by 3.17% points), CIFAR10 (0.1) (by 5.24% points), CIFAR10 (0.6) (by 0.03% points), CIFAR100 (0.1) (by 0.63% points) and CIFAR100 (0.6) (by 2.78% points).

Due per-instance personalization, we see improvements in personalized accuracy, but those improvements are also included in the **both-correct** bars, so solely comparing **personalized-only** bar lengths is not a right comparison. Similarly, we see fewer instances in **global-only** bars due to the increase in instances which fall under **both-correct**.

## D.7 Analysis of Routing Decisions

Now we show probability value analysis of the routing policy for CIFAR10/100 datasets. Here we have fixed the client as the client which had the highest loss difference between its global and personalized models for *Flow*. This analysis was done during the inference stage, on the test dataset of the above-mentioned client. The box plots show statistics on the probability of picking the global route for all the instances. Echoing the observations made in Section 5, in Figure 11, we see a trend in increasing probability for the global parameters for the instances which are correctly classified by only the global model. In the contrary, for the instances which

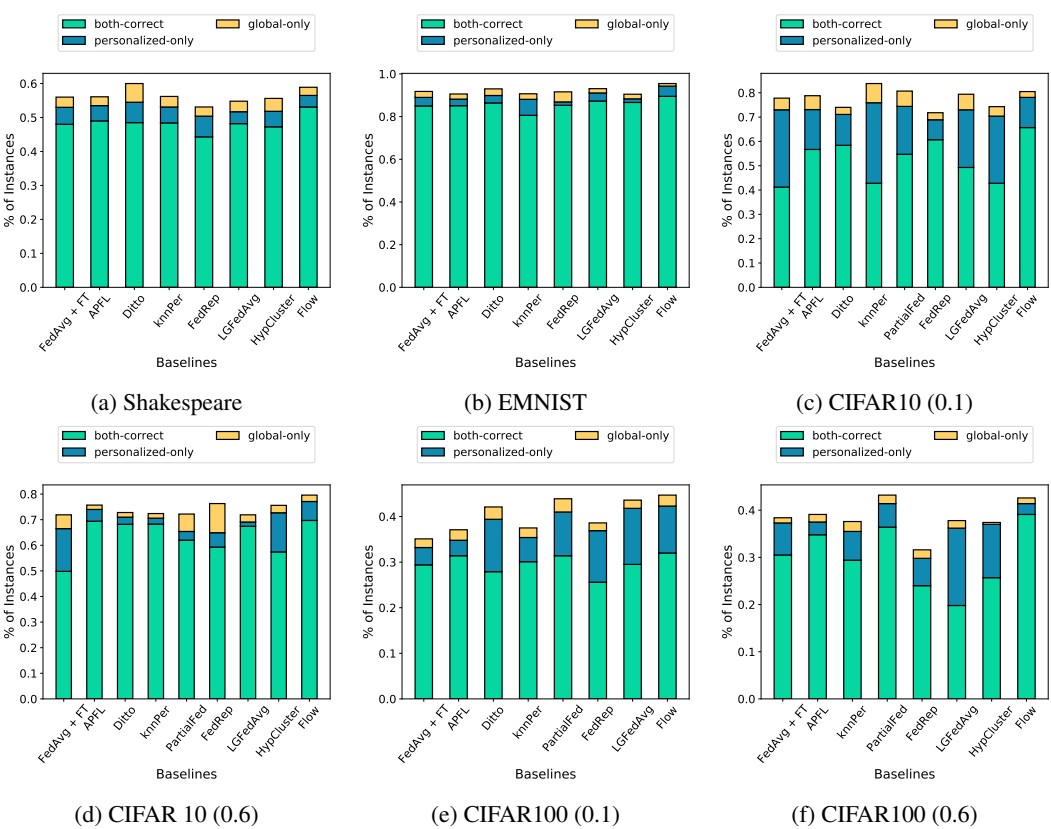

Figure 10: Different combinations of $w_g$ and $w_p$ accuracies.

can only be classified by the personalized model, the probability for taking the global route is lower as the input passes through more layers.

### D.8 Ablation Study: Regularization

Figures 12 and 13 show the validation curves for generalized and personalized accuracy with and without the regularization term used in the policy learning objective as shown in Equation 4. With regularization, we see an improvement of 2.18% (Stackoverflow), 1.86% (Shakespeare), 3.98% (EMNIST), 2.55% (CIFAR10 0.1), 4.36% (CIFAR10 0.6), 0.91% (CIFAR100 0.1), 3.46% (CIFAR100 0.6) for the generalized accuracy. And for the personalized accuracy, we see an improvement of 1.92% (Stackoverflow), 2.02% (Shakespeare), 3.01% (EMNIST), 0.65% (CIFAR10 0.1), 3.98% (CIFAR10 0.6), 2.42% (CIFAR100 0.1), 2.19% (CIFAR100 0.6).

### D.9 Ablation Study: Per-instance Personalization

Figure 14 show the validation curves for 3 *Flow* variants: (a) Per-instance Per-client *Flow*, which is the primary design proposed in this work, (b) Per-instance *Flow*, which makes choices between two global routes solely based on each client's instances, (c) Per-client *Flow*, which is simply FEDAVGFT where the personalization only depends on a client, and not on any specific instances.

With all the datasets, we see a trend of Per-instance *Flow* outperforming Per-client *Flow* by 1.88% (Stackoverflow), 0.82% (Shakespeare), 5.07% (EMNIST), 2.90% (CIFAR10 0.1), 2.41% (CIFAR10 0.6), 7.52% (CIFAR100 0.1), 1.09% (CIFAR100 0.6). We also see the trend of Per-instance *Flow* outperforming Per-Instance Per-Client *Flow* by 3.19% (Stackoverflow), 1.24% (Shakespeare), 0.94% (EMNIST), 0.55% (CIFAR10 0.1), 4.49% (CIFAR10 0.6), 3.88% (CIFAR100 0.1), 1.37% (CIFAR100 0.6).

### D.10 Ablation Study: Soft versus Hard Policy

Table 8 shows the personalized accuracy of the test clients while using soft and hard policies during inference. We see that the accuracy difference between the two designs are statistically insignificant. Hence, using a hard

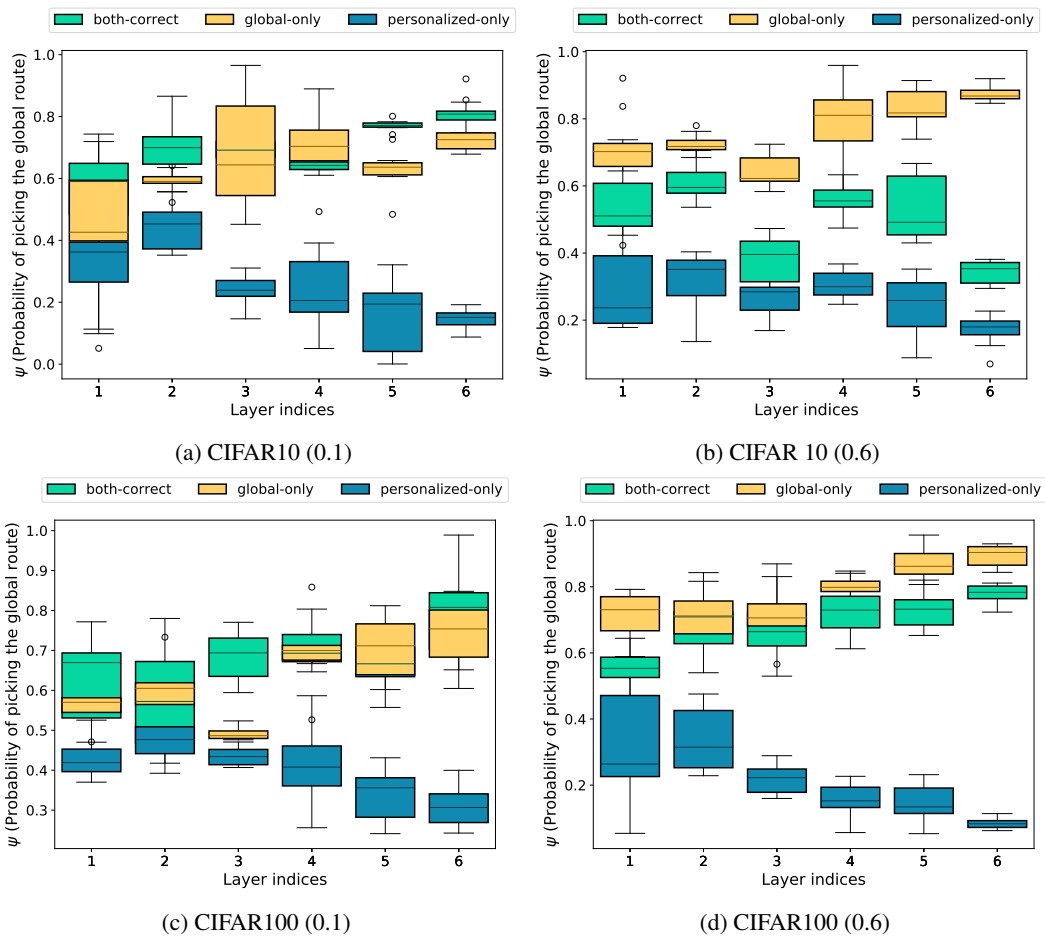

(a) CIFAR10 (0.1)  (b) CIFAR 10 (0.6)

(c) CIFAR100 (0.1)  (d) CIFAR100 (0.6)

Figure 11: Behavior of $\psi_g$ for all instances with respect to each layer of a client with highest loss difference between personalized and global models.

policy for inference not only saves half the compute resources, but also doesn't affect the personalized model's performance.

### D.11  Ablation Study: Dynamic Routing

The accuracy curves are given in Figure 15. The curves show that the phenomena of "dynamic probabilities based on each instance works consistently better than the fixed probabilities for all the clients throughout the training" is indeed observable across all the datasets. The intuition behind that is discussed in Section 5.2, under "Dynamic Routing".

Table 8: Test (personalized) accuracy of two of the *Flow* variants: (a) Soft Policy variant where the probability $\mathbf{q}$ is continuous in the range of $[0, 1]$ during inference. (b) Hard Policy variant where the probability $\mathbf{q}$ is discrete over the set $\{0,1\}$ during inference.

| Datasets | Stackoverflow | Shakespeare | EMNIST |
|---|---|---|---|
| Soft Policy | $29.57\% \pm 0.22\%$ | $57.01\% \pm 0.53\%$ | $94.97\% \pm 1.06\%$ |
| Hard Policy | $29.49\% \pm 0.28\%$ | $56.20\% \pm 0.49\%$ | $94.18\% \pm 1.21\%$ |

| Datasets | CIFAR 10 (0.1) | CIFAR 100 (0.1) | CIFAR 10 (0.6) | CIFAR 100 (0.6) |
|---|---|---|---|---|
| Soft Policy | $77.24\% \pm 1.30\%$ | $42.75\% \pm 0.30\%$ | $77.02\% \pm 0.90\%$ | $39.74\% \pm 0.13\%$ |
| Hard Policy | $76.47\% \pm 1.25\%$ | $42.42\% \pm 0.36\%$ | $77.11\% \pm 0.86\%$ | $40.08\% \pm 0.27\%$ |

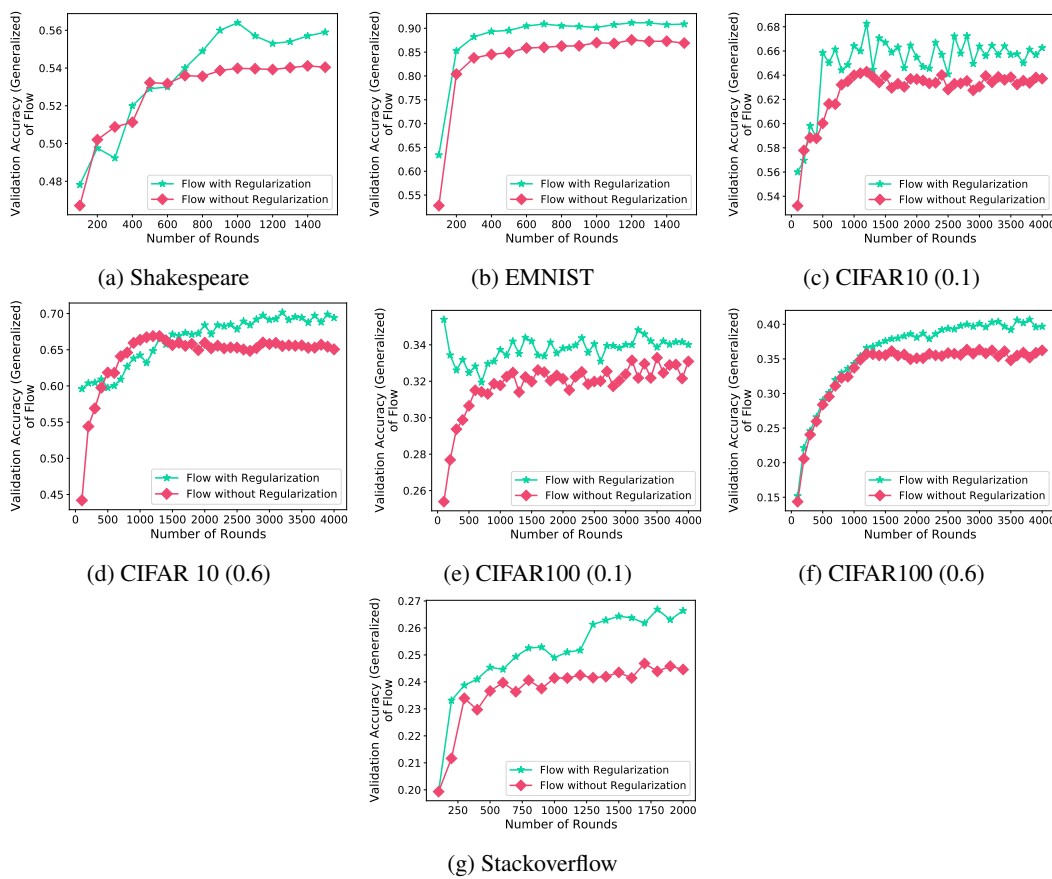

Figure 12: Generalized accuracy of the ablation study on the regularization term used in the policy learning objective.

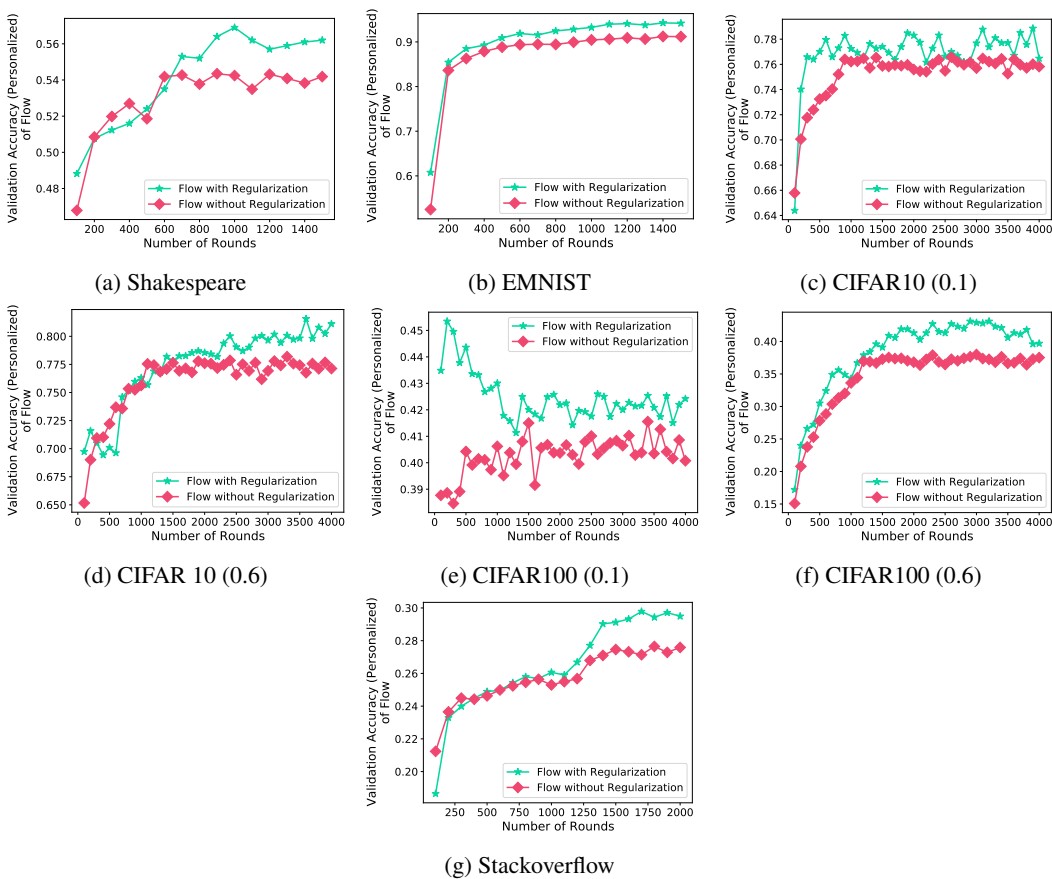

Figure 13: Personalized accuracy of the ablation study on the regularization term used in the policy learning objective.

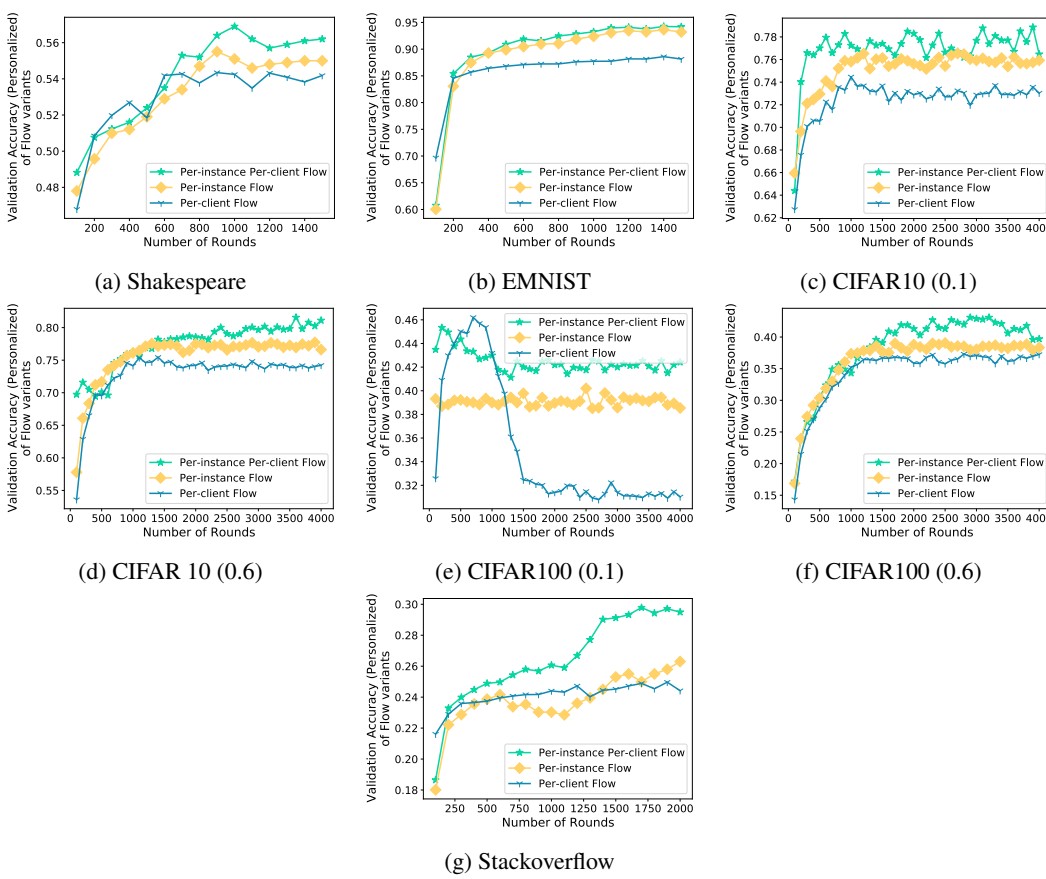

Figure 14: Ablation of the dynamic routing component (Per-client *Flow*), and the local component (Per-instance *Flow*).

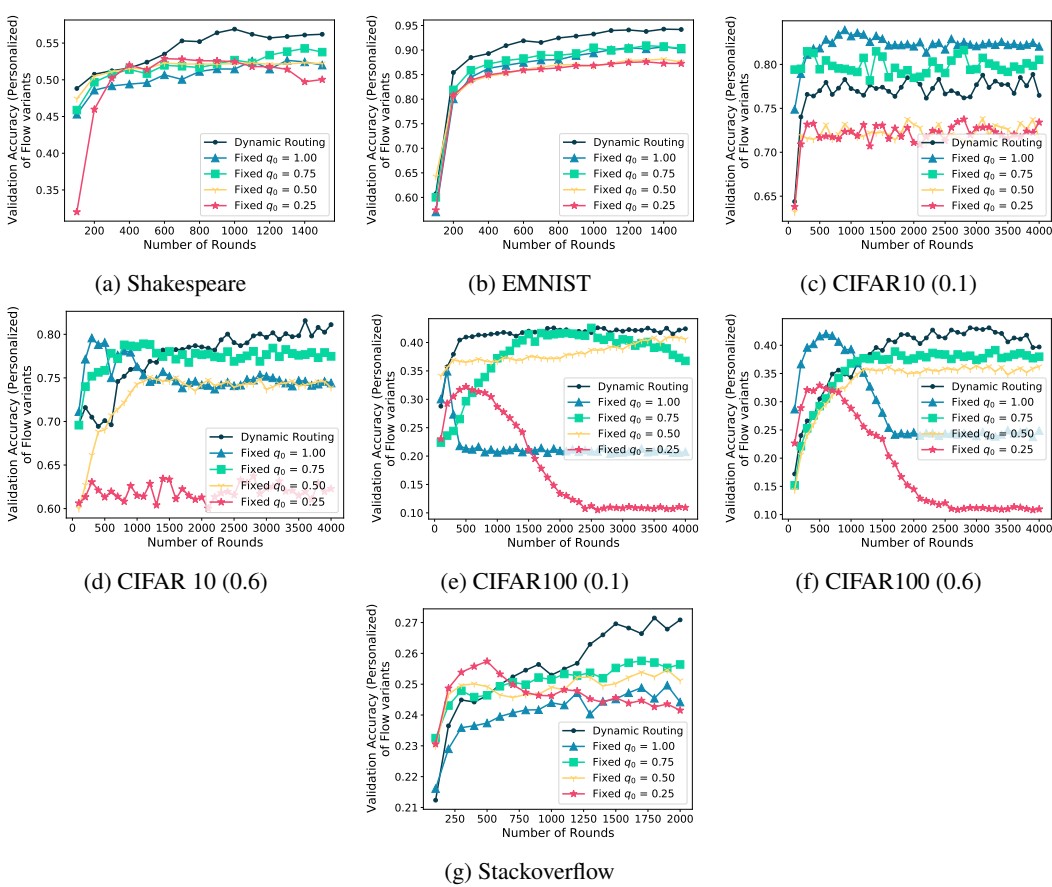

Figure 15: Personalized Accuracy of the Ablation Study on the Dynamic Routing Component.

# E Proofs

## E.1 *Flow*: Detailed

Here we give a detailed version of *Flow* (Algorithm 2) for proving its convergence properties. Here we are assuming that the global and local model output interpolation is model-wise (after the final layer), not layer-wise.

---

**Algorithm 2: *Flow***

**Input:** $R$: Total number of rounds, $r \in [R]$: Round index, $M$: Total number of clients, $m \in [M]$: Client index, $\mathcal{M}$: Set of available clients , $p$: Client sampling rate, $K$: Total local epoch count, $k \in [K]$: Epoch index, $\eta_\ell$: Local learning rate, $w_g^{(r)}$: Global model at $r^{th}$ round, $w_{g,m}^{(r,k)}$: $m^{th}$ client's local update of the global model for $r^{th}$ round and $k^{th}$ epoch, $w_{\ell,m}^{(r,k)}$: $m^{th}$ client's local model for $r^{th}$ round and $k^{th}$ epoch, $w_{p,m}^{(r,k)}$: $m^{th}$ client's personalized model for $r^{th}$ round and $k^{th}$ epoch, $\psi_g^{(r)}$: Global policy model at $r^{th}$ round, $\psi_{g,m}^{(r,k)}$: $m^{th}$ client's routing policy for $r^{th}$ round and $k^{th}$ epoch, $\mathcal{D}_m$: Data distribution of $m^{th}$ client, $\mathcal{S}_m$: Dataset of $m^{th}$ client, $\zeta_{m,\ell}$: Dataset used to train $w_\ell$, $\zeta_{m,g}$: Dataset used to train $w_g$ and $\psi_g$

**Output:** $w_g^{(R+1)}$: Global model at the end of the training

1   Server randomly initializes $w_g^{(1)}$
2   **for** $r \in [R]$ *round* **do**
3      Sample $M$ clients from $\mathcal{M}$ with the rate of $p$
4      Send $w_g^{(r)}, \psi_g^{(r)}$ to all the clients
5      **for** $m \in [M]$ *in parallel* **do**
6         $w_{g,m}^{(r,0)} \leftarrow w_g^{(r)}; \psi_{g,m}^{(r,0)} \leftarrow \psi_g^{(r)}; w_{\ell,m}^{(r,0)} \leftarrow w_{g,m}^{(r,0)}$
7         $\zeta_{m,\ell}, \zeta_{m,g} \leftarrow \mathcal{S}_m$ /* Creating two mutually exclusive datasets       */
8         **for** $k \in [K_1]$ *epochs* **do**
9             $w_{\ell,m}^{(r,k)} \leftarrow w_{\ell,m}^{(r,k-1)} - \eta_\ell \nabla f_m(w_{\ell,m}^{(r,k-1)}; \zeta_{m,\ell})$
10        **end**
11        **for** $k \in [K_2]$ *epochs* **do**
12            $\forall (x_m, y_m) \sim \zeta_{m,g}$, define
$$\tilde{w}_{p,m}^{(r,k-1)}(x_m) \leftarrow \psi_{g,m}^{(r,k-1)}(x_m) \cdot w_{g,m}^{(r,k-1)}(x_m) + (1 - \psi_{g,m}^{(r,k-1)}(x_m)) \cdot w_{\ell,m}^{(r,K)}(x_m)$$
13            $\psi_{g,m}^{(r,k)} \leftarrow \psi_{g,m}^{(r,k-1)} - \eta_\ell \nabla_{\psi_{g,m}^{(r,k-1)}} \left[ f_m(\tilde{w}_{p,m}^{(r,k-1)}; \zeta_{m,g}) \right]$
14            $\forall (x_m, y_m) \sim \zeta_{m,g}$, define
$$w_{p,m}^{(r,k-1)}(x_m) \leftarrow \psi_{g,m}^{(r,k)}(x_m) \cdot w_{g,m}^{(r,k-1)}(x_m) + (1 - \psi_{g,m}^{(r,k)}(x_m)) \cdot w_{\ell,m}^{(r,K)}(x_m)$$
15            $w_{g,m}^{(r,k)} \leftarrow w_{g,m}^{(r,k-1)} - \eta_\ell \nabla_{w_{g,m}^{(r,k-1)}} f_m(w_{p,m}^{(r,k-1)}; \zeta_{m,g})$
16        **end**
17        Send back $w_{g,m}^{(r,K)}, \psi_{g,m}^{(r,K)}, n_m := |\zeta_{m,g}|$
18      **end**
19      $w_g^{(r+1)} \leftarrow \frac{1}{nM} \sum_{m \in [M]} n_m w_{g,m}^{(r,K)}$
20      $\psi_g^{(r+1)} \leftarrow \frac{1}{nM} \sum_{m \in [M]} n_m \psi_{g,m}^{(r,K)}$
21 **end**

---

## E.2 Basics

We perform theoretic analysis of *Flow* based on the following setup: There are total $M$ clients. A client is denoted by a unique integer $m$ associated with it where $m \in [M]$. Each client $m$ has a dataset $\mathcal{S}_m = \{(x_m^{(i)}, y_m^{(i)}); i \in [n_m]\}$ where $(x_m^{(i)}, y_m^{(i)})$ has been sampled from $\mathcal{D}_m$ distribution of the $m^{th}$ client. $n_m = |\mathcal{S}_m|$ is the total sample count of the $m^{th}$ client. Total sample count across all the participating client is $n = \sum_{m \in [M]} n_m$. The ratio of $m^{th}$ client's sample count to total sample count is $\alpha = \frac{n_m}{n}$.

The global distribution is defined as $\mathcal{D} = \sum_{m \in [M]} q_m \mathcal{D}_m$ where $q_m$ is the weight associated with $m^{th}$ client and $\sum_{m \in [M]} q_m = 1$.

Note that $w_{p,m}$ is a combination of outputs of $w_{g,m}$ (Global parameters) and $w_{\ell,m}$ (Local parameters) on each layer. For tractability of analysis, we will assume that the combination is only after the last layer. Hence,

$$w_{p,m}(x_m) \leftarrow \psi_{g,m}(x_m)w_{g,m}(x_m) + (1 - \psi_{g,m}(x_m))w_{\ell,m}(x_m).$$

The local model update rule is,

$$w_{\ell,m}^{(r,k)} \leftarrow w_{\ell,m}^{(r,k-1)} - \eta_\ell \nabla f_m(w_{\ell,m}^{(r,k-1)}(x_m), y_m)$$

where $w_{\ell,m}^{(r,0)} = w_{g,m}^{(r,0)} = w_g^{(r)}$. Indices $r \in [R]$ and $k \in [K]$ are the global round and the local epoch indices. The policy update rule is,

$$\psi_{g,m}^{(r,k)} \leftarrow \psi_{g,m}^{(r,k-1)} - \eta_\ell \nabla_{\psi_g} f_m(w_{p,m}^{(r,k-1)}(x_m), y_m).$$

The global model update rule is,

$$w_{g,m}^{(r,k)} \leftarrow w_{g,m}^{(r,k-1)} - \eta_\ell \nabla_{w_g} f_m(w_{p,m}^{(r,k-1)}(x_m), y_m).$$

We list out all the optimization problems relevant to *Flow*:

- **Local true risk of the personalized model**

$$F_m(w_{p,m}) := \mathbb{E}_{(x_m, y_m \sim \mathcal{D}_m)}[f_m(w_{p,m}(x_m), y_m)]$$

   where $f_m$ is a loss function associated with the $m^{th}$ client.

- **Local empirical risk of the personalized model**

$$\hat{F}_m(w_{p,m}) := \frac{1}{n_m} \sum_{i \in [n_m]} f_m(w_{p,m}(x_m^{(i)}), y_m^{(i)})$$

- **Local true risk of the global model**

$$F_m(w_{g,m}) := \mathbb{E}_{(x_m, y_m \sim \mathcal{D}_m)}[f_m(w_{g,m}(x_m), y_m)]$$

- **Local empirical risk of the global model**

$$\hat{F}_m(w_{g,m}) := \frac{1}{n_m} \sum_{i \in [n_m]} f_m(w_{g,m}(x_m^{(i)}), y_m^{(i)})$$

- **Local minimizer of local empirical risk of the personalized model**

$$w_{p,m}^* \in \mathcal{H} \text{ such that } \hat{F}_m(w_{p,m}) \geq \hat{F}_m(w_{p,m}^*); \; \forall w_{p,m} \in \mathcal{H}, \; \exists \epsilon > 0, \; ||w_{p,m} - w_{p,m}^*|| < \epsilon$$

- **Global true risk of the global model**

$$F(w_g) = \frac{1}{nM} \sum_{m \in [M]} n_m \mathbb{E}_{(x_m, y_m) \sim \mathcal{S}_m}[f_m(w_{g,m}(x_m), y_m)] \text{ where } n = |\mathcal{S}| = |\bigcup_{m \in [M]} \mathcal{S}_m|$$

- **Global empirical risk of the global model**

$$\hat{F}(w_g) = \frac{1}{nM} \sum_{m \in [M]} n_m \hat{F}_m(w_{g,m}(x_m), y_m) = \frac{1}{nM} \sum_{m \in [M]} \sum_{i \in [n_m]} f_m(w_{g,m}(x_m^{(i)}), y_m^{(i)})$$

- **Local minimizer of global empirical risk**

$$w_g^* \in \mathcal{H} \text{ such that } \hat{F}(w_g) \geq \hat{F}(w_g^*); \; \forall w_g \in \mathcal{H}, \; \exists \epsilon > 0, \; ||w_g - w_g^*|| < \epsilon$$

We also use the following assumptions similar to [11, 18, 6]:

**Assumption E.1** (Strong Convexity). $f_m$ is $\mu$-convex for $\mu \geq 0$. Hence,

$$\langle \nabla f_m(w), v - w \rangle \leq f_m(v) - f_m(w) - \frac{\mu}{2}||w - v||^2, \; \forall m \in [M] \text{ and } w, v \in \mathcal{H}.$$

We also generalize our convergence analysis for $\mu = 0$, general convex cases.

**Assumption E.2** (Smoothness). The gradient of $f_m$ is $\beta$-Lipschitz,

$$||\nabla f_m(w) - \nabla f_m(v)|| \leq \beta||w - v||, \; \forall m \in [M] \text{ and } w, v \in \mathcal{H}.$$

**Assumption E.3** (Bounded Local Variance). $h_m(w) := \nabla f_m(w(x_m), y_m)$ is an unbiased stochastic gradient of $f_m$ with variance bounded by $\sigma_\ell^2$.

$$\mathbb{E}_{(x_m, y_m \sim \mathcal{D}_m)} ||h_m(w) - \nabla f_m(w(x_m), y_m)||^2 \leq \sigma_\ell^2, \; \forall m \in [M] \text{ and } w \in \mathcal{H}.$$

**Assumption E.4** ($(G, B)$-Bounded Gradient Dissimilarity). There exists constants $G \geq 0$ and $B \geq 1$ such that

$$\frac{1}{M} \sum_{m \in [M]} ||\nabla f_m(w)||^2 \leq G^2 + 2\beta B^2 (F(w) - F(w^*))$$

for a convex $f_m$. And for a non-convex $f_m$,

$$\frac{1}{M} \sum_{m \in [M]} ||\nabla f_m(w)||^2 \leq G^2 + B^2 ||\nabla F(w)||^2.$$

The derivation is given in Section D.1 of Scaffold [18].

We also use a definition to quantify the diversity of a client's gradient with respect to the global gradient as defined in [29]:

**Definition E.5** (Gradient Diversity). The difference between gradients of the $m^{th}$ client's true risk and the global true risk based on the global model $w$ is,

$$\delta_m = \sup_{w \in \mathcal{H}} ||\nabla f_m(w) - \nabla F(w)||^2$$

## E.3 Convergence Proof for the Global Model: Convex (Strong and General) Cases

A client's local update for one local epoch on the global model, starting with $w_{g,m}^{(r,0)} \leftarrow w_g^{(r)}$, is

$$w_{g,m}^{(r,k+1)} = w_{g,m}^{(r,k)} - \eta_\ell h_m(w_{p,m}^{(r,k)}). \tag{8}$$

And a client's local update for $K$ epochs on the global model, would be

$$w_{g,m}^{(r,K)} = w_{g,m}^{(r,0)} - \eta_\ell \sum_{k=1}^{K} h_m(w_{p,m}^{(r,k-1)}) \tag{9}$$

$$= w_{g,m}^{(r,0)} - \eta_\ell \sum_{k=1}^{K} h_m(\psi_{g,m}^{(r,k)}(x_m) w_{g,m}^{(r,k-1)}(x_m) + (1 - \psi_{g,m}^{(r,k)}(x_m)) w_{\ell,m}^{(r,K)}(x_m), y_m). \tag{10}$$

In both the above cases, the gradient is with respect to $w_g$ parameters.

The global model update is,

$$w_g^{(r+1)} = \frac{1}{nM} \sum_{m \in [M]} n_m w_{g,m}^{(r,K)} \tag{11}$$

We first start with a lemma which binds the deviation between the local model $w_{\ell,m}^{(r,K)}$ and the global model starting point $w_g^{(r)}$ for it at round $r$.

**Lemma E.6** (Local model progress). *If $m^{th}$ client's objective function $f_m$ satisfies Assumptions E.2, E.3, and condition $\eta_\ell \leq \frac{1}{\beta\sqrt{2K(K-1)}}$ in Algorithm 2, the following is satisfied:*

$$\mathbb{E}||w_{\ell,m}^{(r,K)} - w_{\ell,m}^{(r,0)}||^2 \leq 6K^2 \eta_\ell^2 \mathbb{E}||\nabla f_m(w_g^{(r)})||^2 + 3K\eta_\ell^2 \sigma_\ell^2$$

*Proof.*

$$\mathbb{E}||w_{\ell,m}^{(r,K)} - w_{\ell,m}^{(r,0)}||^2 = \mathbb{E}||w_{\ell,m}^{(r,K-1)} - \eta_\ell \nabla f_m(w_{\ell,m}^{(r,K-1)}) - w_{\ell,m}^{(r,0)}||^2 \tag{12}$$

$$\leq \left(1 + \frac{1}{K-1}\right) \mathbb{E}||w_{\ell,m}^{(r,K-1)} - w_{\ell,m}^{(r,0)}||^2 + K\eta_\ell^2 \mathbb{E}||\nabla f_m(w_{\ell,m}^{(r,K-1)})||^2 + \eta_\ell^2 \sigma_\ell^2 \tag{13}$$

Here we have used triangle inequality and variance separation.

$$\leq \left(1 + \frac{1}{K-1}\right) \mathbb{E}||w_{\ell,m}^{(r,K-1)} - w_{\ell,m}^{(r,0)}||^2 + K\eta_\ell^2 \mathbb{E}||\nabla f_m(w_{\ell,m}^{(r,K-1)})||^2 + \eta_\ell^2 \sigma_\ell^2 \tag{14}$$

$$\leq \left(1 + \frac{1}{K-1}\right) \mathbb{E}||w_{\ell,m}^{(r,K-1)} - w_{\ell,m}^{(r,0)}||^2 + \eta_\ell^2 \sigma_\ell^2$$
$$+ K\eta_\ell^2 \mathbb{E}||\nabla f_m(w_{\ell,m}^{(r,K-1)}) - \nabla f_m(w_{\ell,m}^{(r,0)}) + \nabla f_m(w_{\ell,m}^{(r,0)})||^2 \tag{15}$$

$$\leq \left(1 + \frac{1}{K-1}\right) \mathbb{E}||w_{\ell,m}^{(r,K-1)} - w_{\ell,m}^{(r,0)}||^2 + 2K\eta_\ell^2 \mathbb{E}||\nabla f_m(w_{\ell,m}^{(r,K-1)}) - \nabla f_m(w_{\ell,m}^{(r,0)})||^2$$
$$+ 2K\eta_\ell^2 \mathbb{E}||\nabla f_m(w_{\ell,m}^{(r,0)})||^2 + \eta_\ell^2 \sigma_\ell^2 \tag{16}$$

$$\leq \left(1 + \frac{1}{K-1}\right) \mathbb{E}||w_{\ell,m}^{(r,K-1)} - w_{\ell,m}^{(r,0)}||^2 + 2K\beta^2\eta_\ell^2 \mathbb{E}||w_{\ell,m}^{(r,K-1)} - w_{\ell,m}^{(r,0)}||^2$$
$$+ 2K\eta_\ell^2 \mathbb{E}||\nabla f_m(w_{\ell,m}^{(r,0)})||^2 + \eta_\ell^2 \sigma_\ell^2 \tag{17}$$

Assuming $\eta_\ell \leq \frac{1}{\beta\sqrt{2K(K-1)}}$, we get

$$\mathbb{E}||w_{\ell,m}^{(r,K)} - w_{\ell,m}^{(r,0)}||^2 \leq \left(1 + \frac{2}{K-1}\right) \mathbb{E}||w_{\ell,m}^{(r,K-1)} - w_{\ell,m}^{(r,0)}||^2$$
$$+ 2K\eta_\ell^2 \mathbb{E}||\nabla f_m(w_{\ell,m}^{(r,0)})||^2 + \eta_\ell^2 \sigma_\ell^2 \tag{18}$$

Unrolling the above recursion,

$$\mathbb{E}||w_{\ell,m}^{(r,K)} - w_{\ell,m}^{(r,0)}||^2 \leq \sum_{i=1}^{K} \left(2K\eta_\ell^2 \mathbb{E}||\nabla f_m(w_{\ell,m}^{(r,0)})||^2 + \eta_\ell^2 \sigma_\ell^2\right)\left(1 + \frac{2}{K-1}\right)^i \tag{19}$$

$$\leq 3K\left(2K\eta_\ell^2 \mathbb{E}||\nabla f_m(w_{\ell,m}^{(r,0)})||^2 + \eta_\ell^2 \sigma_\ell^2\right) \tag{20}$$

$$= 6K^2\eta_\ell^2 \mathbb{E}||\nabla f_m(w_{\ell,m}^{(r,0)})||^2 + 3K\eta_\ell^2 \sigma_\ell^2 \tag{21}$$

$$\square$$

Now we move forward to a lemma which binds the deviation between the local version of the global model $w_{g,m}^{(r,k)}$ and the global model starting point $w_g^{(r)}$ for it round $r$.

**Lemma E.7** (Local version of the global model progress). *If $m^{th}$ client's objective function $f_m$ satisfies Assumptions E.1, E.2, E.3, and condition $\eta_\ell \leq \frac{1}{\beta\sqrt{2k}}$ in Algorithm 2, the following is satisfied:*

$$\mathbb{E}||w_{g,m}^{(r,k)} - w_{g,m}^{(r,0)}||^2 \leq 8k^3\eta_\ell^2 \mathbb{E}||\psi_{g,m}^{(r,k)}||^2 \mathbb{E}||\nabla f_m(w_g^{(r)})||^2 + 4k\eta_\ell^2 \sigma_\ell^2$$

*Proof.* We start by expanding $w_{g,m}^{(r,k)}$ in terms of its previous epoch iterate.

$$\mathbb{E}||w_{g,m}^{(r,k)} - w_{g,m}^{(r,0)}||^2 = \mathbb{E}||w_{g,m}^{(r,k-1)} - \eta_\ell \nabla_{w_{g,m}^{(r,k-1)}} f_m(w_{p,m}^{(r,k-1)}) - w_{g,m}^{(r,0)}||^2 \tag{22}$$

Using triangle inequality and separation of variance, we get,

$$\leq \left(1 + \frac{1}{k-1}\right) \mathbb{E}||w_{g,m}^{(r,k-1)} - w_{g,m}^{(r,0)}||^2 + k\eta_\ell^2 \mathbb{E}||\nabla_{w_{g,m}^{(r,k-1)}} f_m(w_{p,m}^{(r,k-1)})||^2 + \eta_\ell^2 \sigma_\ell^2 \tag{23}$$

Using the convex property of $f_m$, we get

$$\leq \left(1 + \frac{1}{k-1}\right) \mathbb{E}||w_{g,m}^{(r,k-1)} - w_{g,m}^{(r,0)}||^2 + k\eta_\ell^2 \mathbb{E}||\psi_{g,m}^{(r,k)} \nabla f_m(w_{g,m}^{(r,k-1)})||^2 + \eta_\ell^2 \sigma_\ell^2 \tag{24}$$

$$\leq \left(1 + \frac{1}{k-1}\right) \mathbb{E}||w_{g,m}^{(r,k-1)} - w_{g,m}^{(r,0)}||^2 + \eta_\ell^2 \sigma_\ell^2$$
$$+ k\eta_\ell^2 \mathbb{E}||\psi_{g,m}^{(r,k)}(\nabla f_m(w_{g,m}^{(r,k-1)}) - \nabla f_m(w_{g,m}^{(r,0)}) + \nabla f_m(w_{g,m}^{(r,0)}))||^2 \tag{25}$$

$$\leq \left(1 + \frac{1}{k-1}\right) \mathbb{E}||w_{g,m}^{(r,k-1)} - w_{g,m}^{(r,0)}||^2 + \eta_\ell^2 \sigma_\ell^2 + 2k\eta_\ell^2 \mathbb{E}||\psi_{g,m}^{(r,k)}||^2 \mathbb{E}||\nabla f_m(w_{g,m}^{(r,0)}))||^2$$
$$+ 2k\eta_\ell^2 \mathbb{E}||\psi_{g,m}^{(r,k)}||^2 \mathbb{E}||(\nabla f_m(w_{g,m}^{(r,k-1)}) - \nabla f_m(w_{g,m}^{(r,0)})||^2 \tag{26}$$

$$\leq \left(1 + \frac{1}{k-1} + 2k\eta_\ell^2\beta^2 \mathbb{E}||\psi_{g,m}^{(r,k)}||^2\right) \mathbb{E}||w_{g,m}^{(r,k-1)} - w_{g,m}^{(r,0)}||^2 + \eta_\ell^2 \sigma_\ell^2$$
$$+ 2k\eta_\ell^2 \mathbb{E}||\psi_{g,m}^{(r,k)}||^2 \mathbb{E}||\nabla f_m(w_{g,m}^{(r,0)}))||^2 \tag{27}$$

Unrolling the recursion,

$$\mathbb{E}||w_{g,m}^{(r,k)} - w_{g,m}^{(r,0)}||^2 \leq \sum_{i=1}^{k} \left( \left( 2k\eta_\ell^2 \mathbb{E}||\psi_{g,m}^{(r,k)}||^2 \mathbb{E}||\nabla f_m(w_{g,m}^{(r,0)})||^2 + \eta_\ell^2 \sigma_\ell^2 \right) \right.$$
$$\left. \cdot \left( 1 + \frac{1}{k-1} + 2k\eta_\ell^2 \beta^2 \mathbb{E}||\psi_{g,m}^{(r,k)}||^2 \right)^i \right) \tag{28}$$

Assuming that $\eta_\ell \leq \frac{1}{\beta\sqrt{2k}}$ we get $k\eta_\ell^2\beta^2 \leq 1$,

$$\mathbb{E}||w_{g,m}^{(r,k)} - w_{g,m}^{(r,0)}||^2 \leq \left( 2k\eta_\ell^2 \sum_{i=1}^{k} \mathbb{E}||\psi_{g,m}^{(r,i)}||^2 \mathbb{E}||\nabla f_m(w_{g,m}^{(r,0)})||^2 + \eta_\ell^2 \sigma_\ell^2 \right) \sum_{i=1}^{k} \left( 1 + \frac{1}{k-1} + 2 \right)^i \tag{29}$$

$$\leq 4k \left( 2k^2\eta_\ell^2 \mathbb{E}||\psi_{g,m}^{(r,k)}||^2 \mathbb{E}||\nabla f_m(w_g^{(r)})||^2 + \eta_\ell^2 \sigma_\ell^2 \right) \tag{30}$$

$$\leq 8k^3\eta_\ell^2 \mathbb{E}||\psi_{g,m}^{(r,k)}||^2 \mathbb{E}||\nabla f_m(w_g^{(r)})||^2 + 4k\eta_\ell^2 \sigma_\ell^2 \tag{31}$$

$\square$

**Lemma E.8** (Deviation of the personalized model from the global model). *If $m^{th}$ client's objective function $f_m$ satisfies Assumptions E.1, E.2, E.3, and condition $\eta_\ell \leq \min\left( \frac{1}{\beta\sqrt{2K(K-1)}}, \frac{1}{\beta\sqrt{2K}} \right)$ in Algorithm 2, the following is satisfied:*

$$\mathbb{E}||w_{p,m}^{(r,k)} - w_{g,m}^{(r,0)}||^2 \leq 16k^3\eta_\ell^2 \mathbb{E}||1 - \psi_{g,m}^{(r,k)}||^2 \mathbb{E}||\nabla f_m(w_g^{(r)})||^2 + 8k\eta_\ell^2 \sigma_\ell^2 \mathbb{E}||\psi_{g,m}^{(r,k)}||^2$$
$$+ 12K^2\eta_\ell^2 \mathbb{E}||1 - \psi_{g,m}^{(r,k)}||^2 \mathbb{E}||\nabla f_m(w_g^{(r)})||^2 + 6K\eta_\ell^2 \sigma_\ell^2 \mathbb{E}||\psi_{g,m}^{(r,k)}||^2$$

*Proof.*

$$\mathbb{E}||w_{p,m}^{(r,k)} - w_{g,m}^{(r,0)}||^2 = \mathbb{E}||\psi_{g,m}^{(r,k)} w_{g,m}^{(r,k)} + (1 - \psi_{g,m}^{(r,k)})w_{\ell,m}^{(r,K)} - w_{g,m}^{(r,0)}||^2 \tag{32}$$

$$= \mathbb{E}||\psi_{g,m}^{(r,k)}(w_{g,m}^{(r,k)} - w_{\ell,m}^{(r,k)}) + (w_{\ell,m}^{(r,K)} - w_{g,m}^{(r,0)})||^2 \tag{33}$$

$$= \mathbb{E}||\psi_{g,m}^{(r,k)}(w_{g,m}^{(r,k)} - w_{g,m}^{(r,0)} + w_{g,m}^{(r,0)} - w_{\ell,m}^{(r,k)}) + (w_{\ell,m}^{(r,K)} - w_{g,m}^{(r,0)})||^2 \tag{34}$$

$$\leq 2\mathbb{E}||\psi_{g,m}^{(r,k)}(w_{g,m}^{(r,k)} - w_{g,m}^{(r,0)})||^2 + 2\mathbb{E}||(1 - \psi_{g,m}^{(r,k)})(w_{\ell,m}^{(r,K)} - w_{\ell,m}^{(r,0)})||^2 \tag{35}$$

Using lemmas E.6 and E.7,

$$\mathbb{E}||w_{p,m}^{(r,k)} - w_{g,m}^{(r,0)}||^2 \leq 2\mathbb{E}||\psi_{g,m}^{(r,k)}||^2 \left( 8k^3\eta_\ell^2 \mathbb{E}||\psi_{g,m}^{(r,k)}||^2 \mathbb{E}||\nabla f_m(w_g^{(r)})||^2 + 6K^2\eta_\ell^2 \mathbb{E}||\nabla f_m(w_g^{(r)})||^2 \right)$$
$$+ 2\mathbb{E}||1 - \psi_{g,m}^{(r,k)}||^2 \left( 4k\eta_\ell^2 \sigma_\ell^2 + 3K\eta_\ell^2 \sigma_\ell^2 \right) \tag{36}$$

$$\leq 16k^3\eta_\ell^2 \mathbb{E}||1 - \psi_{g,m}^{(r,k)}||^2 \mathbb{E}||\nabla f_m(w_g^{(r)})||^2 + 8k\eta_\ell^2 \sigma_\ell^2 \mathbb{E}||\psi_{g,m}^{(r,k)}||^2$$
$$+ 12K^2\eta_\ell^2 \mathbb{E}||1 - \psi_{g,m}^{(r,k)}||^2 \mathbb{E}||\nabla f_m(w_g^{(r)})||^2 + 6K\eta_\ell^2 \sigma_\ell^2 \mathbb{E}||\psi_{g,m}^{(r,k)}||^2 \tag{37}$$

$\square$

**Theorem E.9** (Convergence of the Global Model for Convex Cases). *If each client's objective function $f_m$ satisfies Assumptions E.1, E.2, E.3, E.4 using the learning rate $\frac{1}{\mu R} \leq \eta_\ell \leq \min\left( \frac{1}{4\sqrt{10}\beta BK^2}, \frac{1}{8B^2\beta} \right)$ in Algorithm 2, then the following convergence holds:*
*(Strong Convex Case)*

$$\mathbb{E}\left[ F(w_g^{(R)}) \right] - F(w_g^*) \leq \frac{\mu}{\mathbf{q}_0^2 K} \mathbb{E}||w_g^{(0)} - w_g^*||^2 \exp\left( -\frac{\eta_\ell \mu KR}{2M} \right) + \frac{2G^2}{\mathbf{q}_0^2 \mu R}$$
$$+ \frac{40K^2\beta}{\mu^2 R^2} \left( \frac{\beta^2}{\mu R} + 1 \right) \frac{\mathbf{q}_1^2}{\mathbf{q}_0^2} G^2 + \frac{28K\beta}{\mu^2 R^2} \left( \frac{2\beta^2 K}{\mu^2 R^2} + 1 \right) \sigma_\ell^2$$

*(General Convex Case)*

$$\mathbb{E}\left[ F(w_g^{(R)}) \right] - F(w_g^*) \leq \frac{1}{\eta_\ell K \mathbf{q}_0^2 (R+1)} \mathbb{E}||w_g^{(0)} - w_g^*||^2 + \eta_\ell \left( \frac{2G^2}{\mathbf{q}_0^2} \right)^{1/2} +$$
$$+ \eta_\ell^2 \left( 40K^2\beta \frac{\mathbf{q}_1^2}{\mathbf{q}_0^2} G^2 \right)^{1/3} + \eta_\ell^3 \left( 40K^2\beta^3 \frac{\mathbf{q}_1^2}{\mathbf{q}_0^2} G^2 \right)^{1/4} + \eta_\ell^2 \left( 28K\beta\sigma_\ell^2 \right)^{1/3} + \eta_\ell^4 \left( 56K\beta^3\sigma_\ell^2 \right)^{1/5}$$

*where $\mathbf{q}_{0/1}^2$ are the probabilities of picking global/local routes averaged over all the instances sampled from the global distribution.*

*Proof.* From the update rules stated in Equations 10 and 11:

$$w_g^{(r+1)} - w_g^* = \frac{1}{nM} \sum_{m \in [M]} n_m \left[ w_{g,m}^{(r)} - \eta_\ell \sum_{k=1}^{K} h_m(w_{p,m}^{(r,k-1)}) \right] - w_g^* \tag{38}$$

$$= \frac{1}{nM} \sum_{m \in [M]} n_m w_{g,m}^{(r)} - \frac{\eta_\ell}{nM} \sum_{m \in [M]} n_m \sum_{k=1}^{K} h_m(w_{p,m}^{(r,k-1)}) - w_g^* \tag{39}$$

$$= w_g^{(r)} - w_g^* - \frac{\eta_\ell}{nM} \sum_{m \in [M]} n_m \sum_{k=1}^{K} h_m(w_{p,m}^{(r,k-1)}) \tag{40}$$

Taking squared norm and expectation on both sides with respect to the choice of $h_m$,

$$\mathbb{E}\left[||w_g^{(r+1)} - w_g^*||^2\right] \leq \mathbb{E}\left[||w_g^{(r)} - w_g^*||^2\right] - 2\eta_\ell \left\langle \frac{1}{nM} \sum_{m \in [M]} n_m \sum_{k=1}^{K} \mathbb{E}[h_m(w_{p,m}^{(r,k-1)})], w_g^{(r)} - w_g^* \right\rangle$$

$$+ \eta_\ell^2 \mathbb{E}\left[\left|\left| \frac{1}{nM} \sum_{m \in [M]} n_m \sum_{k=1}^{K} h_m(w_{p,m}^{(r,k-1)}) \right|\right|^2\right] \tag{41}$$

Separating mean and variance according to Lemma 4 of Scaffold [18],

$$\leq \mathbb{E}\left[||w_g^{(r)} - w_g^*||^2\right] \underbrace{- 2\eta_\ell \left\langle \frac{1}{nM} \sum_{m \in [M]} n_m \sum_{k=1}^{K} \mathbb{E}[\nabla_{w_{g,m}^{(r,k-1)}} f_m(w_{p,m}^{(r,k-1)})], w_g^{(r)} - w_g^* \right\rangle}_{T_1}$$

$$\underbrace{+ \eta_\ell^2 \mathbb{E}\left[\left|\left| \frac{1}{nM} \sum_{m \in [M]} n_m \sum_{k=1}^{K} \nabla_{w_{g,m}^{(r,k-1)}} f_m(w_{p,m}^{(r,k-1)}) \right|\right|^2\right]}_{T_2} + \frac{\eta_\ell^2 \sigma_\ell^2 K}{M} \tag{42}$$

**Bounding $T_1$**

$$T_1 = -2\eta_\ell \left\langle \frac{1}{nM} \sum_{m \in [M]} n_m \sum_{k=1}^{K} \mathbb{E}[\nabla_{w_{g,m}^{(r,k-1)}} f_m(w_{p,m}^{(r,k-1)})], w_g^{(r)} - w_g^* \right\rangle \tag{43}$$

$$= 2\eta_\ell \left\langle \frac{1}{nM} \sum_{m \in [M]} n_m \sum_{k=1}^{K} \mathbb{E}[\nabla_{w_{g,m}^{(r,k-1)}} f_m(w_{p,m}^{(r,k-1)})], w_g^* - w_g^{(r)} \right\rangle \tag{44}$$

Using perturbed strong convexity lemma (Lemma 5) from [18], we get,

$$T_1 \leq \frac{2\eta_\ell}{nM} \sum_{m \in [M]} n_m \sum_{k=1}^{K} \left( \mathbb{E}[\nabla f_m(w_g^*)] - \nabla f_m(w_g^{(r)}) - \frac{\mu}{4} \mathbb{E}||w_g^{(r)} - w_g^*||^2 + \beta \underbrace{\mathbb{E}||w_{p,m}^{(r,k-1)} - w_g^{(r)}||^2}_{\text{Lemma E.8}} \right) \tag{45}$$

$$\leq -2\eta_\ell K \left( \mathbb{E}[F(w_g^{(r)})] - F(w_g^*) \right) - \frac{\eta_\ell \mu K}{2M} \mathbb{E}||w_g^{(r)} - w_g^*||^2$$

$$+ \frac{2\eta_\ell \beta}{nM} \sum_{m \in [M]} n_m \sum_{k=1}^{K} \left( 16k^3 \eta_\ell^2 \mathbb{E}||1 - \psi_{g,m}^{(r,k)}||^2 \mathbb{E}||\nabla f_m(w_g^{(r)})||^2 + 8k\eta_\ell^2 \sigma_\ell^2 \mathbb{E}||\psi_{g,m}^{(r,k)}||^2 \right.$$

$$\left. + 12K^2 \eta_\ell^2 \mathbb{E}||1 - \psi_{g,m}^{(r,k)}||^2 \mathbb{E}||\nabla f_m(w_g^{(r)})||^2 + 6K \eta_\ell^2 \sigma_\ell^2 \mathbb{E}||\psi_{g,m}^{(r,k)}||^2 \right) \tag{46}$$

$$\leq -2\eta_\ell K \left( \mathbb{E}[F(w_g^{(r)})] - F(w_g^*) \right) - \frac{\eta_\ell \mu K}{2M} \mathbb{E}||w_g^{(r)} - w_g^*||^2$$

$$+ \frac{2\eta_\ell \beta}{nM} \sum_{m \in [M]} n_m \left( 16K^4 \eta_\ell^2 \mathbb{E}||\nabla f_m(w_g^{(r)})||^2 \mathbb{E}||1 - \psi_{g,m}^{(r,K)}||^2 + 8K^2 \eta_\ell^2 \sigma_\ell^2 \mathbb{E}||\psi_{g,m}^{(r,K)}||^2 \right.$$

$$\left. + 12K^3 \eta_\ell^2 \mathbb{E}||\nabla f_m(w_g^{(r)})||^2 \mathbb{E}||1 - \psi_{g,m}^{(r,K)}||^2 + 6K^2 \eta_\ell^2 \sigma_\ell^2 \mathbb{E}||\psi_{g,m}^{(r,K)}||^2 \right), \tag{47}$$

Next, using Assumption E.4,

$$
\begin{aligned}
T_1 \leq &-2\eta_\ell K\left(\mathbb{E}[F(w_g^{(r)})] - F(w_g^*)\right) - \frac{\eta_\ell \mu K}{2M}\mathbb{E}||w_g^{(r)} - w_g^*||^2 \\
&+ 32\eta_\ell^3 K^4 \beta \mathbb{E}||1 - \psi_g^{(r)}||^2\left(G^2 + 2\beta B^2\left(\mathbb{E}\left[F(w_g^{(r)})\right] - F(w_g^*)\right)\right) + 16\eta_\ell^3 K^2 \beta \sigma_\ell^2 \mathbb{E}||\psi_g^{(r)}||^2 \\
&+ 24\eta_\ell^3 K^3 \beta \mathbb{E}||1 - \psi_g^{(r)}||^2\left(G^2 + 2\beta B^2\left(\mathbb{E}\left[F(w_g^{(r)})\right] - F(w_g^*)\right)\right) + 12\eta_\ell^3 K^2 \beta \sigma_\ell^2 \mathbb{E}||\psi_g^{(r)}||^2
\end{aligned}
\tag{48}
$$

$$
\begin{aligned}
\leq &-2\eta_\ell K\left(\mathbb{E}[F(w_g^{(r)})] - F(w_g^*)\right) - \frac{\eta_\ell \mu K}{2M}\mathbb{E}||w_g^{(r)} - w_g^*||^2 \\
&+ 16\eta_\ell^3 K^3 \beta^2 B^2 (4K+3)\mathbb{E}||1 - \psi_g^{(r)}||^2\left(\mathbb{E}[F(w_g^{(r)})] - F(w_g^*)\right) \\
&+ 8\eta_\ell^3 K^3 \beta(4K+3)\mathbb{E}||1 - \psi_g^{(r)}||^2 G^2 + 28\eta_\ell^3 K^2 \beta \sigma_\ell^2 \mathbb{E}||\psi_g^{(r)}||^2
\end{aligned}
\tag{49}
$$

**Bounding $T_2$**

$$
T_2 = \eta_\ell^2 \mathbb{E}\left[\left|\left|\frac{1}{nM}\sum_{m\in[M]} n_m \sum_{k=1}^{K}\nabla_{w_{g,m}^{(r,k-1)}} f_m(w_{p,m}^{(r,k-1)})\right|\right|^2\right]
\tag{50}
$$

$$
= \eta_\ell^2 \mathbb{E}\left[\left|\left|\frac{1}{nM}\sum_{m\in[M]} n_m \sum_{k=1}^{K}(\nabla_{w_{g,m}^{(r,k-1)}} f_m(w_{p,m}^{(r,k-1)}) - \nabla f_m(w_g^{(r)}) + \nabla f_m(w_g^{(r)}))\right|\right|^2\right]
\tag{51}
$$

$$
\begin{aligned}
\leq &\, 2\eta_\ell^2 \mathbb{E}\left[\left|\left|\frac{1}{nM}\sum_{m\in[M]} n_m \sum_{k=1}^{K}(\nabla_{w_{g,m}^{(r,k-1)}} f_m(w_{p,m}^{(r,k-1)}) - \nabla f_m(w_g^{(r)}))\right|\right|^2\right] \\
&+ 2\eta_\ell^2 \mathbb{E}\left[\left|\left|\frac{1}{nM}\sum_{m\in[M]} n_m \sum_{k=1}^{K}\nabla f_m(w_g^{(r)}))\right|\right|^2\right]
\end{aligned}
\tag{52}
$$

$$
\begin{aligned}
\leq &\, 2\eta_\ell^2 \beta^2 K \cdot \frac{1}{nM}\sum_{m\in[M]} n_m \sum_{k=1}^{K}\underbrace{\mathbb{E}\left[\left|\left|w_{p,m}^{(r,k-1)} - w_g^{(r)}\right|\right|^2\right]}_{\text{Lemma E.8}} \\
&+ 2\eta_\ell^2 K \cdot \frac{1}{nM}\sum_{m\in[M]} n_m \sum_{k=1}^{K}\mathbb{E}\left[\left|\left|\nabla f_m(w_g^{(r)}))\right|\right|^2\right]
\end{aligned}
\tag{53}
$$

$$
\begin{aligned}
\leq &\, 16\eta_\ell^4 K^4 \beta^3 B^2 (4K+3)\mathbb{E}||1 - \psi_g^{(r)}||^2\left(\mathbb{E}[F(w_g^{(r)})] - F(w_g^*)\right) \\
&+ 8\eta_\ell^4 K^4 \beta^3 (4K+3)\mathbb{E}||1 - \psi_g^{(r)}||^2 G^2 \\
&+ 56\eta_\ell^5 K^3 \beta^3 \sigma_\ell^2 \mathbb{E}||\psi_g^{(r)}||^2 + 2\eta_\ell^2 K\left(G^2 + 2\beta B^2 \mathbb{E}[F(w_g^{(r)})] - F(w_g^*)\right)
\end{aligned}
\tag{54}
$$

Plugging in $T_1$ and $T_2$ bounds,

$$
\begin{aligned}
\mathbb{E}\left[||w_g^{(r+1)} - w_g^*||^2\right] \leq &\, \mathbb{E}\left[||w_g^{(r)} - w_g^*||^2\right] - 2\eta_\ell K\left(\mathbb{E}[F(w_g^{(r)})] - F(w_g^*)\right) - \frac{\eta_\ell \mu K}{2M}\mathbb{E}||w_g^{(r)} - w_g^*||^2 \\
&+ 16\eta_\ell^3 K^3 \beta^2 B^2 (4K+3)(\eta_\ell \beta + 1)\mathbb{E}||1 - \psi_g^{(r)}||^2\left(\mathbb{E}[F(w_g^{(r)})] - F(w_g^*)\right) \\
&+ 8\eta_\ell^3 K^3 \beta(4K+3)(\eta_\ell \beta^2 + 1)\mathbb{E}||1 - \psi_g^{(r)}||^2 G^2 + 28\eta_\ell^3 K^2 \beta \sigma_\ell^2 \mathbb{E}||\psi_g^{(r)}||^2 \\
&+ 56\eta_\ell^5 K^3 \beta^3 \sigma_\ell^2 \mathbb{E}||\psi_g^{(r)}||^2 + 2\eta_\ell^2 K\left(G^2 + 2\beta B^2 \mathbb{E}[F(w_g^{(r)})] - F(w_g^*)\right)
\end{aligned}
\tag{55}
$$

Rearranging the terms, and replacing $\mathbb{E}||\psi_g^{(r)}||^2$ and $\mathbb{E}||1 - \psi_g^{(r)}||^2$ with $\mathbf{q}_0^2$ (probability of picking global route averaged over the instances sampled from the global distribution) and $\mathbf{q}_1^2$ respectively,

$$
\begin{aligned}
\mathbb{E}\left[||w_g^{(r+1)} - w_g^*||^2\right] \leq &\left(1 - \frac{\eta_\ell \mu K}{2M}\right)\mathbb{E}\left[||w_g^{(r)} - w_g^*||^2\right] \\
&- \left(2\eta_\ell K - 80\eta_\ell^3 K^4 \beta^2 B^2 (\eta_\ell \beta + 1)\mathbf{q}_1^2 - 4\eta_\ell^2 K\beta B^2\right)\left(\mathbb{E}\left[F(w_g^{(r)})\right] - F(w_g^*)\right) \\
&+ 40\eta_\ell^3 K^3 \beta(\eta_\ell \beta^2 + 1)\mathbf{q}_1^2 G^2 + 2\eta_\ell^2 K G^2 + 28\eta_\ell^3 K^2 \beta(2\eta_\ell^2 \beta^2 K + 1)\mathbf{q}_0^2 \sigma_\ell^2
\end{aligned}
\tag{56}
$$

Assuming $\frac{\eta_\ell K}{2} \geq 80\eta_\ell^3 K^4 \beta^2 B^2(\eta_\ell\beta + 1) \implies \eta_\ell \leq \frac{1}{4\sqrt{10}\beta B K^2}$ and $\frac{\eta_\ell K}{2} \geq 4\eta_\ell^2 K\beta B^2 \implies \eta_\ell \leq \frac{1}{8B^2\beta}$, we get

$$\mathbb{E}\left[||w_g^{(r+1)} - w_g^*||^2\right] \leq \left(1 - \frac{\eta_\ell\mu K}{2M}\right)\mathbb{E}\left[||w_g^{(r)} - w_g^*||^2\right] - \eta_\ell K(1 - \mathbf{q_1})^2\left(\mathbb{E}\left[F(w_g^{(r)})\right] - F(w_g^*)\right)$$
$$+ 40\eta_\ell^3 K^3\beta(\eta_\ell\beta^2 + 1)\mathbf{q_1^2}G^2 + 2\eta_\ell^2 KG^2 + 28\eta_\ell^3 K^2\beta(2\eta_\ell^2\beta^2 K + 1)\mathbf{q_0^2}\sigma_\ell^2 \tag{57}$$

Moving $\mathbb{E}\left[F(w_g^{(r)})\right] - F(w_g^*)$ to the left-hand side, and rest of the terms on right-hand side,

$$\eta_\ell K\mathbf{q_0^2}\left(\mathbb{E}\left[F(w_g^{(r)})\right] - F(w_g^*)\right) \leq \left(1 - \frac{\eta_\ell\mu K}{2M}\right)\mathbb{E}\left[||w_g^{(r)} - w_g^*||^2\right] - \mathbb{E}\left[||w_g^{(r+1)} - w_g^*||^2\right]$$
$$+ 40\eta_\ell^3 K^3\beta(\eta_\ell\beta^2 + 1)\mathbf{q_1^2}G^2 + 2\eta_\ell^2 KG^2 + 28\eta_\ell^3 K^2\beta(2\eta_\ell^2\beta^2 K + 1)\mathbf{q_0^2}\sigma_\ell^2 \tag{58}$$

$$\therefore \mathbb{E}\left[F(w_g^{(r)})\right] - F(w_g^*) \leq \frac{1}{\eta_\ell K\mathbf{q_0^2}}\left(1 - \frac{\eta_\ell\mu K}{2M}\right)\mathbb{E}\left[||w_g^{(r)} - w_g^*||^2\right] - \frac{1}{\eta_\ell K\mathbf{q_0^2}}\mathbb{E}\left[||w_g^{(r+1)} - w_g^*||^2\right]$$
$$+ 40\eta_\ell^2 K^2\beta(\eta_\ell\beta^2 + 1)\frac{\mathbf{q_1^2}}{\mathbf{q_0^2}}G^2 + \frac{2\eta_\ell G^2}{\mathbf{q_0^2}} + 28\eta_\ell^2 K\beta(2\eta_\ell^2\beta^2 K + 1)\sigma_\ell^2 \tag{59}$$

Unrolling the recursion over $R$ rounds and then using the linear convergence lemma (Lemma 1) for strong convex case from Scaffold [18],

$$\mathbb{E}\left[F(w_g^{(R)})\right] - F(w_g^*) \leq \frac{\mu}{\mathbf{q_0^2}K}\mathbb{E}||w_g^{(0)} - w_g^*||^2 \exp\left(-\frac{\eta_\ell\mu KR}{2M}\right) + \frac{2G^2}{\mathbf{q_0^2}\mu R}$$
$$+ \frac{40K^2\beta}{\mu^2 R^2}\left(\frac{\beta^2}{\mu R} + 1\right)\frac{\mathbf{q_1^2}}{\mathbf{q_0^2}}G^2 + \frac{28K\beta}{\mu^2 R^2}\left(\frac{2\beta^2 K}{\mu^2 R^2} + 1\right)\sigma_\ell^2 \tag{60}$$

Unrolling the recursion over $R$ rounds and then using the sublinear convergence lemma (Lemma 2) for general convex case from Scaffold [18],

$$\mathbb{E}\left[F(w_g^{(R)})\right] - F(w_g^*) \leq \frac{1}{\eta_\ell K\mathbf{q_0^2}(R + 1)}\mathbb{E}||w_g^{(0)} - w_g^*||^2 + \eta_\ell\left(\frac{2G^2}{\mathbf{q_0^2}}\right)^{1/2} +$$
$$+ \eta_\ell^2\left(40K^2\beta\frac{\mathbf{q_1^2}}{\mathbf{q_0^2}}G^2\right)^{1/2} + \eta_\ell^3\left(40K^2\beta^3\frac{\mathbf{q_1^2}}{\mathbf{q_0^2}}G^2\right)^{1/3} + \eta_\ell^2\left(28K\beta\sigma_\ell^2\right)^{1/3} + \eta_\ell^4\left(56K\beta^3\sigma_\ell^2\right)^{1/5} \tag{61}$$

$\square$

## E.4 Convergence Proof for the Global Model: Non-convex Case

We start with a non-convex version of Lemmas E.7 and E.8,

**Lemma E.10** (Local version of the global model progress). *If $m^{th}$ client's objective function $f_m$ satisfies Assumptions E.2, E.3, in Algorithm 2, the following is satisfied:*

$$\mathbb{E}||w_{g,m}^{(r,k)} - w_{g,m}^{(r,0)}||^2 \leq 4k^2\eta_\ell^2\mathbb{E}||\nabla f_m(w_g^{(r)})||^2 + 2k\eta_\ell^2\sigma_\ell^2 + 4k^2\eta_\ell^2\beta^2\sum_{i=1}^k\mathbb{E}||w_{p,m}^{(r,i-1)} - w_g^{(r)}||^2$$

*Proof.* We start by expanding $w_{g,m}^{(r,k)}$ in terms of its previous epoch iterate.

$$\mathbb{E}||w_{g,m}^{(r,k)} - w_{g,m}^{(r,0)}||^2 = \mathbb{E}||w_{g,m}^{(r,k-1)} - \eta_\ell\nabla_{w_{g,m}^{(r,k-1)}}f_m(w_{p,m}^{(r,k-1)}) - w_{g,m}^{(r,0)}||^2 \tag{62}$$

Using triangle inequality and separation of variance, we get,

$$\leq \left(1 + \frac{1}{k-1}\right)\mathbb{E}||w_{g,m}^{(r,k-1)} - w_{g,m}^{(r,0)}||^2 + k\eta_\ell^2\mathbb{E}||\nabla_{w_{g,m}^{(r,k-1)}}f_m(w_{p,m}^{(r,k-1)})||^2 + \eta_\ell^2\sigma_\ell^2 \tag{63}$$

$$\leq \left(1 + \frac{1}{k-1}\right)\mathbb{E}||w_{g,m}^{(r,k-1)} - w_{g,m}^{(r,0)}||^2 + \eta_\ell^2\sigma_\ell^2$$
$$+ k\eta_\ell^2\mathbb{E}||\nabla f_m(w_{p,m}^{(r,k-1)}) - \nabla f_m(w_{g,m}^{(r,0)}) + \nabla f_m(w_{g,m}^{(r,0)})||^2 \tag{64}$$

$$\tag{65}$$

$$\leq \left(1 + \frac{1}{k-1}\right) \mathbb{E}||w_{g,m}^{(r,k-1)} - w_{g,m}^{(r,0)}||^2 + \eta_\ell^2 \sigma_\ell^2 + 2k\eta_\ell^2 \mathbb{E}||\nabla f_m(w_g^{(r)})||^2$$

$$+ 2k\eta_\ell^2 \mathbb{E}||\nabla f_m(w_{p,m}^{(r,k-1)}) - \nabla f_m(w_g^{(r)})||^2 \tag{66}$$

$$\leq \left(1 + \frac{1}{k-1}\right) \mathbb{E}||w_{g,m}^{(r,k-1)} - w_{g,m}^{(r,0)}||^2 + \eta_\ell^2 \sigma_\ell^2 + 2k\eta_\ell^2 \mathbb{E}||\nabla f_m(w_g^{(r)})||^2$$

$$+ 2k\eta_\ell^2 \beta^2 \mathbb{E}||w_{p,m}^{(r,k-1)} - w_g^{(r)}||^2 \tag{67}$$

$$\tag{68}$$

Unrolling the recursion,

$$\mathbb{E}||w_{g,m}^{(r,k)} - w_{g,m}^{(r,0)}||^2 \leq \sum_{i=1}^{k} \left(2k\eta_\ell^2 \mathbb{E}||\nabla f_m(w_g^{(r)})||^2 + \eta_\ell^2 \sigma_\ell^2 + 2k\eta_\ell^2 \beta^2 \mathbb{E}||w_{p,m}^{(r,k-1)} - w_g^{(r)}||^2\right)$$

$$\cdot \left(1 + \frac{1}{k-1}\right)^i \tag{69}$$

$$\mathbb{E}||w_{g,m}^{(r,k)} - w_{g,m}^{(r,0)}||^2 \leq 2k \left(2k\eta_\ell^2 \mathbb{E}||\nabla f_m(w_g^{(r)})||^2 + \eta_\ell^2 \sigma_\ell^2 + 2k\eta_\ell^2 \beta^2 \sum_{i=1}^{k} \mathbb{E}||w_{p,m}^{(r,i-1)} - w_g^{(r)}||^2\right) \tag{70}$$

$$= 4k^2\eta_\ell^2 \mathbb{E}||\nabla f_m(w_g^{(r)})||^2 + 2k\eta_\ell^2 \sigma_\ell^2 + 4k^2\eta_\ell^2 \beta^2 \sum_{i=1}^{k} \mathbb{E}||w_{p,m}^{(r,i-1)} - w_g^{(r)}||^2 \tag{71}$$

$$\square$$

**Lemma E.11** (Deviation of the personalized model from the global model). *If $m^{th}$ client's objective function $f_m$ satisfies Assumptions E.2, E.3, and condition $\eta_\ell \leq \frac{1}{2\sqrt{2}\beta K}$ in Algorithm 2, the following is satisfied:*

$$\mathbb{E}||w_{p,m}^{(r,k)} - w_{g,m}^{(r,0)}||^2 \leq 20K^3\eta_\ell^2 \mathbb{E}||1 - \psi_{g,m}^{(r,k)}||^2 \mathbb{E}||\nabla f_m(w_g^{(r)})||^2 + 10K^2\eta_\ell^2 \sigma_\ell^2 \mathbb{E}||\psi_{g,m}^{(r,k)}||^2.$$

*Proof.*

$$\mathbb{E}||w_{p,m}^{(r,k)} - w_{g,m}^{(r,0)}||^2 = \mathbb{E}||\psi_{g,m}^{(r,k)} w_{g,m}^{(r,k)} + (1 - \psi_{g,m}^{(r,k)}) w_{\ell,m}^{(r,K)} - w_{g,m}^{(r,0)}||^2 \tag{72}$$

$$= \mathbb{E}||\psi_{g,m}^{(r,k)} (w_{g,m}^{(r,k)} - w_{\ell,m}^{(r,k)}) + (w_{\ell,m}^{(r,K)} - w_{g,m}^{(r,0)})||^2 \tag{73}$$

$$= \mathbb{E}||\psi_{g,m}^{(r,k)} (w_{g,m}^{(r,k)} - w_{g,m}^{(r,0)} + w_{g,m}^{(r,0)} - w_{\ell,m}^{(r,k)}) + (w_{\ell,m}^{(r,K)} - w_{g,m}^{(r,0)})||^2 \tag{74}$$

$$\leq 2\mathbb{E}||\psi_{g,m}^{(r,k)} (w_{g,m}^{(r,k)} - w_{g,m}^{(r,0)})||^2 + 2\mathbb{E}||(1 - \psi_{g,m}^{(r,k)})(w_{\ell,m}^{(r,K)} - w_{\ell,m}^{(r,0)})||^2 \tag{75}$$

Using lemmas E.6 and E.10,

$$\mathbb{E}||w_{p,m}^{(r,k)} - w_{g,m}^{(r,0)}||^2 \leq 2\mathbb{E}||1 - \psi_{g,m}^{(r,k+1)}||^2 \left(4K^2\eta_\ell^2 \mathbb{E}||\nabla f_m(w_g^{(r)})||^2 + 6K^2\eta_\ell^2 \mathbb{E}||\nabla f_m(w_g^{(r)})||^2\right)$$

$$+ 2\mathbb{E}||\psi_{g,m}^{(r,k+1)}||^2 \left(2K\eta_\ell^2 \sigma_\ell^2 + 3K\eta_\ell^2 \sigma_\ell^2 + 4K^2\eta_\ell^2 \beta^2 \sum_{i=1}^{k} \mathbb{E}||w_{p,m}^{(r,i-1)} - w_g^{(r)}||^2\right) \tag{76}$$

Assuming $8K^2\eta_\ell^2 \beta^2 \leq 1 \implies \eta \leq \frac{1}{2\sqrt{2}\beta K}$ and unrolling the recursion over $w_{p,m}^{(r,i-1)} - w_g^{(r)}$,

$$\leq \sum_{i=1}^{k} \left(20K^2\eta_\ell^2 \mathbb{E}||1 - \psi_{g,m}^{(r,k)}||^2 \mathbb{E}||\nabla f_m(w_g^{(r)})||^2 + 10K\eta_\ell^2 \sigma_\ell^2 \mathbb{E}||\psi_{g,m}^{(r,k)}||^2\right) \tag{77}$$

$$\leq 20K^3\eta_\ell^2 \mathbb{E}||1 - \psi_{g,m}^{(r,k)}||^2 \mathbb{E}||\nabla f_m(w_g^{(r)})||^2 + 10K^2\eta_\ell^2 \sigma_\ell^2 \mathbb{E}||\psi_{g,m}^{(r,k)}||^2 \tag{78}$$

$$\square$$

**Theorem E.12** (Convergence of the Global Model for Non-convex Case). *If each client's objective function $f_m$ satisfies Assumptions E.2, E.3, E.4, using the learning rate $\frac{1}{2\beta} \leq \eta_\ell \leq \min\left(\frac{1}{2\sqrt{5}\beta BK^2}, \frac{1}{\sqrt[3]{40K^4\beta^3 B^2}}\right)$ in Algorithm 2, then the following convergence holds:*

$$\frac{1}{R}\sum_{r=1}^{R} \mathbb{E}\left\|\nabla F(w_g^{(r)})\right\|^2 \leq \frac{2}{\eta_\ell \mathbf{q}_0^2 R}\left[\mathbb{E}\left[F(w_g^{(1)})\right] - \mathbb{E}\left[F(w_g^{(R+1)})\right]\right] + \frac{\eta_\ell \beta \sigma_\ell^2 K}{M\mathbf{q}_0^2}$$

$$+ 40\frac{\mathbf{q}_1^2}{\mathbf{q}_0^2} K^4 \beta^2 \eta_\ell G^2 \left(\frac{2\beta\eta_\ell^2 - \eta_\ell}{2}\right) + 20K^3 \beta^2 \eta_\ell \sigma_\ell^2 \left(\beta\eta_\ell^2 - \frac{\eta_\ell}{2}\right).$$

*Proof.* From the update rule stated in Equation 11, and $\beta$-smoothness of $f_m$, we have

$$F(w_g^{(r+1)}) \leq F(w_g^{(r)}) + \left\langle \nabla F(w_g^{(r)}), w_g^{(r+1)} - w_g^{(r)} \right\rangle + \frac{\beta}{2} ||w_g^{(r+1)} - w_g^{(r)}||^2 \tag{79}$$

Taking expectation on both sides,

$$\mathbb{E}\left[F(w_g^{(r+1)})\right] \leq \mathbb{E}\left[F(w_g^{(r)})\right] + \mathbb{E}\left[\left\langle \nabla F(w_g^{(r)}), w_g^{(r+1)} - w_g^{(r)} \right\rangle\right] + \frac{\beta}{2} ||w_g^{(r+1)} - w_g^{(r)}||^2 \tag{80}$$

Using Equation 10 for second and third terms, and using the fact that the expectation is with respect to the choice of $h_m$,

$$\leq \mathbb{E}\left[F(w_g^{(r)})\right] - \eta_\ell \left\langle \nabla F(w_g^{(r)}), \frac{1}{M} \sum_{m \in [M]} \alpha_m \sum_{k=1}^{K} \mathbb{E}\left[h_m(w_{p,m}^{(r,k-1)})\right] \right\rangle$$
$$+ \frac{\beta \eta_\ell^2}{2} \mathbb{E}\left\| \frac{1}{M} \sum_{m \in [M]} \alpha_m \sum_{k=1}^{K} h_m(w_{p,m}^{(r,k-1)}) \right\|^2, \tag{81}$$

where $\alpha_m = \frac{n_m}{n}$, which are the weights for weighted aggregation according to the sample count, as shown in Equation 11.

Separating mean and variance according to Assumption E.3,

$$\mathbb{E}\left[F(w_g^{(r+1)})\right] \leq \mathbb{E}\left[F(w_g^{(r)})\right] - \eta_\ell \left\langle \nabla F(w_g^{(r)}), \frac{1}{M} \sum_{m \in [M]} \alpha_m \sum_{k=1}^{K} \mathbb{E}\left[\nabla_{w_{g,m}^{(r,k-1)}} f_m(w_{p,m}^{(r,k-1)})\right] \right\rangle$$
$$+ \frac{\beta \eta_\ell^2}{2} \mathbb{E}\left[\left\| \frac{1}{M} \sum_{m \in [M]} \alpha_m \sum_{k=1}^{K} \nabla_{w_{g,m}^{(r,k-1)}} f_m(w_{p,m}^{(r,k-1)}) \right\|^2\right] + \frac{\eta_\ell^2 \beta \sigma_\ell^2 K}{2M} \tag{82}$$

Using $\langle a, b \rangle = -\frac{1}{2}||a-b||^2 + \frac{1}{2}||a||^2 + \frac{1}{2}||b||^2$,

$$\mathbb{E}\left[F(w_g^{(r+1)})\right] \leq \mathbb{E}\left[F(w_g^{(r)})\right] - \eta_\ell \left[-\frac{1}{2}\mathbb{E}\left\| \nabla F(w_g^{(r)}) - \frac{1}{M} \sum_{m \in [M]} \alpha_m \sum_{k=1}^{K} \nabla_{w_{g,m}^{(r,k-1)}} f_m(w_{p,m}^{(r,k-1)}) \right\|^2\right]$$
$$- \eta_\ell \left[\frac{1}{2}\mathbb{E}\left\| \nabla F(w_g^{(r)}) \right\|^2 + \frac{1}{2}\mathbb{E}\left\| \frac{1}{M} \sum_{m \in [M]} \alpha_m \sum_{k=1}^{K} \nabla_{w_{g,m}^{(r,k-1)}} f_m(w_{p,m}^{(r,k-1)}) \right\|^2\right]$$
$$+ \frac{\beta \eta_\ell^2}{2} \mathbb{E}\left[\left\| \frac{1}{M} \sum_{m \in [M]} \alpha_m \sum_{k=1}^{K} \nabla_{w_{g,m}^{(r,k-1)}} f_m(w_{p,m}^{(r,k-1)}) \right\|^2\right] + \frac{\eta_\ell^2 \beta \sigma_\ell^2 K}{2M} \tag{83}$$
$$\leq \mathbb{E}\left[F(w_g^{(r)})\right] - \frac{\eta_\ell}{2}\mathbb{E}\left\| \nabla F(w_g^{(r)}) \right\|^2$$
$$- \left(\frac{\eta_\ell}{2} - \frac{\beta \eta_\ell^2}{2}\right) \mathbb{E}\left\| \frac{1}{M} \sum_{m \in [M]} \alpha_m \sum_{k=1}^{K} \nabla_{w_{g,m}^{(r,k-1)}} f_m(w_{p,m}^{(r,k-1)}) \right\|^2$$
$$+ \frac{\eta_\ell}{2}\mathbb{E}\left\| \nabla F(w_g^{(r)}) - \frac{1}{M} \sum_{m \in [M]} \alpha_m \sum_{k=1}^{K} \nabla_{w_{g,m}^{(r,k-1)}} f_m(w_{p,m}^{(r,k-1)}) \right\|^2 + \frac{\eta_\ell^2 \beta \sigma_\ell^2 K}{2M} \tag{84}$$

$$\leq \mathbb{E}\left[F(w_g^{(r)})\right] - \frac{\eta_\ell}{2}\mathbb{E}\left\|\nabla F(w_g^{(r)})\right\|^2$$

$$- \left(\frac{\eta_\ell}{2} - \frac{\beta\eta_\ell^2}{2}\right)\mathbb{E}\left\|\nabla F(w_g^{(r)}) - \frac{1}{M}\sum_{m\in[M]}\alpha_m\sum_{k=1}^K \nabla_{w_{g,m}^{(r,k-1)}}f_m(w_{p,m}^{(r,k-1)}) - \nabla F(w_g^{(r)}))\right\|^2$$

$$+ \frac{\eta_\ell}{2}\mathbb{E}\left\|\nabla F(w_g^{(r)}) - \frac{1}{M}\sum_{m\in[M]}\alpha_m\sum_{k=1}^K \nabla_{w_{g,m}^{(r,k-1)}}f_m(w_{p,m}^{(r,k-1)})\right\|^2 + \frac{\eta_\ell^2\beta\sigma_\ell^2 K}{2M} \tag{85}$$

$$\leq \mathbb{E}\left[F(w_g^{(r)})\right] - \left(\frac{3\eta_\ell}{2} - \beta\eta_\ell^2\right)\mathbb{E}\left\|\nabla F(w_g^{(r)})\right\|^2 + \frac{\eta_\ell^2\beta\sigma_\ell^2 K}{2M}$$

$$- \left(\frac{\eta_\ell}{2} - \beta\eta_\ell^2\right)\mathbb{E}\left\|\nabla F(w_g^{(r)}) - \frac{1}{M}\sum_{m\in[M]}\alpha_m\sum_{k=1}^K \nabla_{w_{g,m}^{(r,k-1)}}f_m(w_{p,m}^{(r,k-1)})\right\|^2 \tag{86}$$

$$\leq \mathbb{E}\left[F(w_g^{(r)})\right] - \left(\frac{3\eta_\ell}{2} - \beta\eta_\ell^2\right)\mathbb{E}\left\|\nabla F(w_g^{(r)})\right\|^2 + \frac{\eta_\ell^2\beta\sigma_\ell^2 K}{2M}$$

$$- \left(\frac{\eta_\ell}{2} - \beta\eta_\ell^2\right)\beta^2 K \cdot \frac{1}{M}\sum_{m\in[M]}\alpha_m\sum_{k=1}^K \mathbb{E}\left\|w_g^{(r)} - w_{p,m}^{(r,k-1)}\right\|^2 \tag{87}$$

Using Lemma E.11,

$$\mathbb{E}\left[F(w_g^{(r+1)})\right] \leq \mathbb{E}\left[F(w_g^{(r)})\right] - \left(\frac{3\eta_\ell}{2} - \beta\eta_\ell^2\right)\mathbb{E}\left\|\nabla F(w_g^{(r)})\right\|^2 + \frac{\eta_\ell^2\beta\sigma_\ell^2 K}{2M}$$

$$- \left(\frac{\eta_\ell}{2} - \beta\eta_\ell^2\right)\beta^2 K \cdot \frac{1}{M}\sum_{m\in[M]}\alpha_m\sum_{k=1}^K \left(20K^3\eta_\ell^2\mathbb{E}||1 - \psi_{g,m}^{(r,k)}||^2\mathbb{E}||\nabla f_m(w_g^{(r)})||^2\right.$$

$$\left. + 10K^2\eta_\ell^2\sigma_\ell^2\mathbb{E}||\psi_{g,m}^{(r,k)}||^2\right) \tag{88}$$

Using Assumption E.4 for non-convex case, we get,

$$\mathbb{E}\left[F(w_g^{(r+1)})\right] \leq \mathbb{E}\left[F(w_g^{(r)})\right] - \left(\frac{3\eta_\ell}{2} - \beta\eta_\ell^2\right)\mathbb{E}\left\|\nabla F(w_g^{(r)})\right\|^2 + \frac{\eta_\ell^2\beta\sigma_\ell^2 K}{2M}$$

$$- \left(\frac{\eta_\ell}{2} - \beta\eta_\ell^2\right)20\beta^2 K^4\eta_\ell^2(G^2 + B^2\mathbb{E}||\nabla F(w_g^{(r)})||^2)\mathbb{E}||1 - \psi_g^{(r)}||^2$$

$$- \left(\frac{\eta_\ell}{2} - \beta\eta_\ell^2\right)\left(10\beta^2 K^3\eta_\ell^2\sigma_\ell^2\mathbb{E}||\psi_g^{(r)}||^2\right) \tag{89}$$

Rearranging the terms to put $\mathbb{E}\left\|\nabla F(w_g^{(r)})\right\|^2$ on left-hand side,

$$\left(\frac{3\eta_\ell}{2} - \beta\eta_\ell^2 - 20K^4\beta^2\eta_\ell^2 B^2\mathbf{q}_1^2\left(\frac{\eta_\ell}{2} - \beta\eta_\ell^2\right)\right)\mathbb{E}\left\|\nabla F(w_g^{(r)})\right\|^2 \leq \mathbb{E}\left[F(w_g^{(r)})\right]$$

$$- \mathbb{E}\left[F(w_g^{(r+1)})\right] - 20\mathbf{q}_1^2 K^4\beta^2\eta_\ell^2 G^2\left(\frac{\eta_\ell}{2} - \beta\eta_\ell^2\right)$$

$$- 10\mathbf{q}_0^2 K^3\beta^2\eta_\ell^2\sigma_\ell^2\left(\frac{\eta_\ell}{2} - \beta\eta_\ell^2\right) + \frac{\eta_\ell^2\beta\sigma_\ell^2 K}{2M} \tag{90}$$

Assuming $10K^4\beta^2\eta_\ell^3 B^2 \leq \frac{\eta_\ell}{2} \implies \eta_\ell \leq \frac{1}{2\sqrt{5}\beta BK^2}$ and $20K^4\beta^3\eta_\ell^4 B^2 \leq \frac{\eta_\ell}{2} \implies \eta_\ell \leq \frac{1}{\sqrt[3]{40K^4\beta^3 B^2}}$,

$$\left(\frac{\eta_\ell}{2}\right)\mathbf{q}_0^2\mathbb{E}\left\|\nabla F(w_g^{(r)})\right\|^2 \leq \mathbb{E}\left[F(w_g^{(r)})\right] - \mathbb{E}\left[F(w_g^{(r+1)})\right] + \frac{\eta_\ell^2\beta\sigma_\ell^2 K}{2M}$$

$$+ 20\mathbf{q}_1^2 K^4\beta^2\eta_\ell^2 G^2\left(\frac{2\beta\eta_\ell^2 - \eta_\ell}{2}\right) + 10\mathbf{q}_0^2 K^3\beta^2\eta_\ell^2\sigma_\ell^2\left(\beta\eta_\ell^2 - \frac{\eta_\ell}{2}\right) \tag{91}$$

$$\mathbb{E}\left\|\nabla F(w_g^{(r)})\right\|^2 \leq \frac{2}{\eta_\ell\mathbf{q}_0^2}\left[\mathbb{E}\left[F(w_g^{(r)})\right] - \mathbb{E}\left[F(w_g^{(r+1)})\right]\right] + \frac{\eta_\ell\beta\sigma_\ell^2 K}{M\mathbf{q}_0^2}$$

$$+ 40\frac{\mathbf{q}_1^2}{\mathbf{q}_0^2}K^4\beta^2\eta_\ell G^2\left(\frac{2\beta\eta_\ell^2 - \eta_\ell}{2}\right) + 20K^3\beta^2\eta_\ell\sigma_\ell^2\left(\beta\eta_\ell^2 - \frac{\eta_\ell}{2}\right) \tag{92}$$

Taking average over all the $R$ rounds,

$$\frac{1}{R}\sum_{r=1}^{R}\mathbb{E}\left\|\nabla F(w_g^{(r)})\right\|^2 \leq \frac{2}{\eta_\ell \mathbf{q}_0^2 R}\left[\mathbb{E}\left[F(w_g^{(1)})\right] - \mathbb{E}\left[F(w_g^{(R+1)})\right]\right] + \frac{\eta_\ell \beta \sigma_\ell^2 K}{M\mathbf{q}_0^2}$$

$$+ 40\frac{\mathbf{q}_1^2}{\mathbf{q}_0^2}K^4\beta^2\eta_\ell G^2\left(\frac{2\beta\eta_\ell^2 - \eta_\ell}{2}\right) + 20K^3\beta^2\eta_\ell\sigma_\ell^2\left(\beta\eta_\ell^2 - \frac{\eta_\ell}{2}\right) \quad (93)$$

$\square$

## E.5 Convergence Proof for the Personalized Model: Convex (Strong and General) Cases

**Lemma E.13** (Local progress of the personalized model). *If $m^{th}$ client's objective function $f_m$ satisfies Assumptions E.1, E.2, E.3, and E.4 and conditioning on $\eta_\ell \leq \frac{1}{\beta\sqrt{6K}}$ in Algorithm 2, the following are satisfied:*

$$\mathbb{E}||w_{p,m}^{(r,K)} - \tilde{w}_{p,m}^{(r,0)}||^2 \leq 18K^2\eta_\ell^2\mathbb{E}\big\|\nabla f_m(\tilde{w}_{p,m}^{(r,0)})\big\|^2 + 108K^4\eta_\ell^4\mathbb{E}||\nabla f_m(w_g^{(r)})||^2 + 126K^3\eta_\ell^4\sigma_\ell^2$$

$$+ 9K^2\eta_\ell^2\mathbb{E}\big\|\psi_{g,m}^{(r,K)}\big\|^2 + 144K^5\eta_\ell^4\mathbb{E}||\psi_{g,m}^{(r,K)}||^2\mathbb{E}||\nabla f_m(w_g^{(r)})||^2$$

*Proof.*

$$\mathbb{E}||w_{p,m}^{(r,K)} - \tilde{w}_{p,m}^{(r,0)}||^2 \leq \mathbb{E}||\psi_{g,m}^{(r,K+1)}w_{g,m}^{(r,K)} + (1 - \psi_{g,m}^{(r,K+1)})w_{\ell,m}^{(r,K)}$$

$$- \psi_{g,m}^{(r,0)}w_{g,m}^{(r,0)} - (1 - \psi_{g,m}^{(r,0)})w_{\ell,m}^{(r,K)}||^2 \quad (94)$$

$$= \mathbb{E}\Big\|\left(\psi_{g,m}^{(r,K)} - \eta_\ell\nabla_{\psi_{g,m}^{(r,K)}}f_m(\tilde{w}_{p,m}^{(r,K)})\right)\left(w_{g,m}^{(r,K-1)} - \eta_\ell\nabla_{w_{g,m}^{(r,K-1)}}f_m(w_{p,m}^{(r,K-1)})\right)$$

$$+ \left(1 - \psi_{g,m}^{(r,K)} + \eta_\ell\nabla_{\psi_{g,m}^{(r,K)}}f_m(\tilde{w}_{p,m}^{(r,K)})\right)w_{\ell,m}^{(r,K)} - \psi_{g,m}^{(r,0)}w_{g,m}^{(r,0)} - (1 - \psi_{g,m}^{(r,0)})w_{\ell,m}^{(r,K)}\Big\|^2 \quad (95)$$

$$= \mathbb{E}||\psi_{g,m}^{(r,K)}w_{g,m}^{(r,K-1)} - \psi_{g,m}^{(r,K)}\eta_\ell\nabla_{w_{g,m}^{(r,K-1)}}f_m(w_{p,m}^{(r,K-1)}) - w_{g,m}^{(r,K-1)}\eta_\ell\nabla_{\psi_{g,m}^{(r,K)}}f_m(\tilde{w}_{p,m}^{(r,K)})$$

$$+ \eta_\ell^2\nabla_{\psi_{g,m}^{(r,K)}}f_m(\tilde{w}_{p,m}^{(r,K)})\nabla_{w_{g,m}^{(r,K-1)}}f_m(w_{p,m}^{(r,K-1)}) + \left(1 - \psi_{g,m}^{(r,K)}\right)w_{\ell,m}^{(r,K)}$$

$$+ w_{\ell,m}^{(r,K)}\eta_\ell\nabla_{\psi_{g,m}^{(r,K)}}f_m(\tilde{w}_{p,m}^{(r,K)}) - \psi_{g,m}^{(r,0)}w_{g,m}^{(r,0)} - (1 - \psi_{g,m}^{(r,0)})w_{\ell,m}^{(r,K)}||^2 \quad (96)$$

Using the convexity of $f_m$,

$$\nabla_{w_{g,m}^{(r,k)}}f_m(w_{p,m}^{(r,k)}) = \nabla_{w_{g,m}^{(r,k)}}f_m(\psi_{g,m}^{(r,k+1)}w_{g,m}^{(r,k)} + (1 - \psi_{g,m}^{(r,k+1)})w_{\ell,m}^{(r,K)}) \quad (97)$$

$$\leq \psi_{g,m}^{(r,k+1)}\nabla f_m(w_{g,m}^{(r,k)}) \quad (98)$$

and

$$\nabla_{\psi_{g,m}^{(r,k)}}f_m(\tilde{w}_{p,m}^{(r,k)}) = \nabla_{\psi_{g,m}^{(r,k)}}f_m(\psi_{g,m}^{(r,k)}[w_{g,m}^{(r,k)} - w_{\ell,m}^{(r,K)}] - w_{\ell,m}^{(r,K)}) \quad (99)$$

$$\leq (w_{g,m}^{(r,k)} - w_{\ell,m}^{(r,K)})\nabla f_m(\psi_{g,m}^{(r,k)}) \quad (100)$$

we get,

$$\mathbb{E}||w_{p,m}^{(r,K)} - \tilde{w}_{p,m}^{(r,0)}||^2 \leq \mathbb{E}||\psi_{g,m}^{(r,K)}w_{g,m}^{(r,K-1)} + \left(1 - \psi_{g,m}^{(r,K)}\right)w_{\ell,m}^{(r,K)} - \psi_{g,m}^{(r,0)}w_{g,m}^{(r,0)} - (1 - \psi_{g,m}^{(r,0)})w_{\ell,m}^{(r,K)}$$

$$- \eta_\ell(\psi_{g,m}^{(r,K)})^2\nabla f_m(w_{g,m}^{(r,K-1)}) + \eta_\ell(w_{\ell,m}^{(r,K)} - w_{g,m}^{(r,K-1)})^2\nabla f_m(w_{g,m}^{(r,K)})||^2 \quad (101)$$

$$\leq \left(1 + \frac{1}{K-1}\right)\mathbb{E}||w_{p,m}^{(r,K-1)} - \tilde{w}_{p,m}^{(r,0)}||^2 + 3K\eta_\ell^2\mathbb{E}\big\|\nabla f_m(w_{p,m}^{(r,K-1)})\big\|^2$$

$$+ 3K\eta_\ell^2\mathbb{E}\big\|w_{\ell,m}^{(r,K)} - w_{g,m}^{(r,K-1)}\big\|^2 + 3K\eta_\ell^2\mathbb{E}\big\|\psi_{g,m}^{(r,K)}\big\|^2 \quad (102)$$

$$\leq \left(1 + \frac{1}{K-1}\right)\mathbb{E}||w_{p,m}^{(r,K-1)} - \tilde{w}_{p,m}^{(r,0)}||^2 + 3K\eta_\ell^2\mathbb{E}\big\|\nabla f_m(w_{p,m}^{(r,K-1)}) - \nabla f_m(\tilde{w}_{p,m}^{(r,0)}) + \nabla f_m(\tilde{w}_{p,m}^{(r,0)})\big\|^2$$

$$+ 6K\eta_\ell^2\mathbb{E}\big\|w_{\ell,m}^{(r,K)} - w_{g,m}^{(r,0)}\big\|^2 + 6K\eta_\ell^2\mathbb{E}\big\|w_{g,m}^{(r,0)} - w_{g,m}^{(r,K-1)}\big\|^2 + 3K\eta_\ell^2\mathbb{E}\big\|\psi_{g,m}^{(r,K)}\big\|^2 \quad (103)$$

Using Lemma E.6 and E.7, and smoothness property,

$$\leq \left(1 + \frac{1}{K-1} + 6K\eta_\ell^2\beta^2\right)\mathbb{E}||w_{p,m}^{(r,K-1)} - \tilde{w}_{p,m}^{(r,0)}||^2 + 6K\eta_\ell^2\mathbb{E}\big\|\nabla f_m(\tilde{w}_{p,m}^{(r,0)})\big\|^2$$

$$+ 6K\eta_\ell^2\left(6K^2\eta_\ell^2\mathbb{E}||\nabla f_m(w_{\ell,m}^{(r,0)})||^2 + 3K\eta_\ell^2\sigma_\ell^2\right) + 3K\eta_\ell^2\mathbb{E}\big\|\psi_{g,m}^{(r,K)}\big\|^2$$

$$+ 6K\eta_\ell^2\left(8K^3\eta_\ell^2\mathbb{E}||\psi_{g,m}^{(r,K)}||^2\mathbb{E}||\nabla f_m(w_g^{(r)})||^2 + 4K\eta_\ell^2\sigma_\ell^2\right) \quad (104)$$

$$\mathbb{E}||w_{p,m}^{(r,K)} - \tilde{w}_{p,m}^{(r,0)}||^2 \leq \sum_{i=1}^{K} \left( 6K\eta_\ell^2 \mathbb{E}||\nabla f_m(\tilde{w}_{p,m}^{(r,0)})||^2 + 36K^3\eta_\ell^4 \mathbb{E}||\nabla f_m(w_g^{(r)})||^2 + 42K^2\eta_\ell^4\sigma_\ell^2 \right.$$
$$\left. + 3K\eta_\ell^2 \mathbb{E}||\psi_{g,m}^{(r,K)}||^2 + 48K^4\eta_\ell^4 \mathbb{E}||\psi_{g,m}^{(r,K)}||^2 \mathbb{E}||\nabla f_m(w_g^{(r)})||^2 \right) \left( 1 + \frac{1}{K-1} + 6K\eta_\ell^2\beta^2 \right)^i \tag{105}$$

Assuming $6K\eta_\ell^2\beta^2 \leq 1 \implies \eta_\ell \leq \frac{1}{\beta\sqrt{6K}}$,

$$\mathbb{E}||w_{p,m}^{(r,K)} - \tilde{w}_{p,m}^{(r,0)}||^2 \leq 3K \left( 6K\eta_\ell^2 \mathbb{E}||\nabla f_m(\tilde{w}_{p,m}^{(r,0)})||^2 + 36K^3\eta_\ell^4 \mathbb{E}||\nabla f_m(w_g^{(r)})||^2 + 42K^2\eta_\ell^4\sigma_\ell^2 \right.$$
$$\left. + 3K\eta_\ell^2 \mathbb{E}||\psi_{g,m}^{(r,K)}||^2 + 48K^4\eta_\ell^4 \mathbb{E}||\psi_{g,m}^{(r,K)}||^2 \mathbb{E}||\nabla f_m(w_g^{(r)})||^2 \right) \tag{106}$$
$$= 18K^2\eta_\ell^2 \mathbb{E}||\nabla f_m(\tilde{w}_{p,m}^{(r,0)})||^2 + 108K^4\eta_\ell^4 \mathbb{E}||\nabla f_m(w_g^{(r)})||^2 + 126K^3\eta_\ell^4\sigma_\ell^2$$
$$+ 9K^2\eta_\ell^2 \mathbb{E}||\psi_{g,m}^{(r,K)}||^2 + 144K^5\eta_\ell^4 \mathbb{E}||\psi_{g,m}^{(r,K)}||^2 \mathbb{E}||\nabla f_m(w_g^{(r)})||^2 \tag{107}$$
$\square$

**Lemma E.14** (Deviation of local parameters from the aggregated global parameters). *If $m^{th}$ client's objective function $f_m$ satisfies Assumptions E.3, E.4, in Algorithm 2, the following is satisfied:*

$$\mathbb{E}||\tilde{w}_{p,m}^{(r+1,0)} - w_{p,m}^{(r,K)}||^2 \leq 18 \left( K\sigma_\ell^2\eta_\ell^2 + \left( \delta_m^\psi + \frac{\delta^\psi}{M} \right) K^2\eta_\ell^2 \right) \left( K\sigma_\ell^2\eta_\ell^2 + \left( \delta_m^{w_g} + \frac{\delta^{w_g}}{M} \right) K^2\eta_\ell^2 \right)$$
$$+ 6(1 + \eta_\ell^2 K^2\beta^4)\eta_\ell^2 K^2 \left( K\sigma_\ell^2\eta_\ell^2 + \left( \delta_m^\psi + \frac{\delta^\psi}{M} \right) K^2\eta_\ell^2 \right) \left( G^2 + B^2 \mathbb{E}||\nabla F(w_g^{(r)})||^2 \right)$$

*Proof.* Stating the aggregate rule from Algorithm 2, Lines 12 ,19 and 20,

$$\mathbb{E}||\tilde{w}_{p,m}^{(r+1,0)} - w_{p,m}^{(r,K)}||^2 = \mathbb{E}\left\| \frac{1}{M}\sum_{c\in[M]} \psi_{g,c}^{(r,K)} \frac{1}{M}\sum_{c\in[M]} w_{g,c}^{(r,K)} + \left( 1 - \frac{1}{M}\sum_{c\in[M]} \psi_{g,c}^{(r,K)} \right) w_{\ell,m}^{(r+1,K)} \right.$$
$$\left. - \psi_{g,m}^{(r,K)} w_{g,m}^{(r,K)} - (1 - \psi_{g,m}^{(r,K)}) w_{\ell,m}^{(r,K)} \right\|^2 \tag{108}$$
$$\leq 2\mathbb{E}\left\| \frac{1}{M}\sum_{c\in[M]} \psi_{g,c}^{(r,K)} \frac{1}{M}\sum_{c\in[M]} w_{g,c}^{(r,K)} - \psi_{g,m}^{(r,K)} w_{g,m}^{(r,K)} \right\|^2$$
$$+ 2\mathbb{E}\left\| \left( 1 - \frac{1}{M}\sum_{c\in[M]} \psi_{g,c}^{(r,K)} \right) w_{\ell,m}^{(r+1,K)} - (1 - \psi_{g,m}^{(r,K)}) w_{\ell,m}^{(r,K)} \right\|^2 \tag{109}$$
$$\leq 2\mathbb{E}\left\| \left( \frac{1}{M}\sum_{c\in[M]} \psi_{g,c}^{(r,K)} - \psi_{g,m}^{(r,K)} \right) \left( \frac{1}{M}\sum_{c\in[M]} w_{g,c}^{(r,K)} - w_{g,m}^{(r,K)} \right) \right\|^2$$
$$+ 2\mathbb{E}\left\| \left( \psi_{g,m}^{(r,K)} - \frac{1}{M}\sum_{c\in[M]} \psi_{g,c}^{(r,K)} \right) \left( w_{\ell,m}^{(r+1,K)} - w_{\ell,m}^{(r,K)} \right) \right\|^2 \tag{110}$$

Using Lemma 8 from [29] and Lemma E.17,

$$\mathbb{E}||\tilde{w}_{p,m}^{(r+1,0)} - w_{p,m}^{(r,K)}||^2 \leq 18 \left( K\sigma_\ell^2\eta_\ell^2 + \left( \delta_m^\psi + \frac{\delta^\psi}{M} \right) K^2\eta_\ell^2 \right) \left( K\sigma_\ell^2\eta_\ell^2 + \left( \delta_m^{w_g} + \frac{\delta^{w_g}}{M} \right) K^2\eta_\ell^2 \right)$$
$$+ 6(1 + \eta_\ell^2 K^2\beta^4)\eta_\ell^2 K^2 \left( K\sigma_\ell^2\eta_\ell^2 + \left( \delta_m^\psi + \frac{\delta^\psi}{M} \right) K^2\eta_\ell^2 \right)$$
$$\left( G^2 + B^2 \mathbb{E}||\nabla F(w_g^{(r)})||^2 \right) \tag{111}$$
$\square$

**Lemma E.15** (One epoch progress of the personalized model). *If $m^{th}$ client's objective function $f_m$ satisfies Assumptions E.1, E.2, E.3, and E.4 in Algorithm 2, the following are satisfied:*

$$\mathbb{E}||w_{p,m}^{(r,k+1)} - w_{p,m}^{(r,k)}||^2 \leq 3\eta_\ell^2 \mathbb{E}||\nabla f_m(w_{p,m}^{(r,k)})||^2 + 3\eta_\ell^2 \mathbb{E}||w_{\ell,m}^{(r,K)} - w_{g,m}^{(r,k)}||^2 + 3\eta_\ell^2 \mathbb{E}||\psi_{g,m}^{(r,k)}||^2$$

*and hence,*

$$\mathbb{E}||w_{p,m}^{(r,K)} - w_{p,m}^{(r,k)}||^2 \leq 6\beta\eta_\ell^2 \left( \mathbb{E}[f_m(w_{p,m}^{(r,K)})] - f(w_{p,m}^*) \right) + 3\eta_\ell^2 K \sum_{i=k}^{K} \mathbb{E}||\psi_{g,m}^{(r,i)}||^2$$
$$+ 36K^3\eta_\ell^4 \mathbb{E}||\nabla f_m(w_g^{(r)})||^2 + 40K^2\eta_\ell^4\sigma_\ell^2$$
$$+ 48K^4\eta_\ell^4 \mathbb{E}||\nabla f_m(w_g^{(r)})||^2 \sum_{i=k}^{K} \mathbb{E}||\psi_{g,m}^{(r,i)}||^2$$

*Proof.*

$$\mathbb{E}||w_{p,m}^{(r,k+1)} - w_{p,m}^{(r,k)}||^2 = \mathbb{E}||\psi_{g,m}^{(r,k+1)} w_{g,m}^{(r,k+1)}$$
$$+ (1 - \psi_{g,m}^{(r,k+1)})w_{\ell,m}^{(r,K)} - \psi_{g,m}^{(r,k)} w_{g,m}^{(r,k)} - (1 - \psi_{g,m}^{(r,k)})w_{\ell,m}^{(r,K)}||^2 \tag{112}$$

$$= \mathbb{E}\left\| \left( \psi_{g,m}^{(r,k)} - \eta_\ell \nabla_{\psi_{g,m}^{(r,k)}} f_m(\tilde{w}_{p,m}^{(r,k)}) \right) \left( w_{g,m}^{(r,k)} - \eta_\ell \nabla_{w_{g,m}^{(r,k)}} f_m(w_{p,m}^{(r,k)}) \right) \right.$$
$$+ \left( 1 - \psi_{g,m}^{(r,k)} + \eta_\ell \nabla_{\psi_{g,m}^{(r,k)}} f_m(\tilde{w}_{p,m}^{(r,k)}) \right) w_{\ell,m}^{(r,K)}$$
$$\left. - \psi_{g,m}^{(r,k)} w_{g,m}^{(r,k)} - (1 - \psi_{g,m}^{(r,k)})w_{\ell,m}^{(r,K)} \right\|^2 \tag{113}$$

$$= \mathbb{E}\left\| \psi_{g,m}^{(r,k)} w_{g,m}^{(r,k)} - \eta_\ell w_{g,m}^{(r,k)} \nabla_{\psi_{g,m}^{(r,k)}} f_m(\tilde{w}_{p,m}^{(r,k)}) - \eta_\ell \psi_{g,m}^{(r,k)} \nabla_{w_{g,m}^{(r,k)}} f_m(w_{p,m}^{(r,k)}) \right.$$
$$+ \eta_\ell^2 \nabla_{\psi_{g,m}^{(r,k)}} f_m(\tilde{w}_{p,m}^{(r,k)}) \nabla_{w_{g,m}^{(r,k)}} f_m(w_{p,m}^{(r,k)}) + (1 - \psi_{g,m}^{(r,k)})w_{\ell,m}^{(r,K)}$$
$$\left. + \eta_\ell w_{\ell,m}^{(r,K)} \nabla_{\psi_{g,m}^{(r,k)}} f_m(\tilde{w}_{p,m}^{(r,k)}) - \psi_{g,m}^{(r,k)} w_{g,m}^{(r,k)} - (1 - \psi_{g,m}^{(r,k)})w_{\ell,m}^{(r,K)} \right\|^2 \tag{114}$$

$$= \mathbb{E}\left\| \eta_\ell \left( w_{\ell,m}^{(r,K)} - w_{g,m}^{(r,k)} \right) \nabla_{\psi_{g,m}^{(r,k)}} f_m(\tilde{w}_{p,m}^{(r,k)}) \right.$$
$$\left. - \eta_\ell \left( \psi_{g,m}^{(r,k+1)} \right) \nabla_{w_{g,m}^{(r,k)}} f_m(w_{p,m}^{(r,k)}) \right\|^2 \tag{115}$$

Using,

$$\nabla_{w_{g,m}^{(r,k)}} f_m(w_{p,m}^{(r,k)}) = \nabla_{w_{g,m}^{(r,k)}} f_m(\psi_{g,m}^{(r,k+1)} w_{g,m}^{(r,k)} + (1 - \psi_{g,m}^{(r,k+1)})w_{\ell,m}^{(r,K)}) \tag{116}$$
$$\leq \psi_{g,m}^{(r,k+1)} \nabla f_m(w_{g,m}^{(r,k)}) \tag{117}$$

and,

$$\nabla_{\psi_{g,m}^{(r,k)}} f_m(\tilde{w}_{p,m}^{(r,k)}) = \nabla_{\psi_{g,m}^{(r,k)}} f_m(\psi_{g,m}^{(r,k)}[w_{g,m}^{(r,k)} - w_{\ell,m}^{(r,K)}] - w_{\ell,m}^{(r,K)}) \tag{118}$$
$$\leq [w_{g,m}^{(r,k)} - w_{\ell,m}^{(r,K)}] \nabla f_m(\psi_{g,m}^{(r,k)}), \tag{119}$$

we get,

$$\leq \eta_\ell^2 \mathbb{E}\left\| - \left( w_{\ell,m}^{(r,K)} - w_{g,m}^{(r,k)} \right)^2 \nabla f_m(\psi_{g,m}^{(r,k)}) - \left( \psi_{g,m}^{(r,k+1)} \right)^2 \nabla f_m(w_{g,m}^{(r,k)}) \right\|^2 \tag{120}$$

$$\leq \eta_\ell^2 \mathbb{E}\left\| \nabla f_m(w_{p,m}^{(r,k)}) + \left( w_{\ell,m}^{(r,K)} - w_{g,m}^{(r,k)} \right) + \psi_{g,m}^{(r,k+1)} \right\|^2 \tag{121}$$

$$\leq 3\eta_\ell^2 \mathbb{E}\left\| \nabla f_m(w_{p,m}^{(r,k)}) \right\|^2 + 3\eta_\ell^2 \mathbb{E}\left\| w_{\ell,m}^{(r,K)} - w_{g,m}^{(r,k)} \right\|^2 + 3\eta_\ell^2 \mathbb{E}\left\| \psi_{g,m}^{(r,k+1)} \right\|^2 \tag{122}$$

From Lemmas E.6 and E.7.

Summing over $i = k$ to $K$,

$$\mathbb{E}||w_{p,m}^{(r,K)} - w_{p,m}^{(r,k)}||^2 = \mathbb{E}||\sum_{i=k}^{K} w_{p,m}^{(r,k+1)} - w_{p,m}^{(r,k)}||^2 \tag{123}$$

$$\leq 3\eta_\ell^2 \sum_{i=k}^{K} \mathbb{E}||\nabla f_m(w_{p,m}^{(r,i)})||^2 + 3\eta_\ell^2 K \sum_{i=k}^{K} \mathbb{E}||\psi_{g,m}^{(r,i)}||^2$$

$$+ 6\eta_\ell^2 \sum_{i=k}^{K} \left( 6K^2\eta_\ell^2 \mathbb{E}||\nabla f_m(w_g^{(r)})||^2 + 3K\eta_\ell^2\sigma_\ell^2 \right)$$

$$+ 6\eta_\ell^2 \sum_{i=k}^{K} \left( 8K^3\eta_\ell^2 \mathbb{E}||\psi_{g,m}^{(r,i)}||^2 \mathbb{E}||\nabla f_m(w_g^{(r)})||^2 + 4K\eta_\ell^2\sigma_\ell^2 \right) \tag{124}$$

$$\leq 6\beta\eta_\ell^2 \left( \mathbb{E}[f_m(w_{p,m}^{(r,K)})] - f(w_{p,m}^*) \right) + 3\eta_\ell^2 K \sum_{i=k}^{K} \mathbb{E}||\psi_{g,m}^{(r,i)}||^2$$
$$+ 36K^3\eta_\ell^4 \mathbb{E}||\nabla f_m(w_g^{(r)})||^2 + 18K^2\eta_\ell^4\sigma_\ell^2$$
$$+ 48K^4\eta_\ell^4 \mathbb{E}||\nabla f_m(w_g^{(r)})||^2 \sum_{i=k}^{K} \mathbb{E}||\psi_{g,m}^{(r,i)}||^2 + 24K^2\eta_\ell^4\sigma_\ell^2 \tag{125}$$

$$\therefore \mathbb{E}||w_{p,m}^{(r,K)} - w_{p,m}^{(r,k)}||^2 \leq 6\beta\eta_\ell^2 \left( \mathbb{E}[f_m(w_{p,m}^{(r,K)})] - f(w_{p,m}^*) \right) + 3\eta_\ell^2 K \sum_{i=k}^{K} \mathbb{E}||\psi_{g,m}^{(r,i)}||^2$$
$$+ 36K^3\eta_\ell^4 \mathbb{E}||\nabla f_m(w_g^{(r)})||^2 + 40K^2\eta_\ell^4\sigma_\ell^2$$
$$+ 48K^4\eta_\ell^4 \mathbb{E}||\nabla f_m(w_g^{(r)})||^2 \sum_{i=k}^{K} \mathbb{E}||\psi_{g,m}^{(r,i)}||^2 \tag{126}$$

$\square$

**Theorem E.16** (Convergence of the Personalized Model for Convex (Strong and General) Cases)**.** *If each client's objective function $f_m$ satisfies Assumptions E.2, E.3, E.4, using the learning rate $\frac{1}{\mu R} \leq \eta_\ell \leq \frac{1}{K\beta^2}$ in Algorithm 2, then the following convergence holds:*
*(Strong Convex Case)*

$$\mathbb{E}\left[ f_m(w_{p,m}^{(R,0)}) \right] - f_m(w_{p,m}^*) \leq \frac{36\mu^2}{RK^3} \mathbb{E}||w_{p,m}^{(1,K)} - w_{p,m}^*||^2 \exp\left( \frac{1}{K-1} - \eta_\ell\mu KR \right)$$
$$+ 12K^2\eta_\ell^2\delta_m^{w_g} + 12K^2\eta_\ell^2 \frac{1}{R}\sum_{r=1}^{R} \mathbb{E}||\nabla F(w_g^{(r+1)})||^2 + 4K\eta_\ell^2\sigma_\ell^2 + \frac{\mathbf{q}_0^2}{2} + 16K^3\eta_\ell^2\mathbf{q}_0^2\delta_m^{w_g}$$
$$+ 16K^3\eta_\ell^2\mathbf{q}_0^2 \frac{1}{R}\sum_{r=1}^{R} \mathbb{E}||\nabla F(w_g^{(r+1)})||^2 + \frac{K^2\eta_\ell^2}{2}\left( \frac{\sigma_\ell^2}{K} + \left( \delta_m^\psi + \frac{\delta^\psi}{M} \right) \right)\left( \frac{\sigma_\ell^2}{K} + \left( \delta_m^{w_g} + \frac{\delta^{w_g}}{M} \right) \right)$$
$$+ \frac{1+\eta_\ell^2 K^2\beta^4}{6}\left( K\sigma_\ell^2\eta_\ell^2 + \left( \delta_m^\psi + \frac{\delta^\psi}{M} \right)K^2\eta_\ell^2 \right)(\frac{2}{R}\sum_{r=1}^{R} \mathbb{E}||\nabla F(w_g^{(r)})||^2 + 2\delta_m^{w_g})$$

*(General Convex Case)*

$$\mathbb{E}\left[ f_m(w_{p,m}^{(R,0)}) \right] - f_m(w_{p,m}^*) \leq \frac{1}{36\eta_\ell^2 K^2 R}\left( 1 + \frac{1}{K-1} \right)\mathbb{E}||w_{p,m}^{(1,K)} - w_{p,m}^*||^2$$
$$+ \eta_\ell^2(12K^2 \frac{1}{R}\sum_{r=1}^{R} \mathbb{E}||\nabla F(w_g^{(r)})||^2)^{1/3} + \eta_\ell^2(12K^2\delta_m^{w_g})^{1/3} + \eta_\ell^2(4K\sigma_\ell^2)^{1/3}$$
$$+ \eta_\ell^2(16K^3\mathbf{q}_0^2 \frac{1}{R}\sum_{r=1}^{R} \mathbb{E}||\nabla F(w_g^{(r+1)})||^2)^{1/3} + \frac{\mathbf{q}_0^2}{2} + \eta_\ell^2(16K^3\mathbf{q}_0^2\delta_m^{w_g})^{1/3}$$
$$+ \eta_\ell^2\left( \frac{K^2}{2}\left( \frac{\sigma_\ell^2}{K} + \left( \delta_m^\psi + \frac{\delta^\psi}{M} \right) \right)\left( \frac{\sigma_\ell^2}{K} + \left( \delta_m^{w_g} + \frac{\delta^{w_g}}{M} \right) \right) \right)^{1/3}$$
$$+ \eta_\ell^2\left( \frac{K^2}{3}\left( \frac{\sigma_\ell^2}{K} + \left( \delta_m^\psi + \frac{\delta^\psi}{M} \right) \right)(\frac{2}{R}\sum_{r=1}^{R} \mathbb{E}||\nabla F(w_g^{(r)})||^2 + 2\delta_m^{w_g}) \right)^{1/3}$$

*where $\frac{1}{R}\sum_{r=1}^{R} \mathbb{E}||\nabla F(w_g^{(r)})||^2$ is bounded as shown in Theorem E.12.*

*Proof.* We restate the update rules of the personalized model in Algorithm 2,

1. For all samples $x_m$, define $\tilde{w}_{p,m}^{(r,k)}(x_m) \leftarrow \psi_{g,m}^{(r,k)}(x_m)w_{g,m}^{(r,k)}(x_m) + (1 - \psi_{g,m}^{(r,k)}(x_m))w_{\ell,m}^{(r,K)}(x_m)$

2. Train policy parameters $\psi_{g,m}^{(r,k+1)} \leftarrow \psi_{g,m}^{(r,k)} - \eta_\ell \nabla_{\psi_{g,m}^{(r,k)}} f_m(\tilde{w}_{p,m}^{(r,k)}(x_m), y_m)$

3. For all samples $x_m$, define
   $w_{p,m}^{(r,k)}(x_m) \leftarrow \psi_{g,m}^{(r,k+1)}(x_m)w_{g,m}^{(r,k)}(x_m) + (1 - \psi_{g,m}^{(r,k+1)}(x_m))w_{\ell,m}^{(r,K)}(x_m)$

4. Train global parameters $w_{g,m}^{(r,k)} \leftarrow w_{g,m}^{(r,k-1)} - \eta_\ell \nabla_{w_{g,m}^{(r,k-1)}} f_m(w_{p,m}^{(r,k)}(x_m), y_m)$

$$\mathbb{E}||w_{p,m}^{(r+1,K)} - w_{p,m}^*||^2 = \mathbb{E}||w_{p,m}^{(r+1,K)} - \tilde{w}_{p,m}^{(r+1,0)} + \tilde{w}_{p,m}^{(r+1,0)} - w_{p,m}^{(r,K)} + w_{p,m}^{(r,K)} - w_{p,m}^*||^2 \quad (127)$$

$$\leq 2K \underbrace{\mathbb{E}||w_{p,m}^{(r+1,K)} - \tilde{w}_{p,m}^{(r+1,0)}||^2}_{\text{Lemma E.13}} + 2K \underbrace{\mathbb{E}||\tilde{w}_{p,m}^{(r+1,0)} - w_{p,m}^{(r,K)}||^2}_{\text{Lemma E.14}}$$

$$+ \left(1 + \frac{1}{K-1}\right)\mathbb{E}||w_{p,m}^{(r,K)} - w_{p,m}^*||^2 \quad (128)$$

And using Assumption E.4,

$$\leq 36K^2\eta_\ell^2 \left[f_m(w_{p,m}^*) - \mathbb{E}\left[f_m(w_{p,m}^{(r+1,0)})\right]\right] + 216K^4\eta_\ell^4\mathbb{E}||\nabla f_m(w_g^{(r+1)})||^2 + 126K^3\eta_\ell^4\sigma_\ell^2$$

$$+ 18K^2\eta_\ell^2\mathbb{E}||\psi_{g,m}^{(r+1,K)}||^2 + 288K^5\eta_\ell^4\mathbb{E}||\psi_{g,m}^{(r+1,K)}||^2\mathbb{E}||\nabla f_m(w_g^{(r+1)})||^2$$

$$+ 18\left(K\sigma_\ell^2\eta_\ell^2 + \left(\delta_m^\psi + \frac{\delta^\psi}{M}\right)K^2\eta_\ell^2\right)\left(K\sigma_\ell^2\eta_\ell^2 + \left(\delta_m^{w_g} + \frac{\delta^{w_g}}{M}\right)K^2\eta_\ell^2\right)$$

$$+ 6(1 + \eta_\ell^2 K^2\beta^4)\eta_\ell^2 K^2\left(K\sigma_\ell^2\eta_\ell^2 + \left(\delta_m^\psi + \frac{\delta^\psi}{M}\right)K^2\eta_\ell^2\right)\mathbb{E}||\nabla f_m(w_g^{(r)})||^2$$

$$+ \left(1 + \frac{1}{K-1} - \mu\eta_\ell\right)\mathbb{E}||w_{p,m}^{(r,K)} - w_{p,m}^*||^2 \quad (129)$$

Rearranging the terms,

$$36K^2\eta_\ell^2\left[\mathbb{E}\left[f_m(w_{p,m}^{(r+1,0)})\right] - f_m(w_{p,m}^*)\right] \leq \left(1 + \frac{1}{K-1} - \mu\eta_\ell\right)\mathbb{E}||w_{p,m}^{(r,K)}$$

$$+ 216K^4\eta_\ell^4\mathbb{E}||\nabla f_m(w_g^{(r+1)})||^2 + 126K^3\eta_\ell^4\sigma_\ell^2 + 18K^2\eta_\ell^2\mathbb{E}||\psi_{g,m}^{(r+1,K)}||^2$$

$$- w_{p,m}^*||^2 - \mathbb{E}||w_{p,m}^{(r,K+1)} - w_{p,m}^*||^2 + 288K^5\eta_\ell^4\mathbb{E}||\psi_{g,m}^{(r+1,K)}||^2\mathbb{E}||\nabla f_m(w_g^{(r+1)})||^2$$

$$+ 18\left(K\sigma_\ell^2\eta_\ell^2 + \left(\delta_m^\psi + \frac{\delta^\psi}{M}\right)K^2\eta_\ell^2\right)\left(K\sigma_\ell^2\eta_\ell^2 + \left(\delta_m^{w_g} + \frac{\delta^{w_g}}{M}\right)K^2\eta_\ell^2\right)$$

$$+ 6(1 + \eta_\ell^2 K^2\beta^4)\eta_\ell^2 K^2\left(K\sigma_\ell^2\eta_\ell^2 + \left(\delta_m^\psi + \frac{\delta^\psi}{M}\right)K^2\eta_\ell^2\right)\mathbb{E}||\nabla f_m(w_g^{(r)})||^2 \quad (130)$$

$$\therefore \mathbb{E}\left[f_m(w_{p,m}^{(r+1,0)})\right] - f_m(w_{p,m}^*) \leq \frac{1}{36\eta_\ell^2 K^2}\left(1 + \frac{1}{K-1} - \mu\eta_\ell\right)\mathbb{E}||w_{p,m}^{(r,K)} - w_{p,m}^*||^2$$

$$- \frac{1}{36\eta_\ell^2 K^2}\mathbb{E}||w_{p,m}^{(r,K+1)} - w_{p,m}^*||^2 + 6K^2\eta_\ell^2\mathbb{E}||\nabla f_m(w_g^{(r+1)})||^2 + 4K\eta_\ell^2\sigma_\ell^2 + \frac{1}{2}\mathbb{E}||\psi_{g,m}^{(r+1,K)}||^2$$

$$+ 8K^3\eta_\ell^2\mathbb{E}||\psi_{g,m}^{(r+1,K)}||^2\mathbb{E}||\nabla f_m(w_g^{(r+1)})||^2 + \frac{K^2\eta_\ell^2}{2}\left(\frac{\sigma_\ell^2}{K} + \left(\delta_m^\psi + \frac{\delta^\psi}{M}\right)\right)\left(\frac{\sigma_\ell^2}{K} + \left(\delta_m^{w_g} + \frac{\delta^{w_g}}{M}\right)\right)$$

$$+ \frac{1 + \eta_\ell^2 K^2\beta^4}{6}\left(K\sigma_\ell^2\eta_\ell^2 + \left(\delta_m^\psi + \frac{\delta^\psi}{M}\right)K^2\eta_\ell^2\right)\mathbb{E}||\nabla f_m(w_g^{(r)})||^2 \quad (131)$$

For strong convex ($\mu > 0$) case, using the linear convergence rate lemma from [18] (Lemma 1) and Definition E.5,

$$\mathbb{E}\left[f_m(w_{p,m}^{(R,0)})\right] - f_m(w_{p,m}^*) \leq \frac{36\mu^2}{RK^3}\mathbb{E}||w_{p,m}^{(1,K)} - w_{p,m}^*||^2 \exp\left(\frac{1}{K-1} - \eta_\ell\mu KR\right)$$

$$+ 12K^2\eta_\ell^2\delta_m^{w_g} + 12K^2\eta_\ell^2\frac{1}{R}\sum_{r=1}^{R}\mathbb{E}||\nabla F(w_g^{(r+1)})||^2 + 4K\eta_\ell^2\sigma_\ell^2 + \frac{\mathbf{q}_0^2}{2} + 16K^3\eta_\ell^2\mathbf{q}_0^2\delta_m^{w_g}$$

$$+ 16K^3\eta_\ell^2\mathbf{q}_0^2\frac{1}{R}\sum_{r=1}^{R}\mathbb{E}||\nabla F(w_g^{(r+1)})||^2 + \frac{K^2\eta_\ell^2}{2}\left(\frac{\sigma_\ell^2}{K} + \left(\delta_m^\psi + \frac{\delta^\psi}{M}\right)\right)\left(\frac{\sigma_\ell^2}{K} + \left(\delta_m^{w_g} + \frac{\delta^{w_g}}{M}\right)\right)$$

$$+ \frac{1 + \eta_\ell^2 K^2\beta^4}{6}\left(K\sigma_\ell^2\eta_\ell^2 + \left(\delta_m^\psi + \frac{\delta^\psi}{M}\right)K^2\eta_\ell^2\right)\left(\frac{2}{R}\sum_{r=1}^{R}\mathbb{E}||\nabla F(w_g^{(r)})||^2 + 2\delta_m^{w_g}\right) \quad (132)$$

For general convex ($\mu = 0$) case, using the sublinear convergence rate lemma from [18] (Lemma 2), and conditioning on $\eta_\ell^2 K^2 \beta^4 \leq 1 \implies \eta_\ell \leq \frac{1}{K\beta^2}$,

$$\mathbb{E}\left[f_m(w_{p,m}^{(R,0)})\right] - f_m(w_{p,m}^*) \leq \frac{1}{36\eta_\ell^2 K^2 R}\left(1 + \frac{1}{K-1}\right)\mathbb{E}||w_{p,m}^{(1,K)} - w_{p,m}^*||^2$$

$$+ \eta_\ell^2(12K^2\frac{1}{R}\sum_{r=1}^R\mathbb{E}||\nabla F(w_g^{(r)})||^2)^{1/3} + \eta_\ell^2(12K^2\delta_m^{w_g})^{1/3} + \eta_\ell^2(4K\sigma_\ell^2)^{1/3} + \frac{\mathbf{q}_0^2}{2}$$

$$+ \eta_\ell^2(16K^3\mathbf{q}_0^2\delta_m^{w_g})^{1/3} + \eta_\ell^2(16K^3\mathbf{q}_0^2\frac{1}{R}\sum_{r=1}^R\mathbb{E}||\nabla F(w_g^{(r+1)})||^2)^{1/3}$$

$$+ \eta_\ell^2\left(\frac{K^2}{2}\left(\frac{\sigma_\ell^2}{K} + \left(\delta_m^\psi + \frac{\delta^\psi}{M}\right)\right)\left(\frac{\sigma_\ell^2}{K} + \left(\delta_m^{w_g} + \frac{\delta^{w_g}}{M}\right)\right)\right)^{1/3}tzo$$

$$+ \eta_\ell^2\left(\frac{K^2}{3}\left(\frac{\sigma_\ell^2}{K} + \left(\delta_m^\psi + \frac{\delta^\psi}{M}\right)\right)(\frac{2}{R}\sum_{r=1}^R\mathbb{E}||\nabla F(w_g^{(r)})||^2 + 2\delta_m^{w_g})\right)^{1/3} \tag{133}$$

$\square$

## E.6   Convergence Proof for the Personalized Model: Non-convex Case

**Lemma E.17** (One round progress of the local model). *If $m^{th}$ client's objective function $f_m$ satisfies Assumptions E.3, E.4, in Algorithm 2, the following is satisfied:*

$$\mathbb{E}||w_{\ell,m}^{(r+1,K)} - w_{\ell,m}^{(r,K)}||^2 \leq (1 - 2\eta_\ell K\beta^2 + \eta_\ell^2 K^2\beta^4)\eta_\ell^2 K^2\left(G^2 + B^2\mathbb{E}||\nabla F(w_g^{(r)})||^2\right)$$

*Proof.*

$$\mathbb{E}||w_{\ell,m}^{(r+1,K)} - w_{\ell,m}^{(r,K)}||^2 = \mathbb{E}||w_g^{(r+1)} - \eta_\ell\sum_{k=1}^K\nabla f_m(w_g^{(r+1)}) - w_g^{(r)} + \eta_\ell\sum_{k=1}^K\nabla f_m(w_g^{(r)})||^2 \tag{134}$$

$$= \mathbb{E}||w_g^{(r+1)} - w_g^{(r)} - \eta_\ell\sum_{k=1}^K\left[\nabla f_m(w_g^{(r+1)}) - \nabla f_m(w_g^{(r)})\right]||^2 \tag{135}$$

$$\leq \mathbb{E}||w_g^{(r+1)} - w_g^{(r)} - \eta_\ell K\beta^2\left(w_g^{(r+1)} - w_g^{(r)}\right)||^2 \tag{136}$$

$$= \mathbb{E}||(1 - \eta_\ell K\beta^2)\left(w_g^{(r+1)} - w_g^{(r)}\right)||^2 \tag{137}$$

$$\leq (1 - \eta_\ell K\beta^2)^2\mathbb{E}\left\|\frac{1}{M}\sum_{c\in[M]}w_{g,m}^{(r,K)} - w_g^{(r)}\right\|^2 \tag{138}$$

$$= (1 - \eta_\ell K\beta^2)^2\mathbb{E}\left\|\frac{1}{M}\sum_{c\in[M]}(w_g^{(r)} - \eta_\ell\sum_{k=1}^K\nabla f_m(w_g^{(r)})) - w_g^{(r)}\right\|^2 \tag{139}$$

$$= (1 - \eta_\ell K\beta^2)^2\mathbb{E}\left\|-\frac{\eta_\ell}{M}\sum_{c\in[M]}\sum_{k=1}^K\nabla f_m(w_g^{(r)})\right\|^2 \tag{140}$$

$$\leq (1 - \eta_\ell K\beta^2)^2\eta_\ell^2\mathbb{E}\left\|\frac{K}{M}\sum_{c\in[M]}\nabla f_m(w_g^{(r)})\right\|^2 \tag{141}$$

$$\leq (1 - \eta_\ell K\beta^2)^2\eta_\ell^2 K^2\left(G^2 + B^2\mathbb{E}||\nabla F(w_g^{(r)})||^2\right) \tag{142}$$

$$= (1 - 2\eta_\ell K\beta^2 + \eta_\ell^2 K^2\beta^4)\eta_\ell^2 K^2\left(G^2 + B^2\mathbb{E}||\nabla F(w_g^{(r)})||^2\right) \tag{143}$$

The last inequality follows from Assumption E.4. $\square$

We proceed with a lemma which binds the deviation of the personalized model $w_p$ of an arbitrary client $m$ over one round, i.e., $w_{p,m}^{(r+1)}$ and $w_{p,m}^{(r)}$, for non-convex case.

**Lemma E.18** (Local progress of personalized model). *If $m^{th}$ client's objective function $f_m$ satisfies Assumptions E.3, E.4, and $\eta_\ell \leq \frac{1}{K\sqrt{12\beta(K-1)}}$, in Algorithm 2, the following is satisfied:*

$$\mathbb{E}||w_{p,m}^{(r,k+1)} - w_{p,m}^{(r,0)}||^2 \leq 18K^5\eta_\ell^2\mathbb{E}||\nabla f_m(w_g^{(r)})||^2 + 9K^4\eta_\ell^2\sigma_\ell^2 + 36K^3\eta_\ell^2\sigma_\ell^2\mathbb{E}||1 - \psi_{g,m}^{(r,k)}||^2$$
$$+ 24K^4\eta_\ell^2\mathbb{E}||\nabla f_m(w_g^{(r)})||^2\mathbb{E}||\psi_{g,m}^{(r,k)}||^2$$

*Proof.* We start with using the update rule stated for the personalized model at the beginning of Theorem E.16,

$$\mathbb{E}||w_{p,m}^{(r,k+1)} - w_{p,m}^{(r,0)}||^2 = \mathbb{E}||\psi_{g,m}^{(r,k+1)}w_{g,m}^{(r,k+1)} + (1 - \psi_{g,m}^{(r,k+1)})w_{\ell,m}^{(r,K)} - w_{p,m}^{(r,0)}||^2 \tag{144}$$

Expanding by one iterate,

$$= \mathbb{E}||\left(\psi_{g,m}^{(r,k)} - \eta_\ell\nabla_{\psi_{g,m}^{(r,k)}}f_m(w_{p,m}^{(r,k)})\right)\left(w_{g,m}^{(r,k)} - \eta_\ell\nabla_{w_{g,m}^{(r,k)}}f_m(w_{p,m}^{(r,k)})\right)$$
$$+ \left(1 - \psi_{g,m}^{(r,k)} + \eta_\ell\nabla_{\psi_{g,m}^{(r,k)}}f_m(w_{p,m}^{(r,k)})\right)w_{\ell,m}^{(r,K)} - w_{p,m}^{(r,0)}||^2 \tag{145}$$

$$= \mathbb{E}||\psi_{g,m}^{(r,k)}w_{g,m}^{(r,k)} - w_{g,m}^{(r,k)}\eta_\ell\nabla_{\psi_{g,m}^{(r,k)}}f_m(w_{p,m}^{(r,k)}) - \psi_{g,m}^{(r,k)}\eta_\ell\nabla_{w_{g,m}^{(r,k)}}f_m(w_{p,m}^{(r,k)})$$
$$+ \eta_\ell^2\nabla_{\psi_{g,m}^{(r,k)}}f_m(w_{p,m}^{(r,k)})\nabla_{w_{g,m}^{(r,k)}}f_m(w_{p,m}^{(r,k)}) + (1 - \psi_{g,m}^{(r,k)})w_{\ell,m}^{(r,K)}$$
$$+ w_{\ell,m}^{(r,K)}\eta_\ell\nabla_{\psi_{g,m}^{(r,k)}}f_m(w_{p,m}^{(r,k)}) - w_{p,m}^{(r,0)}||^2 \tag{146}$$

$$= \mathbb{E}||w_{p,m}^{(r,k)} - w_{p,m}^{(r,0)} + (w_{\ell,m}^{(r,K)} - w_{g,m}^{(r,k)})\eta_\ell\nabla_{\psi_{g,m}^{(r,k)}}f_m(w_{p,m}^{(r,k)})$$
$$\left(-\psi_{g,m}^{(r,k)} + \eta_\ell\nabla_{\psi_{g,m}^{(r,k)}}f_m(w_{p,m}^{(r,k)})\right)\eta_\ell\nabla_{w_{g,m}^{(r,k)}}f_m(w_{p,m}^{(r,k)})||^2 \tag{147}$$

$$\leq 3\mathbb{E}||w_{p,m}^{(r,k)} - w_{p,m}^{(r,0)}||^2 + 3\mathbb{E}||w_{\ell,m}^{(r,K)} - w_{g,m}^{(r,k)}||^2 + 3\eta_\ell^2\mathbb{E}||\nabla_{w_{g,m}^{(r,k)}}f_m(w_{p,m}^{(r,k)})||^2 \tag{148}$$

The inequality was derived from the fact that $\mathbb{E}|| - \eta_\ell\nabla_{\psi_{g,m}^{(r,k)}}f_m(w_{p,m}^{(r,k)})||^2 = \mathbb{E}||\psi_{g,m}^{(r,k+1)} - \psi_{g,m}^{(r,k)}||^2 \leq 1$. Unrolling the recursion across $r \in [R]$, then using Lemmas E.6 and E.10 and Assumption E.4,

$$\mathbb{E}||w_{p,m}^{(r,K)} - w_{p,m}^{(r,0)}||^2 \leq \sum_{k=1}^{K}\left(\left(1 + \frac{1}{K-1}\right)\mathbb{E}||w_{\ell,m}^{(r,K)} - w_{g,m}^{(r,k)}||^2 + K\eta_\ell^2\mathbb{E}||\nabla_{w_{g,m}^{(r,k)}}f_m(w_{p,m}^{(r,k)})||^2\right) \tag{149}$$

$$\leq \left(6K^4\eta_\ell^2\mathbb{E}||\nabla f_m(w_g^{(r)})||^2 + 3K^3\eta_\ell^2\sigma_\ell^2 + 12K^2\eta_\ell^2\sigma_\ell^2\mathbb{E}||1 - \psi_{g,m}||^2\right.$$
$$\left. + 4\left(1 + \frac{1}{K-1}\right)K^3\eta_\ell^2\mathbb{E}||\nabla f_m(w_g^{(r)})||^2\mathbb{E}||\psi_{g,m}||^2\right)$$
$$\cdot\sum_{k=1}^{K}\left(1 + \frac{1}{K-1} + 12K^2\eta_\ell^2\beta\right)^k \tag{150}$$

Assuming $\frac{1}{K-1} \geq 12K^2\eta_\ell^2\beta \implies \eta_\ell \leq \frac{1}{K\sqrt{12(K-1)\beta}}$,

$$\mathbb{E}||w_{p,m}^{(r,K)} - w_{p,m}^{(r,0)}||^2 \leq \left(6K^4\eta_\ell^2\mathbb{E}||\nabla f_m(w_g^{(r)})||^2 + 3K^3\eta_\ell^2\sigma_\ell^2 + 12K^2\eta_\ell^2\mathbb{E}||1 - \psi_{g,m}^{(r,k)}||^2\right.$$
$$\left. + 4\left(1 + \frac{1}{K-1}\right)K^3\eta_\ell^2\mathbb{E}||\nabla f_m(w_g^{(r)})||^2\mathbb{E}||\psi_{g,m}^{(r,k)}||^2\sigma_\ell^2\right)3K \tag{151}$$

$$= 18K^5\eta_\ell^2\mathbb{E}||\nabla f_m(w_g^{(r)})||^2 + 9K^4\eta_\ell^2\sigma_\ell^2 + 36K^3\eta_\ell^2\sigma_\ell^2\mathbb{E}||1 - \psi_{g,m}^{(r,k)}||^2$$
$$+ 24K^4\eta_\ell^2\mathbb{E}||\nabla f_m(w_g^{(r)})||^2\mathbb{E}||\psi_{g,m}^{(r,k)}||^2 \tag{152}$$

□

**Lemma E.19** (One round progress of personalized model). *If $m^{th}$ client's objective function $f_m$ satisfies Assumptions E.3, E.4, in Algorithm 2, the following is satisfied:*

$$\mathbb{E}||w_{p,m}^{(r+1,K)} - w_{p,m}^{(r,K)}||^2 \leq 72(1+\eta_\ell^2)K^3\eta_\ell^2 \left(5K(G^2 + B^2\mathbb{E}||\nabla F(w_g^{(r)})||^2) + 12\sigma_\ell^2\right)$$

$$+ 36\left(K\sigma_\ell^2\eta_\ell^2 + \left(\delta_m^\psi + \frac{\delta^\psi}{M}\right)K^2\eta_\ell^2\right)\left(K\sigma_\ell^2\eta_\ell^2 + \left(\delta_m^{w_g} + \frac{\delta^{w_g}}{M}\right)K^2\eta_\ell^2\right)$$

$$+ 12(1+\eta_\ell^2K^2\beta^4)\eta_\ell^2K^2\left(K\sigma_\ell^2\eta_\ell^2 + \left(\delta_m^\psi + \frac{\delta^\psi}{M}\right)K^2\eta_\ell^2\right)$$

$$\left(G^2 + B^2\mathbb{E}||\nabla F(w_g^{(r)})||^2\right)$$

*Proof.*

$$\mathbb{E}\big\|w_{p,m}^{(r+1,K)} - w_{p,m}^{(r,K)}\big\|^2 = \mathbb{E}\big\|w_{p,m}^{(r+1,K)} - w_{p,m}^{(r+1,0)} + w_{p,m}^{(r+1,0)} - w_{p,m}^{(r,K)}\big\|^2 \tag{153}$$

$$\leq 2\mathbb{E}\big\|w_{p,m}^{(r+1,K)} - w_{p,m}^{(r+1,0)}\big\|^2 + 2\mathbb{E}\big\|w_{p,m}^{(r+1,0)} - w_{p,m}^{(r,K)}\big\|^2 \tag{154}$$

Using the Lemmas E.18 and E.14, we proceed as

$$\leq 72(1+\eta_\ell^2)K^3\eta_\ell^2 \left(5K(G^2 + B^2\mathbb{E}||\nabla F(w_g^{(r)})||^2) + 12\sigma_\ell^2\right)$$

$$+ 36\left(K\sigma_\ell^2\eta_\ell^2 + \left(\delta_m^\psi + \frac{\delta^\psi}{M}\right)K^2\eta_\ell^2\right)\left(K\sigma_\ell^2\eta_\ell^2 + \left(\delta_m^{w_g} + \frac{\delta^{w_g}}{M}\right)K^2\eta_\ell^2\right)$$

$$+ 12(1+\eta_\ell^2K^2\beta^4)\eta_\ell^2K^2\left(K\sigma_\ell^2\eta_\ell^2 + \left(\delta_m^\psi + \frac{\delta^\psi}{M}\right)K^2\eta_\ell^2\right)\left(G^2 + B^2\mathbb{E}||\nabla F(w_g^{(r)})||^2\right) \tag{155}$$

$\square$

**Theorem E.20** (Convergence of the Personalized Model for Non-convex Cases). *If each client's objective function $f_m$ satisfies Assumptions E.2, E.3, E.4 using the learning rate $\eta_\ell \leq \frac{1}{K\sqrt{12\beta}}$ in Algorithm 2, then the following convergence holds:*

$$\frac{1}{R}\sum_{r=1}^{R}\mathbb{E}||\nabla f_m(w_{p,m}^{(r,K)})||^2 \leq \frac{2}{R}\left(\mathbb{E}\left[f_m(w_{p,m}^{(1,K)})\right] - \mathbb{E}\left[f_m(w_{p,m}^{(R,K)})\right]\right)$$

$$+ 6(1+\eta_\ell^2)K\left(5K(G^2 + B^2\frac{1}{R}\sum_{r=1}^{R}\mathbb{E}||\nabla F(w_g^{(r)})||^2) + 12\sigma_\ell^2\right)$$

$$+ 3K\eta_\ell^2\left(\sigma_\ell^2 + \left(\delta_m^\psi + \frac{\delta^\psi}{M}\right)K\right)\left(\sigma_\ell^2 + \left(\delta_m^{w_g} + \frac{\delta^{w_g}}{M}\right)K\right)$$

$$+ (1+\eta_\ell^2K^2\beta^4)\eta_\ell^2K\left(\sigma_\ell^2 + \left(\delta_m^\psi + \frac{\delta^\psi}{M}\right)K\right)\left(G^2 + B^2\frac{1}{R}\sum_{r=1}^{R}\mathbb{E}||\nabla F(w_g^{(r)})||^2\right)$$

*Proof.* According to the update rule of Equation 10 and $\beta$-smoothness of $f_m$, we have,

$$f_m(w_{p,m}^{(r+1,K)}) \leq f_m(w_{p,m}^{(r,K)}) + \left\langle\nabla f_m(w_{p,m}^{(r,K)}), w_{p,m}^{(r+1,K)} - w_{p,m}^{(r,K)}\right\rangle + \frac{\beta}{2}||w_{p,m}^{(r+1,K)} - w_{p,m}^{(r,K)}||^2 \tag{156}$$

Taking expectation on both sides,

$$\mathbb{E}\left[f_m(w_{p,m}^{(r+1,K)})\right] \leq \mathbb{E}\left[f_m(w_{p,m}^{(r,K)})\right] + \frac{\beta}{2}\mathbb{E}||w_{p,m}^{(r+1,K)} - w_{p,m}^{(r,K)}||^2$$

$$+ \mathbb{E}\left\langle\nabla f_m(w_{p,m}^{(r,K)}), w_{p,m}^{(r+1,K)} - w_{p,m}^{(r,K)}\right\rangle \tag{157}$$

Using $\langle a, b \rangle = \frac{1}{2}||a||^2 + \frac{1}{2}||b||^2 - \frac{1}{2}||a-b||^2$

$$\mathbb{E}\left[f_m(w_{p,m}^{(r+1,K)})\right] \le \mathbb{E}\left[f_m(w_{p,m}^{(r,K)})\right] + \frac{1}{2}\mathbb{E}||\nabla f_m(w_{p,m}^{(r,K)})||^2 + \left(\frac{\beta+1}{2}\right)\mathbb{E}||w_{p,m}^{(r+1,K)} - w_{p,m}^{(r,K)}||^2$$
$$- \frac{1}{2}\mathbb{E}||\nabla f_m(w_{p,m}^{(r,K)}) - (w_{p,m}^{(r+1,K)} - w_{p,m}^{(r,K)})||^2 \tag{158}$$

$$\le \mathbb{E}\left[f_m(w_{p,m}^{(r,K)})\right] + \frac{1}{2}\mathbb{E}||\nabla f_m(w_{p,m}^{(r,K)})||^2 + \left(\frac{\beta+1}{2}\right)\mathbb{E}||w_{p,m}^{(r+1,K)} - w_{p,m}^{(r,K)}||^2$$
$$- \mathbb{E}||\nabla f_m(w_{p,m}^{(r,K)})||^2 - \mathbb{E}||(w_{p,m}^{(r+1,K)} - w_{p,m}^{(r,K)})||^2 \tag{159}$$

$$\le \mathbb{E}\left[f_m(w_{p,m}^{(r,K)})\right] - \frac{1}{2}\mathbb{E}||\nabla f_m(w_{p,m}^{(r,K)})||^2 + \left(\frac{\beta-1}{2}\right)\mathbb{E}||w_{p,m}^{(r+1,K)} - w_{p,m}^{(r,K)}||^2 \tag{160}$$

Rearranging the terms to put $\frac{1}{2}\mathbb{E}||\nabla f_m(w_{p,m}^{(r,K)})||^2$ at LHS,

$$\frac{1}{2}\mathbb{E}||\nabla f_m(w_{p,m}^{(r,K)})||^2 \le \mathbb{E}\left[f_m(w_{p,m}^{(r,K)})\right] - \mathbb{E}\left[f_m(w_{p,m}^{(r+1,K)})\right] + \left(\frac{\beta-1}{2}\right)\underbrace{\mathbb{E}||w_{p,m}^{(r+1,K)} - w_{p,m}^{(r,K)}||^2}_{\text{Lemma E.19}} \tag{161}$$

$$\mathbb{E}||\nabla f_m(w_{p,m}^{(r,K)})||^2 \le 2\left(\mathbb{E}\left[f_m(w_{p,m}^{(r,K)})\right] - \mathbb{E}\left[f_m(w_{p,m}^{(r+1,K)})\right]\right)$$
$$+ 72\beta(1+\eta_\ell^2)K^3\eta_\ell^2\left(5K(G^2 + B^2\mathbb{E}||\nabla F(w_g^{(r)})||^2) + 12\sigma_\ell^2\right)$$
$$+ 36\beta\left(K\sigma_\ell^2\eta_\ell^2 + \left(\delta_m^\psi + \frac{\delta^\psi}{M}\right)K^2\eta_\ell^2\right)\left(K\sigma_\ell^2\eta_\ell^2 + \left(\delta_m^{w_g} + \frac{\delta^{w_g}}{M}\right)K^2\eta_\ell^2\right)$$
$$+ 12\beta(1+\eta_\ell^2K^2\beta^4)\eta_\ell^2K^2\left(K\sigma_\ell^2\eta_\ell^2 + \left(\delta_m^\psi + \frac{\delta^\psi}{M}\right)K^2\eta_\ell^2\right)$$
$$\cdot\left(G^2 + B^2\mathbb{E}||\nabla F(w_g^{(r)})||^2\right) \tag{162}$$

Taking an average over all the rounds $r \in [R]$,

$$\frac{1}{R}\sum_{r=1}^R\mathbb{E}||\nabla f_m(w_{p,m}^{(r,K)})||^2 \le \frac{2}{R}\left(\mathbb{E}\left[f_m(w_{p,m}^{(1,K)})\right] - \mathbb{E}\left[f_m(w_{p,m}^{(R,K)})\right]\right)$$
$$+ 72\beta(1+\eta_\ell^2)K^3\eta_\ell^2\left(5K(G^2 + B^2\frac{1}{R}\sum_{r=1}^R\mathbb{E}||\nabla F(w_g^{(r)})||^2) + 12\sigma_\ell^2\right)$$
$$+ 36\beta K^2\eta_\ell^4\left(\sigma_\ell^2 + \left(\delta_m^\psi + \frac{\delta^\psi}{M}\right)K\right)\left(\sigma_\ell^2 + \left(\delta_m^{w_g} + \frac{\delta^{w_g}}{M}\right)K\right)$$
$$+ 12\beta(1+\eta_\ell^2K^2\beta^4)\eta_\ell^4K^3\left(\sigma_\ell^2 + \left(\delta_m^\psi + \frac{\delta^\psi}{M}\right)K\right)\left(G^2 + B^2\frac{1}{R}\sum_{r=1}^R\mathbb{E}||\nabla F(w_g^{(r)})||^2\right) \tag{163}$$

Assuming $12K^2\eta_\ell^2\beta \le 1 \le 1 \implies \eta_\ell \le \frac{1}{K\sqrt{12\beta}}$,

$$\frac{1}{R}\sum_{r=1}^R\mathbb{E}||\nabla f_m(w_{p,m}^{(r,K)})||^2 \le \frac{2}{R}\left(\mathbb{E}\left[f_m(w_{p,m}^{(1,K)})\right] - \mathbb{E}\left[f_m(w_{p,m}^{(R,K)})\right]\right)$$
$$+ 6(1+\eta_\ell^2)K\left(5K(G^2 + B^2\frac{1}{R}\sum_{r=1}^R\mathbb{E}||\nabla F(w_g^{(r)})||^2) + 12\sigma_\ell^2\right)$$
$$+ 3K\eta_\ell^2\left(\sigma_\ell^2 + \left(\delta_m^\psi + \frac{\delta^\psi}{M}\right)K\right)\left(\sigma_\ell^2 + \left(\delta_m^{w_g} + \frac{\delta^{w_g}}{M}\right)K\right)$$
$$+ (1+\eta_\ell^2K^2\beta^4)\eta_\ell^2K\left(\sigma_\ell^2 + \left(\delta_m^\psi + \frac{\delta^\psi}{M}\right)K\right)\left(G^2 + B^2\frac{1}{R}\sum_{r=1}^R\mathbb{E}||\nabla F(w_g^{(r)})||^2\right) \tag{164}$$

Plugging in Theorem E.12 to get bounds on $\sum_{r=1}^R\mathbb{E}||\nabla F(w_g^{(r)})||^2$ would get us bounds on $\frac{1}{R}\sum_{r=1}^R\mathbb{E}||\nabla f_m(w_{p,m}^{(r,K)})||^2$. $\qquad\square$

