705 $\qquad\qquad\qquad\qquad\qquad\qquad\qquad\qquad\qquad\qquad\qquad\qquad\qquad\qquad\qquad\qquad\qquad\qquad\qquad\qquad\qquad\qquad\qquad\square$

706 **Lemma D.8** (Deviation of the personalized model from the global model). *If $m^{th}$ client's objective function*
707 *$f_m$ satisfies Assumptions D.1, D.2, D.3, and condition $\eta_\ell \leq \min\left( \frac{1}{\beta\sqrt{2K(K-1)}}, \frac{1}{\beta\sqrt{2K}} \right)$ in Algorithm 2, the*
708 *following is satisfied:*

$$\mathbb{E}||w_{p,m}^{(r,k)} - w_{g,m}^{(r,0)}||^2 \leq 16k^3\eta_\ell^2 \mathbb{E}||1 - \psi_{g,m}^{(r,k)}||^2 \mathbb{E}||\nabla f_m(w_g^{(r)})||^2 + 8k\eta_\ell^2\sigma_\ell^2 \mathbb{E}||\psi_{g,m}^{(r,k)}||^2$$
$$+ 12K^2\eta_\ell^2 \mathbb{E}||1 - \psi_{g,m}^{(r,k)}||^2 \mathbb{E}||\nabla f_m(w_g^{(r)})||^2 + 6K\eta_\ell^2\sigma_\ell^2 \mathbb{E}||\psi_{g,m}^{(r,k)}||^2$$

*Proof.*

$$\mathbb{E}||w_{p,m}^{(r,k)} - w_{g,m}^{(r,0)}||^2 = \mathbb{E}||\psi_{g,m}^{(r,k)} w_{g,m}^{(r,k)} + (1 - \psi_{g,m}^{(r,k)})w_{\ell,m}^{(r,K)} - w_{g,m}^{(r,0)}||^2 \tag{32}$$

$$= \mathbb{E}||\psi_{g,m}^{(r,k)}(w_{g,m}^{(r,k)} - w_{\ell,m}^{(r,k)}) + (w_{\ell,m}^{(r,K)} - w_{g,m}^{(r,0)})||^2 \tag{33}$$

$$= \mathbb{E}||\psi_{g,m}^{(r,k)}(w_{g,m}^{(r,k)} - w_{g,m}^{(r,0)} + w_{g,m}^{(r,0)} - w_{\ell,m}^{(r,k)}) + (w_{\ell,m}^{(r,K)} - w_{g,m}^{(r,0)})||^2 \tag{34}$$

$$\leq 2\mathbb{E}||\psi_{g,m}^{(r,k)}(w_{g,m}^{(r,k)} - w_{g,m}^{(r,0)})||^2 + 2\mathbb{E}||(1 - \psi_{g,m}^{(r,k)})(w_{\ell,m}^{(r,K)} - w_{\ell,m}^{(r,0)})||^2 \tag{35}$$

709 Using lemmas D.6 and D.7,

$$\mathbb{E}||w_{p,m}^{(r,k)} - w_{g,m}^{(r,0)}||^2 \leq 2\mathbb{E}||\psi_{g,m}^{(r,k)}||^2 \left( 8k^3\eta_\ell^2 \mathbb{E}||\psi_{g,m}^{(r,k)}||^2 \mathbb{E}||\nabla f_m(w_g^{(r)})||^2 + 6K^2\eta_\ell^2 \mathbb{E}||\nabla f_m(w_g^{(r)})||^2 \right)$$
$$+ 2\mathbb{E}||1 - \psi_{g,m}^{(r,k)}||^2 \left( 4k\eta_\ell^2\sigma_\ell^2 + 3K\eta_\ell^2\sigma_\ell^2 \right) \tag{36}$$
$$\leq 16k^3\eta_\ell^2 \mathbb{E}||1 - \psi_{g,m}^{(r,k)}||^2 \mathbb{E}||\nabla f_m(w_g^{(r)})||^2 + 8k\eta_\ell^2\sigma_\ell^2 \mathbb{E}||\psi_{g,m}^{(r,k)}||^2$$
$$+ 12K^2\eta_\ell^2 \mathbb{E}||1 - \psi_{g,m}^{(r,k)}||^2 \mathbb{E}||\nabla f_m(w_g^{(r)})||^2 + 6K\eta_\ell^2\sigma_\ell^2 \mathbb{E}||\psi_{g,m}^{(r,k)}||^2 \tag{37}$$

710 $\qquad\qquad\qquad\qquad\qquad\qquad\qquad\qquad\qquad\qquad\qquad\qquad\qquad\qquad\qquad\qquad\qquad\qquad\qquad\qquad\qquad\qquad\qquad\square$