# OpenReview forum: "Flow: Per-instance Personalized Federated Learning"
_NeurIPS.cc/2023/Conference — NeurIPS 2023 poster_

### Official Review · Reviewer_MbeB · 2023-07-05

**Soundness:** 3 good
**Presentation:** 3 good
**Contribution:** 3 good
**Rating:** 5
**Confidence:** 4

**Summary:**

The paper proposes a per-instance and per-client personalization approach Flow that creates personalized models via dynamic routing to improve both the performance of the personalized model and the generalizability of the global model. Convergence analysis for both global and personalized models are provided. Empirical evaluation shows that Flow can achieve better performance in terms of generalization and personalized accuracy on several vision and language tasks in cross-device FL setting.

**Strengths:**

S1: The paper provides two essential observations that limits the performance of existing personalized FL methods, which well motivates the proposal of training dynamic network to benefit from both the global model and the personalized model.

S2: The basic idea behind Flow that improving the performance of the personalized model via dynamic routing is very intuitive. Theoretical analysis and experimental evaluation also show the effectiveness.

S3: The experiments are well designed which not only show the superior performance of the proposed Flow against existing state-of-the-art personalized FL methods, but also give in-depth analysis of how the proposed dynamic routing strategy works.

S4: The paper is well-written and very easy-to-follow.


**Weaknesses:**

W1: The routing module is not very intuitive to me. Why the model is designed as a fully connected network rather than a sequential model such as RNN? The authors should elaborate more on this issue.

W2: The effectiveness of the alternative training for the global parameters (Equation 5) is not validated. More evaluations are needed.


**Questions:**

Q1: Why the routing model is designed as a fully connected network rather than a sequential model such as RNN? See W1.

Q2: Whether the alternative training of the global parameters is effective? See W2.

Q3: Theorem 4.1 and 4.2 are not the standard form of convergence because they do not show the global optima.


**Limitations:**

See W1-W2.

---

> ### Author Rebuttal · Authors · 2023-08-08
>
> **W1/Q1: Routing module being fully-connected network**
>
> A: Our first choice was a fully-connected network for the policy module because that would take the least amount of computation resources and time. Besides, we did not have any reasons to justify the use of RNNs to generate probabilities of using local or global layers given an input representation. It is because we are already using hidden states as intermediate representations for the policy modules for global models based on RNNs.
>
> **W2/Q2: Effectiveness of the alternative training of the global model**
>
> A: We experimented with three modes of training for the global model: (a) Global model trained first, then the policy module, (b) Policy module trained first, then the global model, (c) Global model and Policy module trained alternatively. The results are shown in Figure 3, in the PDF attached to the global response titled “Author Rebuttal by Authors”. We see that the alternate training results in a more stable training compared to the other two modes of training. These results are in conformance with other works [11, 12] which have also used alternative training for policy and model weights training.
>
> [11] DARTS: Differentiable Architecture Search (Liu et al., ICLR 2019)
>
> [12] Searching for A Robust Neural Architecture in Four GPU Hours (Dong et al., CVPR 2019)
>
> **Q3: Global optima in theorems**
>
> A: The theorems 4.1 and 4.2 are for non-convex cases, where the standard form does not have the global optima [13]. The strong-convex and general-convex cases are covered in Appendix D, where Theorem D.9 (for the global model) and Theorem D.16 (for the personalized model) have the global optima.
>
> [13] SCAFFOLD: Stochastic Controlled Averaging for Federated Learning (Appendix D.2, Theorem V) (Karimireddy et al., ICML 2020)

---

### Official Review · Reviewer_cLqQ · 2023-07-06

**Soundness:** 3 good
**Presentation:** 3 good
**Contribution:** 3 good
**Rating:** 6
**Confidence:** 5

**Summary:**

The paper proposes a per-instance personalized federated learning algorithm called Flow. The algorithm creates dynamic personalized models that adapt to each client's individual data instances. The personalized models allow each instance to determine whether it prefers the local parameters or the global counterpart for making predictions, resulting in improved accuracy. The paper provides theoretical analysis and empirical results demonstrating the superiority of Flow compared to state-of-the-art personalization approaches.

**Strengths:**

1) The paper proposes a novel approach for per-instance personalized federated learning, which is not explored in previous work.

2) The proposed problem “each clients’ data instance is limited to using its personalized model for prediction, while some instances could benefit from better generalization of the global model” is a real PFL problem need to be solved.

3) Both CV and NLP datasets were used for the experiment evaluation. The logic of the paper is clear and easy to understand.


**Weaknesses:**

1) There are additional computational and communication overheads that are not analyzed inside the paper.

2) some hyperparameters are not introudced: for example, the values of K1 and K2.

3) There are no converge curves compared to other baseline algorithms.


**Questions:**

- What is split data used for?
- Why is there no "Ablation on Dynamic Routing Component" like Figure 4(c)  about other datasets except for stackoverflow. I am very interested in the results, but I cannot find them in the appendix.
- In Table 2, why did LGFedAvg's Acc_g achieve a relatively high score and even surpass FedAvg at the CIFAR10(0.6) setting? Didn't LGFedAvg only aggregate classification layers? In terms of my own experiments, it is reasonable that it should be worse than the fully aggregated FedAvg.
- the seed selection "0, 44, and 56" seems deliberate to me.
- Dynamic routing is done by layer-wise to select the layer of the local model or the layer of the global model. My concern is how to ensure that the input space and output space of the local or global model are matched.


**Limitations:**

Instead of using a real-world federal learning dataset, the division is performed on a general NLP/CV dataset. Routing module may cause the risk of leaking client privacy distribution.

---

> ### Author Rebuttal · Authors · 2023-08-08
>
> **W1:  Computational and communication overheads**
>
> A: Please refer to the answer to W2/Q2 of reviewer QKWb
>
> **W2: Hyperparameters**
>
> A: For Stackoverflow and Shakespeare, $K = K_1 = K_2$  is set to 3 and 5 respectively. For EMNIST, CIFAR10/100 variations, $K = K_1 = K_2$  is set to 3. These hyperparameters, along with the rest of the model and training hyperparameters are mentioned in Appendix B.
>
> **W3: Convergence curves**
>
> A:  Please refer to Figure 5, in the PDF attached to the global response titled “Author Rebuttal by Authors”, to see the conference curves for the global model (generalized accuracy). We will add curves for both the generalized and personalized accuracies in the final version of the paper.
>
> **Q1: Utility of the split data**
>
> A: We experimented with the Stackoverflow Next Word Prediction task on initial rounds for the local and global training dataset splits. The plot is attached as Figure 1, in the PDF attached to the global response titled “Author Rebuttal by Authors”. We observe that for the dataset split size ratio of 0.75:0.25 for local and global datasets respectively, the global model does not get sufficient samples to converge, resulting in worse personalized model performance since the personalized model is based on the global model. While a split of 0.25:0.75 for local and global datasets has closer performance to that of a 0.50:0.50 split, lesser data (and hence fewer iterations) to the local model leads to local model weights being similar to that of global model, diminishing the impact of personalization.
>
> **Q2: Ablation on Dynamic Routing Component (for rest of the datasets)**
>
> A: You can find the results on other datasets in the PDF attached to the global response titled “Author Rebuttal by Authors” as Figure 4. We will add the results in the appendix. Thank you for pointing it out.
>
> **Q3: Global accuracy of LGFedAvg**
>
> A: The global accuracy of LGFedAvg for CIFAR10 (0.6) is indeed lower than FedAvg, as it is for the following other datasets too: Stackoverflow, Shakespeare, EMNIST, CIFAR10 (0.1). We do see LGFedAvg surpassing FedAvg CIFAR100 (0.1 and 0.6) setups [Table 4, Appendix C.1], which can be attributed to poor performance of FedAvg due to higher class count across clients, which is also displayed by a decrease in CIFAR100 performance as compared to CIFAR10 for all baselines.
>
> **Q4: Seed selection**
>
> A: The seeds were randomly chosen. Choosing different seeds doesn’t impact the results. We have re-run Stackoverflow experiment with seeds in {1, 2, 3, 4, 5 , 6, 7, 8, 9}, the results show no significant difference. See Figure 2, in the PDF attached to the global response titled “Author Rebuttal by Authors”:
>
> **Q5: Enforcing that the input and output spaces of the local and global models are the same**
>
> A: Like many other personalization works [5, 6, 7] in federated learning, the assumption is that the personalized or local models have the same architecture as the global model. To ensure that the input space has been matched, during training, we have used a soft policy. This means that the input to an intermediate layer is a linear combination of outputs from both the local and the global layers. During inference, even with the hard policy, the outputs from a layer can be fed to the next layer of both the global and the local model because of the same model architecture. The similar logic applies for the output space.
>
> [5] Adaptive Personalized Federated Learning (Deng et al., arxiv 2003.13461)
>
> [6] Ditto: Fair and Robust Federated Learning Through Personalization (Li et al., ICML 2021)
>
> [7] Three Approaches for Personalization with Applications to Federated Learning (Mansour et al., 2002.10619)
>
> **L1: Real-world datasets**
>
> A: We have picked both artificial (CIFAR10/100) and real-world heterogeneous datasets (EMNIST, Stackoverflow, Shakespeare) especially suited for the federated setting (more details in Appendix B). These datasets have been used in many federated learning literatures [8, 9, 10] to address the heterogeneity issue.
>
> [8] Adaptive Federated Optimization (Reddi et al., ICLR 2021)
>
> [9] Motley: Benchmarking Heterogeneity and Personalization in Federated Learning (Wu et al., arxiv 2206.09262)
>
> [10] Communication-Efficient Learning of Deep Networks from Decentralized Data (McMahan et al., AISTATS 2017)
>
> **L2: Routing module may cause the risk of leaking client privacy distribution**
>
> A: With Flow, we aim to improve the performance of personalization in federated settings, by introducing a per-instance and dynamically personalized model. The interplay between personalization for higher accuracy and other techniques for better privacy guarantees is another interesting topic to explore which we leave to future work.

---

### Official Review · Reviewer_QKWb · 2023-07-08

**Soundness:** 3 good
**Presentation:** 3 good
**Contribution:** 3 good
**Rating:** 6
**Confidence:** 3

**Summary:**

This paper presents a novel approach to address the data heterogeneity issue in Federated Learning, resulting in improved accuracy for clients. The proposed approach, called Flow, is a per-instance and per-client personalized Federated Learning algorithm that uses dynamic routing to create adaptive personalized models for each client's individual data instances. The paper provides theoretical analysis on the convergence of Flow and empirically demonstrates its superiority in improving clients' accuracy compared to state-of-the-art personalization approaches on both vision and language-based tasks.

**Strengths:**

The paper presents a novel approach to address the data heterogeneity issue in Federated Learning, which is a significant challenge in real-world applications.
The proposed approach, called Flow, is a per-instance and per-client personalized Federated Learning algorithm that uses dynamic routing to create adaptive personalized models for each client's individual data instances.
The paper provides theoretical analysis on the convergence of Flow, which adds to the credibility of the proposed approach.
The paper empirically demonstrates the superiority of Flow in improving clients' accuracy compared to state-of-the-art personalization approaches on both vision and language-based tasks.
The paper is well-structured and easy to follow, with clear explanations of the proposed approach and experimental results.

**Weaknesses:**

The motivation of this paper is not convincing enough for me, how can per-instance FL solve the challenges is not clear.
The paper does not provide a detailed analysis of the computational and communication costs of Flow, which could be important factors in practical applications.

**Questions:**

see above

**Limitations:**

see above

---

> ### Author Rebuttal · Authors · 2023-08-08
>
> **W1/Q1: Motivation behind Flow. How does per-instance FL solve the stated challenges?**
>
> A: The challenge we are addressing with Flow is the limited performance improvement from personalization in federated learning. Our empirical results indicate that the above challenge comes from two reasons: (a) Some clients’ personalized models are not able to outperform the global model (and therefore defeating the purpose of personalization), and (b) For the clients’ personalized models which do have better accuracy compared to the global model, some instances of those clients are still incorrectly classified by the personalized models but correctly classified by the global model.
>
> Motivated by the above empirical results, we develop Flow to allow each instance on a client to choose the best module (the local model layers or the global model layers) to execute. Flow  creates a dynamic personalized model that combines the local and the global model using a policy module. For each layer, the policy module decides whether to use a local layer or a global layer depending on each instance of the client. The policy module makes Flow adaptive to each client’s individual data instances (per-instance) as well as their data distribution (per-client). The dynamic personalized model improves the performance of personalization because (1) It increases the number of clients that favor the personalized model by 1.15-1.34%  (2) It decreases the number of misclassified instances of a client’s personalized model from 4.74-7.93% of best performing baselines to 1.12-2.42%.
>
> **W2/Q2: Computational and communication costs of Flow**
>
> A: We agree with reviewers that Flow introduces additional storage and computational overhead compared to the canonical method FedAvg with Fine Tuning (noted as FedAvgFT).
>
> However, compared to other state-of-the-art personalization methods such as Ditto and APFL, Flow requires similar or even less storage and computational overhead. To illustrate, the table below compares the storage, computational overhead, and communication cost for personalized models from Flow and baselines using the CNN for EMNIST, and RNN for Stackoverflow. We will add the table and similar analysis for all the models used in the paper in the appendix of the final version.
>
> |  | Local Storage of Personalized Model (unit: parameter count) for general case | Local Storage of Personalized Model (unit: parameter count) for Stackoverflow RNN case | Computational Overhead of Personalized Model of the RNN used for Stackoverflow (unit: FLOPs for training) | Communication Cost (unit: parameter count) for general case | Communication Cost (unit: parameter count) for Stackoverflow RNN case |
> |---|---|---|---|---|---|
> | FedAvgFT | \|$w_g$\|  | 72.38M | Not Applicable | \|$w_g$\| | 72.38M |
> | knnPer | \|$w_g$\| + #instances * \|intermediate representation\| | 72.42M | 12.46M | \|$w_g$\| | 72.38M |
> | PartialFed | \|$w_g$\| + 2 * #layers of $w_g$ | 72.38M | 36.9M | \|$w_g$\| | 72.38M |
> | APFL | 3 * \|$w_g$\| | 217.14M | 73.8M | \|$w_g$\| | 72.38M |
> | Ditto | 2 * \|$w_g$\| | 144.76M | 36.9M | \|$w_g$\| | 72.38M |
> | FedRep | \|$w_g$\| | 72.38M | 51M | \|$w_g$ base\| | 70.98M |
> | LGFedAvg | \|$w_g$\| | 72.38M | 10.5M | \|$w_g$ head\| | 1.39M |
> | HypCluster | 2 * \|$w_g$\| | 144.76M | 36.9M | \|$w_g$\| | 72.38M |
> | Flow (ours) | 2 * \|$w_g$\| + \|$\psi_g$\| | 145.651M | 39.57M  | \|$w_g$\| + \|$\psi_g$\| | 73.271M |
>
> In the above table, $w_g$ is the global model
>
> For Flow, $\psi_g$ is the policy module. For our experiments on Flow, the policy module has 1.23% for Stackoverflow RNN, 8.39% for Shakespeare RNN, 27.86% for EMNIST CNN, 35.51% for CIFAR10/100 parameters of $w_g$.
>
> For EMNIST CNN, for one epoch, the dynamic routing takes 0.3M FLOPs, while the rest of the model computations take 2.9M FLOPs. Hence the overhead is 10.34%. For Stackoverflow RNN one epoch, the dynamic routing takes 0.89M FLOPs, while the rest of the model computations take 12.34M FLOPs, making the overhead 7.21%. We believe the additional computation overhead from the routing module is relatively small.
>
> Below, we will highlight the computational and storage analysis on Stackoverflow dataset for the best performing methods, APFL and Hypcluster. The computational overhead is calculated for one epoch of training, which is then multiplied with however many epochs the baseline needs for convergence. APFL needs to train two separate local models for more numbers of epochs than Flow, hence the higher computational overhead. In comparison with HypCluster, Flow introduces 7.23% overhead. With respect to both APFL and HypCluster, Flow only introduces 1.23% storage overhead.

---

### Official Review · Reviewer_iGNe · 2023-07-09

**Soundness:** 3 good
**Presentation:** 3 good
**Contribution:** 3 good
**Rating:** 5
**Confidence:** 4

**Summary:**

This paper proposed the Flow, which achieves per-instance and per-client personalized federated learning by dynamic routing between a global model and a personalized model. Flow creates a personalized model that adapts to each client's data instances and distribution. In each federated learning round, clients have both global and local model parameters, which are fine-tuned to fit the local data distribution. The dynamic routing module predicts for each instance using either local or global model parameters based on the instance preference, thereby improving the accuracy of the personalized model. The authors also improve the accuracy of the global model by identifying the instances on the clients that are in agreement with the global data distribution. The extensive experiments have been conducted to evaluate the advantage of the proposed Flow.

**Strengths:**

1) The authors proposed a new personalized FL algorithm that utilizes the dynamic routing strategy to generate dynamic inference models that best fit the local data, which is a new aspect of personalized FL.
2) The experiments are comprehensive, and the code is included.
3) The writing is clear.


**Weaknesses:**

1）The Flow proposed in this paper takes a different approach compared to general local model personalization methods in FL literature. In some typical personalized FL algorithms, a large global model is generated on the server. When the global model is deployed for the personalized requirements of clients, NAS or dynamic routing are adopted to get a smaller model which can better fits the local data distribution (e.g., OFA and AdaptiveNet[1-2]). In contrast, the authors suggest storing both the personalized local models and global models on the client, and dynamically selecting the inference path based on routing decisions, which can adapt to local data distribution and instances. However, this approach introduces additional local storage and computational overhead. Especially when the global model has a large parameter size, the proposed algorithm is limited.
2) Some metrics proposed in the paper are heuristic. For example, in line 124, the authors divide the local dataset into two parts because they considered that fine-tuning the local model with the entire local data may lead to overfitting.
3) The selection of baselines lacks some personalized FL algorithms based on NAS or NAS-inspired methods. Flow and these baselines are both algorithms based on path-searching, and it would be interesting to supplement the related comparative experimental results.

4) Some claims are unclear. For example, in the abstract, it is a little confused that the personalized models could achieve lower accuracy than the global model.
It seems that the per-instance personalization FL is more fine-grainded than the current personalization FL. However, this would also introduce additional overhead. The authors need to discuss the overhead of this operation.

5) The authors only consider two types of models for the clients to select: i.e., the persoanlized model and the global model. However, in conditional computing paradigm, such as mixture of expert (MoE) framework, each instance can be routed to different kinds of experts.

References:
1) Once-for-All: Train One Network and Specialize it for Efficient Deployment
2) AdaptiveNet: Post-deployment Neural Architecture Adaptation for Diverse Edge Environments


**Questions:**

1) Is there any experimental results to support the heuristic step in line 124? What does the proportion of data used for fine-tuning impact the final experimental performance?
2) How does it compare to the existing flexible personalized FL algorithms, such as some algorithms based on NAS?
3) What is the additional training time overhead for fine-tuning the local models?
4) What is the additional time overhead for dynamic routing decisions during inference?
5) Does the current reuslts can be extended to the more general dynamic routing framework, such as MoE?


**Limitations:**

Please refer to the weakness discussed above.

---

> ### Author Rebuttal · Authors · 2023-08-08
>
> **W1/Q3: Additional local storage and computational overhead of a personalized model (Especially with larger models)**
>
> A: Please refer to the answer to W2/Q2 of reviewer QKWb
>
> **W2/Q1: Heuristics to divide the local dataset**
>
> A: We experimented with the Stackoverflow Next Word Prediction task on initial rounds for the local and global training dataset splits. The plot is attached as Figure 1, in the PDF attached to the global response titled “Author Rebuttal by Authors”. We observe that for the dataset split size ratio of 0.75:0.25 for local and global datasets respectively, the global model does not get sufficient samples to converge, resulting in worse personalized model performance since the personalized model is based on the global model. While a split of 0.25:0.75 for local and global datasets has closer performance to that of a 0.50:0.50 split, lesser data (and hence fewer iterations) to the local model leads to local model weights being similar to that of global model, diminishing the impact of personalization.
>
> **W3/Q2: NAS-based methods as baselines**
>
> A: Our literature search yielded two papers on NAS under federated setup for personalization:
>
> [1] Personalized Neural Architecture Search for Federated Learning (FL Workshop @ NeurIPS ‘21)
>
> In FedPNAS, the personalization happens similar to FedRep, as only the head of the model is getting personalized. For personalization, they need to compute hessian of the personalized parameters, which is computationally expensive, hence they restrict personalization to a smaller subset of the parameters. Moreover their search space is computationally expensive, on the scale of $10^6$. Further, this paper’s experimental depth with respect to tasks and datasets is quite small, and code implementing the algorithm is not available to reproduce the results or adapt to a broader array of tasks and datasets.
>
> [2] SPIDER: Searching Personalized Neural Architecture for Federated Learning (FL Workshop @ AAAI ‘22)
>
> The setting of SPIDER demands an over-parameterized global model, and finding a suitable personalized model path in that over-parameterized network. Besides the communication and computation overhead of the bigger global model, and storage of a separate local model, the goal of this work is to find a personalized architecture per-client to fit a computational budget. It still doesn’t address the issue of personalization harming the client’s accuracy due to a few instances which fall under the global distribution. Besides, SPIDER requires all the clients to participate throughout the federated training process as the personalized models would be progressively updated with a new path, which is not a practical scenario for cross-device FL, which Flow is targeting.
>
> Since the reviewer mentioned the following works, we also comment on them below:
>
> [3] Once-for-All: Train One Network and Specialize it for Efficient Deployment
>
> [4] AdaptiveNet: Post-deployment Neural Architecture Adaptation for Diverse Edge Environments
>
> Due to the heterogeneous nature of FL clients, OFA and AdaptiveNet face the same challenges as FedAvg (i.e., slow or poor convergence). This is because they require fully training a super-network first (which will be based on FedAvg), and then progressively shrinking it or doing the post-deployment adaptation to make the inference faster and more accurate. Note that one of our baselines KnnPer also personalizes the global model after training the global model to converge using FedAvg. The results of KnnPer show that “personalization after training a global model to converge” performs worse than “personalization during training”.
>
> We also want to draw the reviewer’s attention to another baseline we use, PartialFed, which makes a layer-wise choice between global and local models to create a personalized model, but sends the personalized models back to the server in lieu of a separate global model. This method faces the same issues as FedAvg in the heterogeneous FL setup.
>
> **W4/Q4: Overhead of the dynamic routing operation**
>
> A: We have discussed the dynamic routing operation overheads in answer to W2/Q2 of reviewer QKWb, under the table.
>
> Moreover, from Table 1, we can see that compared to some other state-of-the-art personalization techniques (e.g., FedRep, APFL), Flow requires similar or even less computations.
>
> **W5/Q5: Extension to a general dynamic routing framework**
>
> A: With this work, under the context of federated learning, we have only considered two models because for each client, there can only be one global model and one local model. Only using two models for dynamic routing is not a limitation of this work, but a requirement for our setting of personalized federated learning.
>
> Still, we can extend this framework to more than 2 “experts” by having softmax of Equation 2 accommodate as many routing path probabilities as there are experts. And then taking the linear combination of all the weighted model outputs in Equation 3.

---

### Author Rebuttal · Authors · 2023-08-08

We are attaching a one-page PDF containing more experiment results on

Figure 1: Different dataset splits for trainings of local and global parameters

Figure 2: More runs on Stackoverflow dataset for Flow, on seeds {1, 2, 3, 4, 5, 6, 7, 8, 9}

Figure 3: Different training modes on EMNIST to showcase how alternate training of global and policy parameters result in more stable training

Figure 4: Ablation on Dynamic Routing Component for rest of the datasets

Figure 5: Convergence curves for all the datasets

---

### Comment · Area_Chair_Kv92 · 2023-08-21
**to reviewers**

Dear reviewers,

Thank you for your service in reviewing for NeurIPS this year!

Please carefully read the authors' rebuttal and respond to them by the followings:

1) Acknowledge that you have read the rebuttal
2) Indicate whether and how you plan to change your score (if necessary)
3) Ask the authors follow-up questions (if any)

Thanks again!

Sincerely,
AC

---

### Decision · Program_Chairs · 2023-09-21

**Decision:**

Accept (poster)

**Comment:**

In this paper, the authors proposes a new personalized federated learning method named Flow, which allows the output of each instance is dynamically determined by global or local model, which is new for pFL research. Some analysis and experiments demonstrate its advantage. To this end, I recommend accepting this submission.

However, I still have the concern about the additional cost, which may limit its application scope. This is also pointed by reviewers; although the authors discuss this one in the rebuttal, this concern is still there.

Please further improve this submission according to all the discussions between reviewers and the authors. Hope they find the discussion useful and make this submission a better one.